# SCORE-BASED GENERATIVE MODELING WITH CRITICALLY-DAMPED LANGEVIN DIFFUSION

**Tim Dockhorn**[1,2,3,∗]      **Arash Vahdat**[1]      **Karsten Kreis**[1]

[1]NVIDIA      [2]University of Waterloo      [3]Vector Institute
tim.dockhorn@uwaterloo.ca,    {avahdat,kkreis}@nvidia.com

## ABSTRACT

Score-based generative models (SGMs) have demonstrated remarkable synthesis quality. SGMs rely on a diffusion process that gradually perturbs the data towards a tractable distribution, while the generative model learns to denoise. The complexity of this denoising task is, apart from the data distribution itself, uniquely determined by the diffusion process. We argue that current SGMs employ overly simplistic diffusions, leading to unnecessarily complex denoising processes, which limit generative modeling performance. Based on connections to statistical mechanics, we propose a novel *critically-damped Langevin diffusion* (CLD) and show that CLD-based SGMs achieve superior performance. CLD can be interpreted as running a joint diffusion in an extended space, where the auxiliary variables can be considered "velocities" that are coupled to the data variables as in Hamiltonian dynamics. We derive a novel score matching objective for CLD and show that the model only needs to learn the score function of the conditional distribution of the velocity given data, an easier task than learning scores of the data directly. We also derive a new sampling scheme for efficient synthesis from CLD-based diffusion models. We find that CLD outperforms previous SGMs in synthesis quality for similar network architectures and sampling compute budgets. We show that our novel sampler for CLD significantly outperforms solvers such as Euler–Maruyama. Our framework provides new insights into score-based denoising diffusion models and can be readily used for high-resolution image synthesis. Project page and code: https://nv-tlabs.github.io/CLD-SGM.

## 1 INTRODUCTION

Score-based generative models (SGMs) and denoising diffusion probabilistic models have emerged as a promising class of generative models (Sohl-Dickstein et al., 2015; Song et al., 2021c;b; Vahdat et al., 2021; Kingma et al., 2021). SGMs offer high quality synthesis and sample diversity, do not require adversarial objectives, and have found applications in image (Ho et al., 2020; Nichol & Dhariwal, 2021; Dhariwal & Nichol, 2021; Ho et al., 2021), speech (Chen et al., 2021; Kong et al., 2021; Jeong et al., 2021), and music synthesis (Mittal et al., 2021), image editing (Meng et al., 2021; Sinha et al., 2021; Furusawa et al., 2021), super-resolution (Saharia et al., 2021; Li et al., 2021), image-to-image translation (Sasaki et al., 2021), and 3D shape generation (Luo & Hu, 2021; Zhou et al., 2021). SGMs use a diffusion process to gradually add noise to the data, transforming a complex data distribution into an analytically tractable prior distribution. A neural network is then utilized to learn the score function—the gradient of the log probability density—of the perturbed data. The learnt scores can be used to solve a stochastic differential equation (SDE) to synthesize new samples. This corresponds to an iterative denoising process, inverting the forward diffusion.

In the seminal work by Song et al. (2021c), it has been shown that the score function that needs to be learnt by the neural network is uniquely determined by the forward diffusion process. Consequently, the complexity of the learning problem depends, other than on the data itself, only on the diffusion. Hence, the diffusion process is the key component of SGMs that needs to be revisited to further improve SGMs, for example, in terms of synthesis quality or sampling speed.

---

∗Work done during internship at NVIDIA.

Figure 1: In critically-damped Langevin diffusion, the data $\mathbf{x}_t$ is augmented with a velocity $\mathbf{v}_t$. A diffusion coupling $\mathbf{x}_t$ and $\mathbf{v}_t$ is run in the joint data-velocity space (probabilities in **red**). Noise is injected only into $\mathbf{v}_t$. This leads to smooth diffusion trajectories (**green**) for the data $\mathbf{x}_t$. Denoising only requires $\nabla_{\mathbf{v}_t} \log p(\mathbf{v}_t | \mathbf{x}_t)$.

Inspired by statistical mechanics (Tuckerman, 2010), we propose a novel forward diffusion process, the *critically-damped Langevin diffusion (CLD)*. In CLD, the data variable, $\mathbf{x}_t$ (time $t$ along the diffusion), is augmented with an additional "velocity" variable $\mathbf{v}_t$ and a diffusion process is run in the joint data-velocity space. Data and velocity are coupled to each other as in Hamiltonian dynamics, and noise is injected only into the velocity variable. As in Hamiltonian Monte Carlo (Duane et al., 1987; Neal, 2011), the Hamiltonian component helps to efficiently traverse the joint data-velocity space and to transform the data distribution into the prior distribution more smoothly. We derive the corresponding score matching objective and show that for CLD the neural network is tasked with learning only the score of the conditional distribution of velocity given data $\nabla_{\mathbf{v}_t} \log p_t(\mathbf{v}_t | \mathbf{x}_t)$, which is arguably easier than learning the score of diffused data directly. Using techniques from molecular dynamics (Bussi & Parrinello, 2007; Tuckerman, 2010; Leimkuhler & Matthews, 2013), we also derive a new SDE integrator tailored to CLD's reverse-time synthesis SDE.

We extensively validate CLD and the novel SDE solver: **(i)** We show that the neural networks learnt in CLD-based SGMs are smoother than those of previous SGMs. **(ii)** On the CIFAR-10 image modeling benchmark, we demonstrate that CLD-based models outperform previous diffusion models in synthesis quality for similar network architectures and sampling compute budgets. We attribute these positive results to the Hamiltonian component in the diffusion and to CLD's easier score function target, the score of the velocity-data conditional distribution $\nabla_{\mathbf{v}_t} \log p_t(\mathbf{v}_t | \mathbf{x}_t)$. **(iii)** We show that our novel sampling scheme for CLD significantly outperforms the popular Euler–Maruyama method. **(iv)** We perform ablations on various aspects of CLD and find that CLD does not have difficult-to-tune hyperparameters.

In summary, we make the following technical contributions: **(i)** We propose CLD, a novel diffusion process for SGMs. **(ii)** We derive a score matching objective for CLD, which requires only the conditional distribution of velocity given data. **(iii)** We propose a new type of denoising score matching ideally suited for scalable training of CLD-based SGMs. **(iv)** We derive a tailored SDE integrator that enables efficient sampling from CLD-based models. **(v)** Overall, we provide novel insights into SGMs and point out important new connections to statistical mechanics.

## 2 BACKGROUND

Consider a diffusion process $\mathbf{u}_t \in \mathbb{R}^d$ defined by the Itô SDE

$$d\mathbf{u}_t = \boldsymbol{f}(\mathbf{u}_t, t)\, dt + \boldsymbol{G}(\mathbf{u}_t, t)\, d\mathbf{w}_t, \quad t \in [0, T], \tag{1}$$

with continuous time variable $t \in [0, T]$, standard Wiener process $\mathbf{w}_t$, drift coefficient $\boldsymbol{f} \colon \mathbb{R}^d \times [0, T] \to \mathbb{R}^d$ and diffusion coefficient $\boldsymbol{G} \colon \mathbb{R}^d \times [0, T] \to \mathbb{R}^{d \times d}$. Defining $\bar{\mathbf{u}}_t := \mathbf{u}_{T-t}$, a corresponding reverse-time diffusion process that inverts the above forward diffusion can be derived (Anderson, 1982; Haussmann & Pardoux, 1986; Song et al., 2021c) (with positive $dt$ and $t \in [0, T]$):

$$d\bar{\mathbf{u}}_t = \left[ -\boldsymbol{f}(\bar{\mathbf{u}}_t, T-t) + \boldsymbol{G}(\bar{\mathbf{u}}_t, T-t)\boldsymbol{G}(\bar{\mathbf{u}}_t, T-t)^\top \nabla_{\bar{\mathbf{u}}_t} \log p_{T-t}(\bar{\mathbf{u}}_t) \right] dt + \boldsymbol{G}(\bar{\mathbf{u}}_t, T-t) d\mathbf{w}_t, \tag{2}$$

where $\nabla_{\bar{\mathbf{u}}_t} \log p_{T-t}(\bar{\mathbf{u}}_t)$ is the score function of the marginal distribution over $\bar{\mathbf{u}}_t$ at time $T - t$.

The reverse-time process can be used as a generative model. In particular, Song et al. (2021c) model data $\mathbf{x}$, setting $p(\mathbf{u}_0) = p_{\text{data}}(\mathbf{x})$. Currently used SDEs (Song et al., 2021c; Kim et al., 2021) have drift and diffusion coefficients of the simple form $\boldsymbol{f}(\mathbf{x}_t, t) = f(t)\mathbf{x}_t$ and $\boldsymbol{G}(\mathbf{x}_t, t) = g(t)\boldsymbol{I}_d$. Generally, $\boldsymbol{f}$ and $\boldsymbol{G}$ are chosen such that the SDE's marginal, equilibrium density is approximately Normal at time $T$, i.e., $p(\mathbf{u}_T) \approx \mathcal{N}(\mathbf{0}, \boldsymbol{I}_d)$. We can then initialize $\mathbf{x}_0$ based on a sample drawn from a complex

data distribution, corresponding to a far-from-equilibrium state. While the state $\mathbf{x}_0$ relaxes towards equilibrium via the forward diffusion, we can learn a model $\mathbf{s}_{\boldsymbol{\theta}}(\mathbf{x}_t, t)$ for the score $\nabla_{\mathbf{x}_t} \log p_t(\mathbf{x}_t)$, which can be used for synthesis via the reverse-time SDE in Eq. (2). If $\boldsymbol{f}$ and $\boldsymbol{G}$ take the simple form from above, the denoising score matching (Vincent, 2011) objective for this task is:

$$\min_{\boldsymbol{\theta}} \mathbb{E}_{t \sim \mathcal{U}[0,T]} \mathbb{E}_{\mathbf{x}_0 \sim p(\mathbf{x}_0)} \mathbb{E}_{\mathbf{x}_t \sim p_t(\mathbf{x}_t|\mathbf{x}_0)} \left[ \lambda(t) \| \mathbf{s}_{\boldsymbol{\theta}}(\mathbf{x}_t, t) - \nabla_{\mathbf{x}_t} \log p_t(\mathbf{x}_t|\mathbf{x}_0) \|_2^2 \right] \quad (3)$$

If $\boldsymbol{f}$ and $\boldsymbol{G}$ are affine, the conditional distribution $p_t(\mathbf{x}_t|\mathbf{x}_0)$ is Normal and available analytically (Särkkä & Solin, 2019). Different $\lambda(t)$ result in different trade-offs between synthesis quality and likelihood in the generative model defined by $\mathbf{s}_{\boldsymbol{\theta}}(\mathbf{x}_t, t)$ (Song et al., 2021b; Vahdat et al., 2021).

## 3 CRITICALLY-DAMPED LANGEVIN DIFFUSION

We propose to augment the data $\mathbf{x}_t \in \mathbb{R}^d$ with auxiliary *velocity*[1] variables $\mathbf{v}_t \in \mathbb{R}^d$ and utilize a diffusion process that is run in the joint $\mathbf{x}_t$-$\mathbf{v}_t$-space. With $\mathbf{u}_t = (\mathbf{x}_t, \mathbf{v}_t)^\top \in \mathbb{R}^{2d}$, we set

$$\boldsymbol{f}(\mathbf{u}_t, t) \coloneqq \left( \begin{pmatrix} 0 & \beta M^{-1} \\ -\beta & -\Gamma\beta M^{-1} \end{pmatrix} \otimes \boldsymbol{I}_d \right) \mathbf{u}_t, \qquad \boldsymbol{G}(\mathbf{u}_t, t) \coloneqq \begin{pmatrix} 0 & 0 \\ 0 & \sqrt{2\Gamma\beta} \end{pmatrix} \otimes \boldsymbol{I}_d, \quad (4)$$

where $\otimes$ denotes the Kronecker product. The coupled SDE that describes the diffusion process is

$$\begin{pmatrix} d\mathbf{x}_t \\ d\mathbf{v}_t \end{pmatrix} = \underbrace{\begin{pmatrix} M^{-1}\mathbf{v}_t \\ -\mathbf{x}_t \end{pmatrix} \beta dt}_{\text{Hamiltonian component} =: H} + \underbrace{\begin{pmatrix} \mathbf{0}_d \\ -\Gamma M^{-1}\mathbf{v}_t \end{pmatrix} \beta dt + \begin{pmatrix} 0 \\ \sqrt{2\Gamma\beta} \end{pmatrix} d\mathbf{w}_t}_{\text{Ornstein-Uhlenbeck process} =: O}, \quad (5)$$

which corresponds to *Langevin dynamics* in each dimension. That is, each $x_i$ is independently coupled to a velocity $v_i$, which explains the blockwise structure of $\boldsymbol{f}$ and $\boldsymbol{G}$. The *mass* $M \in \mathbb{R}^+$ is a hyperparameter that determines the coupling between the $\mathbf{x}_t$ and $\mathbf{v}_t$ variables; $\beta \in \mathbb{R}^+$ is a constant time rescaling chosen such that the diffusion converges to its equilibrium distribution within $t \in [0, T]$ (in practice, we set $T=1$) when initialized from a data-defined non-equilibrium state and is analogous to $\beta(t)$ in previous diffusions (we could also use time-dependent $\beta(t)$, but found constant $\beta$'s to work well, and therefore opted for simplicity); $\Gamma \in \mathbb{R}^+$ is a *friction* coefficient that determines the strength of the noise injection into the velocities. Notice that the SDE in Eq. (5) consists of two components. The $H$ term represents a Hamiltonian component. Hamiltonian dynamics are frequently used in Markov chain Monte Carlo methods to accelerate sampling and efficiently explore complex probability distributions (Neal, 2011). The Hamiltonian component in our diffusion process plays a similar role and helps to quickly and smoothly converge the initial joint data-velocity distribution to the equilibrium, or prior (see Fig. 1). Furthermore, Hamiltonian dynamics on their own are trivially invertible (Tuckerman, 2010), which intuitively is also beneficial in our situation when using this diffusion for training SGMs. The $O$ term corresponds to an Ornstein-Uhlenbeck process (Särkkä & Solin, 2019) in the velocity component, which injects noise such that the diffusion dynamics properly converge to equilibrium for any $\Gamma>0$. It can be shown that the equilibrium distribution of this diffusion is $p_{\text{EQ}}(\mathbf{u}) = \mathcal{N}(\mathbf{x}; \mathbf{0}_d, \boldsymbol{I}_d) \mathcal{N}(\mathbf{v}; \mathbf{0}_d, M\boldsymbol{I}_d)$ (see App. B.2).

There is a crucial balance between $M$ and $\Gamma$ (McCall, 2010): For $\Gamma^2 < 4M$ (*underdamped* Langevin dynamics) the Hamiltonian component dominates, which implies oscillatory dynamics of $\mathbf{x}_t$ and $\mathbf{v}_t$ that slow down convergence to equilibrium. For $\Gamma^2 > 4M$ (*overdamped* Langevin dynamics) the $O$-term dominates which also slows down convergence, since the accelerating effect by the Hamiltonian component is suppressed due to the strong noise injection. For $\Gamma^2 = 4M$ (*critical damping*), an ideal balance is achieved and convergence to $p_{\text{EQ}}(\mathbf{u})$ occurs as fast as possible in a smooth manner without oscillations (also see discussion in App. A.1) (McCall, 2010). Hence, we propose to set $\Gamma^2 = 4M$ and call the resulting diffusion *critically-damped Langevin diffusion (CLD)* (see Fig. 1).

Diffusions such as the VPSDE (Song et al., 2021c) correspond to overdamped Langevin dynamics with high friction coefficients $\Gamma$ (see App. A.2). Furthermore, in previous works noise is injected directly into the data variables (pixels, for images). In CLD, only the velocity variables are subject to direct noise and the data is perturbed only indirectly due to the coupling between $\mathbf{x}_t$ and $\mathbf{v}_t$.

---

[1] We call the auxiliary variables *velocities*, as they play a similar role as velocities in physical systems. Formally, our velocity variables would rather correspond to physical momenta, but the term momentum is already widely used in machine learning and our mass $M$ is unitless anyway.

### 3.1 SCORE MATCHING OBJECTIVE

Considering the appealing convergence properties of CLD, we propose to utilize CLD as forward diffusion process in SGMs. To this end, we initialize the joint $p(\mathbf{u}_0)=p(\mathbf{x}_0)\,p(\mathbf{v}_0)=p_{\mathrm{data}}(\mathbf{x}_0)\mathcal{N}(\mathbf{v}_0;\mathbf{0}_d,\gamma M \boldsymbol{I}_d)$ with hyperparameter $\gamma<1$ and let the distribution diffuse towards the tractable equilibrium—or prior—distribution $p_{\mathrm{EQ}}(\mathbf{u})$. We can then learn the corresponding score functions and define CLD-based SGMs. Following a similar derivation as Song et al. (2021b), we obtain the score matching (SM) objective (see App. B.3):

$$\min_{\boldsymbol{\theta}} \mathbb{E}_{t\sim\mathcal{U}[0,T]}\mathbb{E}_{\mathbf{u}_t\sim p_t(\mathbf{u}_t)}\left[\lambda(t)\|s_{\boldsymbol{\theta}}(\mathbf{u}_t,t)-\nabla_{\mathbf{v}_t}\log p_t(\mathbf{u}_t)\|_2^2\right] \qquad (6)$$

Notice that this objective requires only the velocity gradient of the log-density of the joint distribution, i.e., $\nabla_{\mathbf{v}_t}\log p_t(\mathbf{u}_t)$. This is a direct consequence of injecting noise into the velocity variables *only*. Without loss of generality, $p_t(\mathbf{u}_t)=p_t(\mathbf{x}_t,\mathbf{v}_t)=p_t(\mathbf{v}_t|\mathbf{x}_t)p_t(\mathbf{x}_t)$. Hence,

$$\nabla_{\mathbf{v}_t}\log p_t(\mathbf{u}_t) = \nabla_{\mathbf{v}_t}\left[\log p_t(\mathbf{v}_t|\mathbf{x}_t)+\log p_t(\mathbf{x}_t)\right]=\nabla_{\mathbf{v}_t}\log p_t(\mathbf{v}_t|\mathbf{x}_t) \qquad (7)$$

This means that in CLD the neural network-defined score model $s_{\boldsymbol{\theta}}(\mathbf{u}_t,t)$ only needs to learn the score of the conditional distribution $p_t(\mathbf{v}_t|\mathbf{x}_t)$, an arguably easier task than learning the score of $p_t(\mathbf{x}_t)$, as in previous works, or of the joint $p_t(\mathbf{u}_t)$. This is the case, because our velocity distribution is initialized from a simple Normal distribution, such that $p_t(\mathbf{v}_t|\mathbf{x}_t)$ is closer to a Normal distribution for all $t\geq0$ (and for any $\mathbf{x}_t$) than $p_t(\mathbf{x}_t)$ itself. This is most evident at $t=0$: The data and velocity distributions are independent at $t=0$ and the score of $p_0(\mathbf{v}_0|\mathbf{x}_0)=p_0(\mathbf{v}_0)$ simply corresponds to the score of the Normal distribution $p_0(\mathbf{v}_0)$ from which the velocities are initialized, whereas the score of the data distribution $p_0(\mathbf{x}_0)$ is highly complex and can even be unbounded (Kim et al., 2021). We empirically verify the reduced complexity of the score of $p_t(\mathbf{v}_t|\mathbf{x}_t)$ in Fig. 2. We find that the score that needs to be learnt by the model is more similar to a score corresponding to a Normal distribution for CLD than for the VPSDE. We also measure the complexity of the neural networks that were learnt to model this score via the squared Frobenius norm of their Jacobians. We find that the CLD-based SGMs have significantly simpler and smoother neural networks than VPSDE-based SGMs for most $t$, in particular when leveraging a mixed score formulation (see next section).

### 3.2 SCALABLE TRAINING

**A Practical Objective.** We cannot train directly with Eq. (6), since we do not have access to the marginal distribution $p_t(\mathbf{u}_t)$. As presented in Sec. 2, we could employ denoising score matching (DSM) and instead sample $\mathbf{u}_0$,

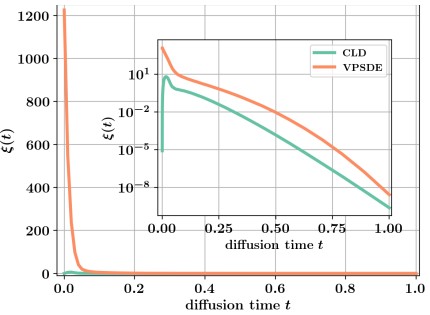

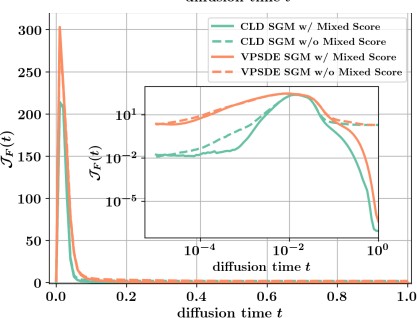

Figure 2: *Top:* Difference $\xi(t)$ (via $L2$ norm) between score of diffused data and score of Normal distribution. *Bottom:* Frobenius norm of Jacobian $\mathcal{J}_F(t)$ of the neural network defining the score function for different $t$. The underlying data distribution is a mixture of Normals. *Insets:* Different axes (see App. E.1 for detailed definitions of $\xi(t)$ and $\mathcal{J}_F(t)$).

and diffuse those samples, which would lead to a tractable objective. However, recall that in CLD the distribution at $t=0$ is the product of a complex data distribution and a Normal distribution over the initial velocity. Therefore, we propose a hybrid version of score matching (Hyvärinen, 2005) and denoising score matching (Vincent, 2011), which we call *hybrid score matching* (HSM). In HSM, we draw samples from $p_0(\mathbf{x}_0)=p_{\mathrm{data}}(\mathbf{x}_0)$ as in DSM, but then diffuse those samples while marginalizing over the full initial velocity distribution $p_0(\mathbf{v}_0)=\mathcal{N}(\mathbf{v};\mathbf{0}_d,\gamma M \boldsymbol{I}_d)$ as in regular SM (HSM is discussed in detail in App. C). Since $p_0(\mathbf{v}_0)$ is Normal (and $\boldsymbol{f}$ and $\boldsymbol{G}$ affine), $p(\mathbf{u}_t|\mathbf{x}_0)$ is also Normal and this remains tractable. We can write this HSM objective as:

$$\min_{\boldsymbol{\theta}} \mathbb{E}_{t\in[0,T]}\mathbb{E}_{\mathbf{x}_0\sim p_0(\mathbf{x}_0)}\mathbb{E}_{\mathbf{u}_t\sim p_t(\mathbf{u}_t|\mathbf{x}_0)}\left[\lambda(t)\|s_{\boldsymbol{\theta}}(\mathbf{u}_t,t)-\nabla_{\mathbf{v}_t}\log p_t(\mathbf{u}_t|\mathbf{x}_0)\|_2^2\right]. \qquad (8)$$

In HSM, the expectation over $p_0(\mathbf{v}_0)$ is essentially solved analytically, while DSM would use a sample-based estimate. Hence, HSM reduces the variance of training objective gradients compared to pure DSM, which we validate in App. C.1. Furthermore, when drawing a sample $\mathbf{u}_0$ to diffuse in

DSM, we are essentially placing an infinitely sharp Normal with unbounded score (Kim et al., 2021) at $\mathbf{u}_0$, which requires undesirable modifications or truncation tricks for stable training (Song et al., 2021c; Vahdat et al., 2021). Hence, with DSM we could lose some benefits of the CLD framework discussed in Sec. 3.1, whereas HSM is tailored to CLD and fundamentally avoids such unbounded scores. Closed form expressions for the perturbation kernel $p_t(\mathbf{u}_t|\mathbf{x}_0)$ are provided in App. B.1.

**Score Model Parametrization.** **(i)** Ho et al. (2020) found that it can be beneficial to parameterize the score model to predict the noise that was used in the reparametrized sampling to generate perturbed samples $\mathbf{u}_t$. For CLD, $\mathbf{u}_t = \boldsymbol{\mu}_t(\mathbf{x}_0) + \boldsymbol{L}_t\boldsymbol{\epsilon}_{2d}$, where $\boldsymbol{\Sigma}_t = \boldsymbol{L}_t\boldsymbol{L}_t^\top$ is the Cholesky decomposition of $p_t(\mathbf{u}_t|\mathbf{x}_0)$'s covariance matrix, $\boldsymbol{\epsilon}_{2d} \sim \mathcal{N}(\boldsymbol{\epsilon}_{2d}; \mathbf{0}_{2d}, \boldsymbol{I}_{2d})$, and $\boldsymbol{\mu}_t(\mathbf{x}_0)$ is $p_t(\mathbf{u}_t|\mathbf{x}_0)$'s mean. Furthermore, $\nabla_{\mathbf{v}_t} \log p_t(\mathbf{u}_t|\mathbf{x}_0) = -\ell_t\boldsymbol{\epsilon}_{d:2d}$, where $\boldsymbol{\epsilon}_{d:2d}$ denotes those $d$ components of $\boldsymbol{\epsilon}_{2d}$ that actually affect $\nabla_{\mathbf{v}_t} \log p_t(\mathbf{u}_t|\mathbf{x}_0)$ (since we take velocity gradients only, not all are relevant).

$$\text{With} \quad \boldsymbol{\Sigma}_t = \underbrace{\begin{pmatrix} \Sigma_t^{xx} & \Sigma_t^{xv} \\ \Sigma_t^{xv} & \Sigma_t^{vv} \end{pmatrix}}_{\text{``per-dimension'' covariance matrix}} \otimes \boldsymbol{I}_d, \quad \text{we have} \quad \ell_t := \sqrt{\frac{\Sigma_t^{xx}}{\Sigma_t^{xx}\Sigma_t^{vv} - (\Sigma_t^{xv})^2}}.$$

**(ii)** Vahdat et al. (2021) showed that it can be beneficial to assume that the diffused marginal distribution is Normal at all times and parametrize the model with a Normal score and a residual "correction". For CLD, the score is indeed Normal at $t = 0$ (due to the independently initialized $\mathbf{x}$ and $\mathbf{v}$ at $t$=0). Similarly, the target score is close to Normal for large $t$, as we approach the equilibrium.

Based on **(i)** and **(ii)**, we parameterize $s_{\boldsymbol{\theta}}(\mathbf{u}_t, t) = -\ell_t\alpha_{\boldsymbol{\theta}}(\mathbf{u}_t, t)$ with $\alpha_{\boldsymbol{\theta}}(\mathbf{u}_t, t) = \ell_t^{-1}\mathbf{v}_t/\Sigma_t^{vv} + \alpha'_{\boldsymbol{\theta}}(\mathbf{u}_t, t)$, where $\Sigma_t^{vv}$ corresponds to the $v$-$v$ component of the "per-dimension" covariance matrix of the Normal distribution $p_t(\mathbf{u}_t|\mathbf{x}_0 = \mathbf{0}_d)$. In other words, we assumed $p_0(\mathbf{x}_0) = \delta(\mathbf{x})$ when defining the analytic term of the score model. Formally, $-\mathbf{v}/\Sigma_t^{vv}$ is the score of a Normal distribution with covariance $\hat{\Sigma}_t^{vv}\boldsymbol{I}_d$. Following Vahdat et al. (2021), we refer to this parameterization as *mixed score parameterization*. Alternative model parameterizations are possible, but we leave their exploration to future work. With this definition, the HSM training objective becomes (details in App. B.3):

$$\min_{\boldsymbol{\theta}} \mathbb{E}_{t\sim\mathcal{U}[0,T]}\mathbb{E}_{\mathbf{x}_0\sim p_0(\mathbf{x}_0)}\mathbb{E}_{\boldsymbol{\epsilon}_{2d}\sim\mathcal{N}(\boldsymbol{\epsilon}_{2d};\mathbf{0}_{2d},\boldsymbol{I}_{2d})} \left[\lambda(t)\ell_t^2\|\boldsymbol{\epsilon}_{d:2d} - \alpha_{\boldsymbol{\theta}}(\boldsymbol{\mu}_t(\mathbf{x}_0)+\boldsymbol{L}_t\boldsymbol{\epsilon}_{2d}, t)\|_2^2\right], \quad (9)$$

which corresponds to training the model to predict the noise only injected into the velocity during reparametrized sampling of $\mathbf{u}_t$, similar to noise prediction in Ho et al. (2020); Song et al. (2021c).

**Objective Weightings.** For $\lambda(t) = \Gamma\beta$, the objective corresponds to maximum likelihood learning (Song et al., 2021b) (see App. B.3). Analogously to prior work (Ho et al., 2020; Vahdat et al., 2021; Song et al., 2021b), an objective better suited for high quality image synthesis can be obtained by setting $\lambda(t) = \ell_t^{-2}$, which corresponds to "dropping the variance prefactor" $\ell_t^2$.

### 3.3 SAMPLING FROM CLD-BASED SGMS

To sample from the CLD-based SGM we can either directly simulate the reverse-time diffusion process (Eq. (2)) or, alternatively, solve the corresponding probability flow ODE (Song et al., 2021c;b) (see App. B.5). To simulate the SDE of the reverse-time diffusion process, previous works often relied on Euler-Maruyama (EM) (Kloeden & Platen, 1992) and related methods (Ho et al., 2020; Song et al., 2021c; Jolicoeur-Martineau et al., 2021a). We derive a new solver, tailored to CLD-based models. Here, we provide the high-level ideas and derivations (see App. D for details).

Our generative SDE can be written as (with $\bar{\mathbf{u}}_t = \mathbf{u}_{T-t}$, $\bar{\mathbf{x}}_t = \mathbf{x}_{T-t}$, $\bar{\mathbf{v}}_t = \mathbf{v}_{T-t}$):

$$\begin{pmatrix} d\bar{\mathbf{x}}_t \\ d\bar{\mathbf{v}}_t \end{pmatrix} = \underbrace{\begin{pmatrix} -M^{-1}\bar{\mathbf{v}}_t \\ \bar{\mathbf{x}}_t \end{pmatrix}\beta dt}_{A_H} + \underbrace{\begin{pmatrix} \mathbf{0}_d \\ -\Gamma M^{-1}\bar{\mathbf{v}}_t \end{pmatrix}\beta dt + \begin{pmatrix} \mathbf{0}_d \\ \sqrt{2\Gamma\beta}d\mathbf{w}_t \end{pmatrix}}_{A_O} + \underbrace{\begin{pmatrix} \mathbf{0}_d \\ 2\Gamma\left[\mathbf{s}(\bar{\mathbf{u}}_t, T-t) + M^{-1}\bar{\mathbf{v}}_t\right] \end{pmatrix}\beta dt}_{S}$$

It consists of a Hamiltonian component $A_H$, an Ornstein-Uhlenbeck process $A_O$, and the score model term $S$. We could use EM to integrate this SDE; however, standard Euler methods are not well-suited for Hamiltonian dynamics (Leimkuhler & Reich, 2005; Neal, 2011). Furthermore, if $S$ was 0, we could solve the SDE in closed form. This suggests the construction of a novel integrator.

We use the Fokker-Planck operator[2] formalism (Tuckerman, 2010; Leimkuhler & Matthews, 2013; 2015). Using a similar notation as Leimkuhler & Matthews (2013), the Fokker-Planck equation

---

[2]The *Fokker-Planck operator* is also known as *Kolmogorov operator*. If the underlying dynamics is fully Hamiltonian, it corresponds to the *Liouville operator* (Leimkuhler & Matthews, 2015; Tuckerman, 2010).

corresponding to the generative SDE is $\partial p_t(\bar{\mathbf{u}}_t)/\partial t = (\hat{\mathcal{L}}_A^* + \hat{\mathcal{L}}_S^*) p_t(\bar{\mathbf{u}}_t)$, where $\hat{\mathcal{L}}_A^*$ and $\hat{\mathcal{L}}_S^*$ are the non-commuting Fokker-Planck operators corresponding to the $A := A_H + A_O$ and $S$ terms, respectively. Expressions for $\hat{\mathcal{L}}_A^*$ and $\hat{\mathcal{L}}_S^*$ can be found in App. D. We can construct a formal, but intractable solution of the generative SDE as $\bar{\mathbf{u}}_t = e^{t(\hat{\mathcal{L}}_A^* + \hat{\mathcal{L}}_S^*)} \bar{\mathbf{u}}_0$, where the operator $e^{t(\hat{\mathcal{L}}_A^* + \hat{\mathcal{L}}_S^*)}$ (known as the *classical propagator* in statistical physics) propagates states $\bar{\mathbf{u}}_0$ for time $t$ according to the dynamics defined by the combined operators $\hat{\mathcal{L}}_A^* + \hat{\mathcal{L}}_S^*$. Although this operation is not analytically tractable, it can serve as starting point to derive a practical integrator. Using the symmetric Trotter theorem or Strang splitting formula as well as the Baker–Campbell–Hausdorff formula (Trotter, 1959; Strang, 1968; Tuckerman, 2010), it can be shown that:

$$e^{t(\hat{\mathcal{L}}_A^* + \hat{\mathcal{L}}_S^*)} = \lim_{N \to \infty} \left[ e^{\frac{\delta t}{2} \hat{\mathcal{L}}_A^*} e^{\delta t \hat{\mathcal{L}}_S^*} e^{\frac{\delta t}{2} \hat{\mathcal{L}}_A^*} \right]^N \approx \left[ e^{\frac{\delta t}{2} \hat{\mathcal{L}}_A^*} e^{\delta t \hat{\mathcal{L}}_S^*} e^{\frac{\delta t}{2} \hat{\mathcal{L}}_A^*} \right]^N + \mathcal{O}(N\delta t^3), \quad (10)$$

for large $N \in \mathbb{N}^+$ and time step $\delta t := t/N$. The expression suggests that instead of directly evaluating the intractable $e^{t(\hat{\mathcal{L}}_A^* + \hat{\mathcal{L}}_S^*)}$, we can discretize the dynamics over $t$ into $N$ pieces of step size $\delta t$, such that we only need to apply the *individual* $e^{\frac{\delta t}{2} \hat{\mathcal{L}}_A^*}$ and $e^{\delta t \hat{\mathcal{L}}_S^*}$ many times one after another for small steps $\delta t$. A finer discretization results in a smaller error (since $N = t/\delta t$, the error effectively scales as $\mathcal{O}(\delta t^2)$ for fixed $t$). Hence, this implies an integration method. Indeed, $e^{\frac{\delta t}{2} \hat{\mathcal{L}}_A^*} \bar{\mathbf{u}}_t$ is available in closed form, as mentioned before; however, $e^{\delta t \hat{\mathcal{L}}_S^*} \bar{\mathbf{u}}_t$ is not. Therefore, we approximate this latter component of the integrator via a standard Euler step. Thus, the integrator formally has an error of the same order as standard EM methods. Nevertheless, as long as the dynamics is not dominated by the $S$ component, our proposed integration scheme is expected to be more accurate than EM, since we split off the analytically tractable part and only use an Euler approximation for the $S$ term. Recall that the model only needs to learn the score of the conditional distribution $p_t(\mathbf{v}_t|\mathbf{x}_t)$, which is close to Normal for much of the diffusion, in which case the $S$ term will indeed be small. This suggests that the generative SDE dynamics are in fact dominated by $A_H$ and $A_O$ in practice. Note that only the propagator $e^{\delta t \hat{\mathcal{L}}_S^*}$ is computationally expensive, as it involves evaluating the neural network. We coin our novel SDE integrator for CLD-based SGMs *Symmetric Splitting CLD Sampler* (SSCS). A detailed derivation, analyses, and a formal algorithm are presented in App. D.

## 4 RELATED WORK

**Relations to Statistical Mechanics and Molecular Dynamics.** Learning a mapping between a simple, tractable and a complex distribution as in SGMs is inspired by annealed importance sampling (Neal, 2001) and the Jarzynski equality from non-equilibrium statistical mechanics (Jarzynski, 1997a;b; 2011; Bahri et al., 2020). However, after Sohl-Dickstein et al. (2015), little attention has been given to the origins of SGMs in statistical mechanics. Intuitively, in SGMs the diffusion process is initialized in a non-equilibrium state $\mathbf{u}_0$ and we would like to bring the system to equilibrium, i.e., the tractable prior distribution, *as quickly and as smoothly as possible* to enable efficient denoising. This "equilibration problem" is a much-studied problem in statistical mechanics, particularly in molecular dynamics, where a molecular system is often simulated in thermodynamic equilibrium. Algorithms to quickly and smoothly bring a system to and maintain at equilibrium are known as *thermostats*. In fact, CLD is inspired by the Langevin thermostat (Bussi & Parrinello, 2007). In molecular dynamics, advanced thermostats are required in particular for "multiscale" systems that show complex behaviors over multiple time- and length-scales. Similar challenges also arise when modeling complex data, such as natural images. Hence, the vast literature on thermostats (Andersen, 1980; Nosé, 1984; Hoover, 1985; Martyna et al., 1992; Hünenberger, 2005; Bussi et al., 2007; Ceriotti et al., 2009; 2010; Tuckerman, 2010) may be valuable for the development of future SGMs. Also the framework for developing SSCS is borrowed from statistical mechanics. The same techniques have been used to derive molecular dynamics algorithms (Tuckerman et al., 1992; Bussi & Parrinello, 2007; Ceriotti et al., 2010; Leimkuhler & Matthews, 2013; 2015; Kreis et al., 2017).

**Further Related Work.** Generative modeling by learning stochastic processes has a long history (Movellan, 2008; Lyu, 2009; Sohl-Dickstein et al., 2011; Bengio et al., 2014; Alain et al., 2016; Goyal et al., 2017; Bordes et al., 2017; Song & Ermon, 2019; Ho et al., 2020). We build on Song et al. (2021c), which introduced the SDE framework for modern SGMs. Nachmani et al. (2021) recently introduced non-Gaussian diffusion processes with different noise distributions. However, the noise is still injected directly into the data, and no improved sampling schemes or training objectives are introduced. Vahdat et al. (2021) proposed LSGM, which is complementary to CLD: we improve

the diffusion process itself, whereas LSGM "simplifies the data" by first embedding it into a smooth latent space. LSGM is an overall more complicated framework, as it is trained in two stages and relies on additional encoder and decoder networks. Recently, techniques to accelerate sampling from pre-trained SGMs have been proposed (San-Roman et al., 2021; Watson et al., 2021; Kong & Ping, 2021; Song et al., 2021a). Importantly, these methods usually do not permit straightforward log-likelihood estimation. Furthermore, they are originally not based on the continuous time framework, which we use, and have been developed primarily for discrete-step diffusion models.

A complementary work to CLD is "Gotta Go Fast" (GGF) (Jolicoeur-Martineau et al., 2021a), which introduces an adaptive SDE solver for SGMs, tuned towards image synthesis. GGF uses standard Euler-based methods under the hood (Kloeden & Platen, 1992; Roberts, 2012), in contrast to our SSCS that is derived from first principles. Furthermore, our SDE integrator for CLD does not make any data-specific assumptions and performs extremely well even without adaptive step sizes.

Some works study SGMs for maximum likelihood training (Song et al., 2021b; Kingma et al., 2021; Huang et al., 2021). Note that we did not focus on training our models towards high likelihood. Furthermore, Chen et al. (2020) and Huang et al. (2020) recently trained augmented Normalizing Flows, which have conceptual similarities with our velocity augmentation. Methods leveraging auxiliary variables similar to our velocities are also used in statistics—such as Hamiltonian Monte Carlo (Neal, 2011)—and have found applications, for instance, in Bayesian machine learning (Chen et al., 2014; Ding et al., 2014; Shang et al., 2015). As shown in Ma et al. (2019), our velocity is equivalent to momentum in gradient descent and related methods (Polyak, 1964; Kingma & Ba, 2015). Momentum accelerates optimization; the velocity in CLD accelerates mixing in the diffusion process. Lastly, our CLD method can be considered as a second-order Langevin algorithm, but even higher-order schemes are possible (Mou et al., 2021) and could potentially further improve SGMs.

# 5 EXPERIMENTS

**Architectures.** We focus on image synthesis and implement CLD-based SGMs using NCSN++ and DDPM++ (Song et al., 2021c) with 6 input channels (for velocity and data) instead of 3.

**Relevant Hyperparameters.** CLD's hyperparameters are chosen as $\beta=4$, $\Gamma=1$ (or equivalently $M^{-1}=4$) in all experiments. We set the variance scaling of the inital velocity distribution to $\gamma=0.04$ and use the proposed HSM objective with the weighting $\lambda(t)=\ell_t^{-2}$, which promotes image quality.

**Sampling.** We generate model samples via: **(i)** Probability flow using a Runge–Kutta 4(5) method; reverse-time generative SDE sampling using either **(ii)** EM or **(iii)** our SSCS. For methods without adaptive stepsize (EM and SSCS), we use evaluation times chosen according to a quadratic function, like previous work (Song et al., 2021a; Kong & Ping, 2021; Watson et al., 2021) (indicated by QS).

**Evaluation.** We measure image sample quality for CIFAR-10 via Fréchet inception distance (FID) with 50k samples (Heusel et al., 2017). We also evaluate an upper bound on the negative log-likelihood (NLL): $-\log p(\mathbf{x}_0) \leq -\mathbb{E}_{\mathbf{v}_0 \sim p(\mathbf{v}_0)} \log p_\varepsilon(\mathbf{x}_0, \mathbf{v}_0) - H$, where $H$ is the entropy of $p(\mathbf{v}_0)$ and $\log p_\varepsilon(\mathbf{x}_0, \mathbf{v}_0)$ is an unbiased estimate of $\log p(\mathbf{x}_0, \mathbf{v}_0)$ from the probability flow ODE (Grathwohl et al., 2019; Song et al., 2021c). As in Vahdat et al. (2021), the stochasticity of $\log p_\varepsilon(\mathbf{x}, \mathbf{v})$ prevents us from performing importance weighted NLL estimation over the velocity distribution (Burda et al., 2015). We also record the number of function—neural network—evaluations (NFEs) during synthesis when comparing sampling methods. All implementation details in App. B.5 and E.

## 5.1 IMAGE GENERATION

Following Song et al. (2021c), we focus on the widely used CIFAR-10 unconditional image generation benchmark. Our CLD-based SGM achieves an FID of 2.25 based on the probability flow ODE and an FID of 2.23 via simulating the generative SDE (Tab. 1). The only models marginally outperforming CLD are LSGM (Vahdat et al., 2021) and NSCN++/VESDE with 2,000 step predictor-corrector (PC) sampling (Song et al., 2021c). However, LSGM uses a model with $\approx 475M$ parameters to achieve its high performance, while we obtain our numbers with a model of $\approx 100M$ parameters. For a fairer comparison, we trained a smaller LSGM also with $\approx 100M$ parameters, which is reported as "LSGM-100M" in Tab. 1 (details in App. E.2.7). Our model has a significantly better FID score than "LSGM-100M". In contrast to NSCN++/VESDE, we achieve extremely strong results with much fewer NFEs (for example, see $n \in \{150, 275, 500\}$ in Tab. 3 and also Tab. 2)—the VESDE performs poorly in this regime. We conclude that when comparing models with similar

Table 1: Unconditional CIFAR-10 generative performance.

| Class | Model | NLL↓ | FID↓ |
|---|---|---|---|
| Score | CLD-SGM (Prob. Flow) *(ours)* | ≤3.31 | 2.25 |
| | CLD-SGM (SDE) *(ours)* | - | 2.23 |
| Score | DDPM++, VPSDE (Prob. Flow) (Song et al., 2021c) | 3.13 | 3.08 |
| | DDPM++, VPSDE (SDE) (Song et al., 2021c) | - | 2.41 |
| | DDPM++, sub-VP (Prob. Flow) (Song et al., 2021c) | 2.99 | 2.92 |
| | DDPM++, sub-VP (SDE) (Song et al., 2021c) | - | 2.41 |
| | NCSN++, VESDE (SDE) (Song et al., 2021c) | - | 2.20 |
| | LSGM (Vahdat et al., 2021) | ≤3.43 | 2.10 |
| | LSGM-100M (Vahdat et al., 2021) | ≤2.96 | 4.60 |
| | DDPM (Ho et al., 2020) | ≤3.75 | 3.17 |
| | NCSN (Song & Ermon, 2019) | - | 25.3 |
| | Adversarial DSM (Jolicoeur-Martineau et al., 2021b) | - | 6.10 |
| | Likelihood SDE (Song et al., 2021b) | 2.84 | 2.87 |
| | DDIM (100 steps) (Song et al., 2021a) | - | 4.16 |
| | FastDDPM (100 steps) (Kong & Ping, 2021) | - | 2.86 |
| | Improved DDPM (Nichol & Dhariwal, 2021) | 3.37 | 2.90 |
| | VDM (Kingma et al., 2021) | ≤2.49 | 7.41 (4.00) |
| | UDM (Kim et al., 2021) | 3.04 | 2.33 |
| | D3PM (Austin et al., 2021) | ≤3.44 | 7.34 |
| | Gotta Go Fast (Jolicoeur-Martineau et al., 2021a) | - | 2.44 |
| | DDPM Distillation (Luhman & Luhman, 2021) | - | 9.36 |
| GANs | SNGAN (Miyato et al., 2018) | - | 21.7 |
| | SNGAN+DGflow (Ansari et al., 2021) | - | 9.62 |
| | AutoGAN (Gong et al., 2019) | - | 12.4 |
| | TransGAN (Jiang et al., 2021) | - | 9.26 |
| | StyleGAN2 w/o ADA (Karras et al., 2020) | - | 8.32 |
| | StyleGAN2 w/ ADA (Karras et al., 2020) | - | 2.92 |
| | StyleGAN2 w/ Diffaug (Zhao et al., 2020) | - | 5.79 |
| Aut.-Reg., Flows, VAEs, EBMs | DistAug (Jun et al., 2020) | 2.53 | 42.90 |
| | PixelCNN (Oord et al., 2016) | 3.14 | 65.9 |
| | Glow (Kingma & Dhariwal, 2018) | 3.35 | 48.9 |
| | Residual Flow (Chen et al., 2019) | 3.28 | 46.37 |
| | NVAE (Vahdat & Kautz, 2020) | 2.91 | 23.5 |
| | NCP-VAE (Aneja et al., 2021) | - | 24.08 |
| | DC-VAE Parmar et al. (2021) | - | 17.90 |
| | IGEBM (Du & Mordatch, 2019) | - | 40.6 |
| | VAEBM (Xiao et al., 2021) | - | 12.2 |
| | Recovery EBM (Gao et al., 2021) | 3.18 | 9.58 |

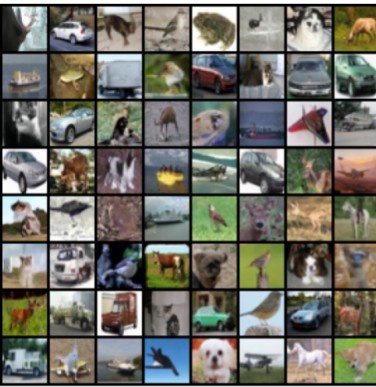

Figure 3: CIFAR-10 samples.

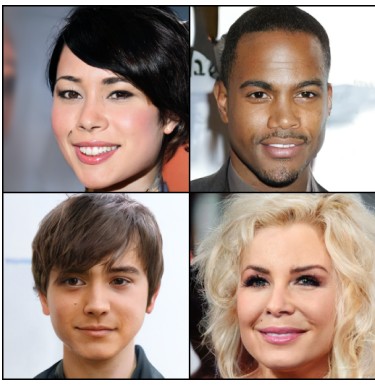

Figure 4: CelebA-HQ-256 samples.

network capacity and under NFE budgets ≤500, our CLD-SGM outperforms all published results in terms of FID. We attribute these positive results to our easier score matching task. Furthermore, our model reaches an NLL bound of 3.31, which is on par with recent works such as Nichol & Dhariwal (2021); Austin et al. (2021); Vahdat et al. (2021) and indicates that our model is not dropping modes. However, our bound is potentially quite loose (see discussion in App. B.5) and the true NLL might be significantly lower. We did not focus on training our models towards high likelihood.

To demonstrate that CLD is also suitable for high-resolution image synthesis, we additionally trained a CLD-SGM on CelebA-HQ-256, but without careful hyperparameter tuning due to limited compute resources. Model samples in Fig. 4 appear diverse and high-quality (additional samples in App. F).

## 5.2 SAMPLING SPEED AND SYNTHESIS QUALITY TRADE-OFFS

We analyze the sampling speed vs. synthesis quality trade-off for CLD-SGMs and study SSCS's performance under different NFE budgets (Tabs. 2 and 3). We compare to Song et al. (2021c) and use EM to solve the generative SDE for their VPSDE and PC (reverse-diffusion + Langevin sampler) for the VESDE model. We also compare to the GGF (Jolicoeur-Martineau et al., 2021a) solver for the generative SDE as well as probability flow ODE sampling with a higher-order adaptive step size solver. Further, we compare to LSGM (Vahdat et al., 2021) (using our LSGM-100M), which also uses probability flow sampling. With one exception (VESDE with 2,000 NFE) our CLD-SGM outperforms all baselines, both for adaptive and fixed-step size methods. More results in App. F.2.

Several observations stand out: **(i)** As expected (Sec. 3.3), for CLD, SSCS significantly outperforms EM under limited NFE budgets. When using a fine discretization of the SDE (high NFE), the two perform similarly, which is also expected, as the errors of both methods will become negligible. **(ii)** In the adaptive solver setting, using a simpler ODE solver, we even outperform GGF, which is tuned towards image synthesis. **(iii)** Our CLD-SGM also outperforms the LSGM-100M model in terms of FID. It is worth noting, however, that LSGM was designed primarily for faster synthesis, which it achieves by modeling a smooth distribution in latent space instead of the more complex data distribution directly. This suggests that it would be promising to combine LSGM with CLD and train a CLD-based LSGM, combining the strengths of the two approaches. It would also be interesting to develop a more advanced, adaptive SDE solver that leverages SSCS as the backbone

Table 2: *(right)* Performance using adaptive stepsize solvers (ODE is based on probability flow, GGF simulates generative SDE). †: taken from Jolicoeur-Martineau et al. (2021a). LSGM corresponds to the small LSGM-100M model for fair comparison (details in App. E.2.7). Error tolerances were chosen to obtain similar NFEs.

Table 3: *(bottom)* Performance using non-adaptive stepsize solvers (for PC, QS performed poorly). †: 2.23 FID is our evaluation, Song et al. (2021c) reports 2.20 FID. See Tab. 9 in App. F.2 for extended results.

| Model | Solver | NFEs↓ | FID↓ |
|---|---|---|---|
| CLD | ODE | 312 | **2.25** |
| VPSDE | GGF | 330 | 2.56† |
| VESDE | GGF | 488 | 2.99† |
| CLD | ODE | 147 | **2.71** |
| VPSDE | ODE | 141 | 2.76 |
| VPSDE | GGF | 151 | 2.73† |
| VESDE | ODE | 182 | 7.63 |
| VESDE | GGF | 170 | 10.15† |
| LSGM | ODE | 131 | 4.60 |

| Model | Sampler | FID at $n$ function evaluations ↓ | | | | | |
|---|---|---|---|---|---|---|---|
| | | $n$=50 | $n$=150 | $n$=275 | $n$=500 | $n$=1000 | $n$=2000 |
| CLD | EM-QS | 52.7 | 7.00 | 3.24 | 2.41 | **2.27** | 2.23 |
| CLD | SSCS-QS | **20.5** | **3.07** | **2.38** | **2.25** | 2.30 | 2.29 |
| VPSDE | EM-QS | 28.2 | 4.06 | 2.65 | 2.47 | 2.66 | 2.60 |
| VESDE | PC | 460 | 216 | 11.2 | 3.75 | 2.43 | **2.23**† |

and, for example, potentially test our method within a framework like GGF. Our current SSCS only allows for fixed step sizes—nevertheless, it achieves excellent performance.

Table 4: Mass hyperparameter.

| $M^{-1}$ | NLL↓ | FID↓ |
|---|---|---|
| 1 | ≤3.30 | 3.23 |
| 4 | ≤3.37 | 3.14 |
| 16 | ≤3.26 | 3.16 |

## 5.3 ABLATION STUDIES

We perform ablation studies to study CLD's new hyperparameters (run with a smaller version of our CLD-SGM used above; App. E for details).

**Mass Parameter:** Tab. 4 shows results for a CLD-SGM trained with different $M^{-1}$ (also recall that $M^{-1}$ and $\Gamma$ are tied together via $\Gamma^2 = 4M$; we are always in the critical-damping regime). Different mass values perform mostly similarly. Intuitively, training with smaller $M^{-1}$ means that noise flows from the velocity variables $\mathbf{v}_t$ into the data $\mathbf{x}_t$ more slowly, which necessitates a larger time rescaling $\beta$. We found that simply tying $M^{-1}$ and $\beta$ together via $\beta = 8\sqrt{M}$ works well and did not further fine-tune.

Table 5: Initial velocity distribution width.

| $\gamma$ | NLL↓ | FID↓ |
|---|---|---|
| 0.04 | ≤3.37 | 3.14 |
| 0.4 | ≤3.15 | 3.21 |
| 1 | ≤3.15 | 3.27 |

**Initial Velocity Distribution:** Tab. 5 shows results for a CLD-SGM trained with different initial velocity variance scalings $\gamma$. Varying $\gamma$ similarly has only a small effect, but small $\gamma$ seems slightly beneficial for FID, while the NLL bound suffers a bit. Due to our focus on synthesis quality as measued by FID, we opted for small $\gamma$. Intuitively, this means that the data at $t$=0 is "at rest", and noise flows from the velocity into the data variables only slowly.

**Mixed Score:** Similar to previous work (Vahdat et al., 2021), we find training with the mixed score (MS) parametrization (Sec. 3.2) beneficial. With MS, we achieve an FID of 3.14, without only 3.56.

**Hybrid Score Matching:** We also tried training with regular DSM, instead of HSM. However, training often became unstable. As discussed in Sec. 3.2, this is likely because when using standard DSM our CLD would suffer from unbounded scores close to $t$=0, similar to previous SDEs (Kim et al., 2021). Consequently, we consider our novel HSM a crucial element for training CLD-SGMs.

We conclude that CLD does not come with difficult-to-tune hyperparameters. We expect our chosen hyperparameters to immediately translate to new tasks and models. In fact, we used the same $M^{-1}$, $\gamma$, MS and HSM settings for CIFAR-10 and CelebA-HQ-256 experiments without fine-tuning.

## 6 CONCLUSIONS

We presented *critically-damped Langevin diffusion*, a novel diffusion process for training SGMs. CLD diffuses the data in a smoother, easier-to-denoise manner compared to previous SGMs, which results in smoother neural network-parametrized score functions, fast synthesis, and improved expressivity. Our experiments show that CLD outperforms previous SGMs on image synthesis for similar-capacity models and sampling compute budgets, while our novel SSCS is superior to EM in CLD-based SGMs. From a technical perspective, in addition to proposing CLD, we derive CLD's score matching objective termed as HSM, a variant of denoising score matching suited for CLD, and we derive a tailored SDE integrator for CLD. Inspired by methods used in statistical mechanics, our work provides new insights into SGMs and implies promising directions for future research.

We believe that CLD can potentially serve as the backbone diffusion process of next generation SGMs. Future work includes using CLD-based SGMs for generative modeling tasks beyond images, combining CLD with techniques for accelerated sampling from SGMs, adapting CLD-based SGMs towards maximum likelihood, and utilizing other thermostating methods from statistical mechanics.

## 7 ETHICS AND REPRODUCIBILITY

Our paper focuses on fundamental algorithmic advances to improve the generative modeling performance of SGMs. As such, the proposed CLD does not imply immediate ethical concerns. However, we validate CLD on image synthesis benchmarks. Generative modeling of images has promising applications, for example for digital content creation and artistic expression (Bailey, 2020), but can also be used for nefarious purposes (Vaccari & Chadwick, 2020; Mirsky & Lee, 2021; Nguyen et al., 2021). It is worth mentioning that compared to generative adversarial networks (Goodfellow et al., 2014), a very popular class of generative models, SGMs have the promise to model the data more faithfully, without dropping modes and introducing problematic biases. Generally, the ethical impact of our work depends on its application domain and the task at hand.

To aid reproducibility of the results and methods presented in our paper, we made source code to reproduce the main results of the paper publicly available, including detailed instructions; see our project page `https://nv-tlabs.github.io/CLD-SGM` and the code repository `https://github.com/nv-tlabs/CLD-SGM`. Furthermore, all training details and hyperparameters are already in detail described in the Appendix, in particular in App. E.

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

CONTENTS

## A    LANGEVIN DYNAMICS

Here, we discuss different aspects of Langevin dynamics. Recall the Langevin dynamics, Eq. (5), from the main paper:

$$
\begin{pmatrix} d\mathbf{x}_t \\ d\mathbf{v}_t \end{pmatrix} = \underbrace{\begin{pmatrix} M^{-1}\mathbf{v}_t \\ -\mathbf{x}_t \end{pmatrix} \beta dt}_{\text{Hamiltonian component} =: H} + \underbrace{\begin{pmatrix} \mathbf{0}_d \\ -\Gamma M^{-1}\mathbf{v}_t \end{pmatrix} \beta dt + \begin{pmatrix} 0 \\ \sqrt{2\Gamma\beta} \end{pmatrix} d\mathbf{w}_t}_{\text{Ornstein-Uhlenbeck process} =: O} . \tag{11}
$$

### A.1    DIFFERENT DAMPING RATIOS

As discussed in Sec. 3, Langevin dynamics can be run with different ratios between mass $M$ and squared friction $\Gamma^2$. To recap from the main paper:

**(i)** For $\Gamma^2 < 4M$ (*underdamped* Langevin dynamics), the Hamiltonian component dominates, which implies oscillatory dynamics of $\mathbf{x}_t$ and $\mathbf{v}_t$ that slow down convergence to equilibrium.

**(ii)** For $\Gamma^2 > 4M$ (*overdamped* Langevin dynamics), the $O$-term dominates which also slows down convergence, since the accelerating effect by the Hamiltonian component is suppressed due to the strong noise injection.

**(iii)** For $\Gamma^2 = 4M$ (*critical-damping*), an ideal balance is achieved and convergence to $p_{\text{EQ}}(\mathbf{u})$ occurs quickly in a smooth manner without oscillations.

In Fig. 5, we visualize diffusion trajectories according to Langevin dynamics run in the different damping regimes. We observe that underdamped Langevin dynamics show undesired oscillatory behavior, while overdamped Langevin dynamics perform very inefficiently, too. Critical-damping achieves a good balance between the two and mixes and converges quickly. In fact, it can be shown to be optimal in terms of convergence; see, for example, McCall (2010).

Consequently, we propose to set $\Gamma^2 = \Gamma^2_{\text{critical}} := 4M$ in CLD.

### A.2    VERY HIGH FRICTION LIMIT AND CONNECTIONS TO PREVIOUS SDES IN SGMS

Let us re-write the above Langevin dynamics and consider the more general case with time-dependent $\beta(t)$:

$$
d\mathbf{x}_t = M^{-1}\mathbf{v}_t\beta(t)dt, \tag{12}
$$

$$
d\mathbf{v}_t = - \underbrace{\mathbf{x}_t\beta(t)dt}_{\textbf{(ii): potential term}} - \underbrace{\Gamma M^{-1}\mathbf{v}_t\beta(t)dt}_{\textbf{(iii): friction term}} + \underbrace{\sqrt{2\Gamma\beta(t)}d\mathbf{w}_t}_{\textbf{(iv): noise term}} . \tag{13}
$$

To solve this SDE, let us assume a simple Euler-based integration scheme, with the update equation for a single step at time $t$ (this integration scheme would not be optimal, as discussed in Sec. 3.3., however, it would be accurate for sufficiently small time steps and we just need this to make the connection to previous works like the VPSDE):

$$
\mathbf{x}_{n+1} = \mathbf{x}_n + \beta(t)M^{-1}\mathbf{v}_{n+1}\delta t, \tag{14}
$$

$$
\mathbf{v}_{n+1} = \underbrace{\mathbf{v}_n}_{\textbf{(i): current step velocity}} - \underbrace{\beta(t)\mathbf{x}_n\delta t}_{\textbf{(ii): potential term}} - \underbrace{\beta(t)\Gamma M^{-1}\mathbf{v}_n\delta t}_{\textbf{(iii): friction term}} + \underbrace{\sqrt{2\beta(t)\Gamma}\mathcal{N}(\mathbf{0}_d, \delta t\mathbf{I}_d)}_{\textbf{(iv): noise term}}, \tag{15}
$$

Now, let us assume a friction coefficient $\Gamma = \Gamma_{\text{max}} := \frac{M}{\beta(t)\delta t}$. Since the time step $\delta t$ is usually very small, this correspond to a very high friction. In fact, it can be considered the maximum friction limit, at which the friction is so large that the current step velocity **(i)** is completely cancelled out by the friction term **(iii)**. We obtain:

$$
\mathbf{x}_{n+1} = \mathbf{x}_n + \beta(t)M^{-1}\mathbf{v}_{n+1}\delta t \tag{16}
$$

$$
\mathbf{v}_{n+1} = -\beta(t)\mathbf{x}_t\delta t + \sqrt{2\frac{M}{\delta t}}\mathcal{N}(\mathbf{0}_d, \delta t\mathbf{I}_d). \tag{17}
$$

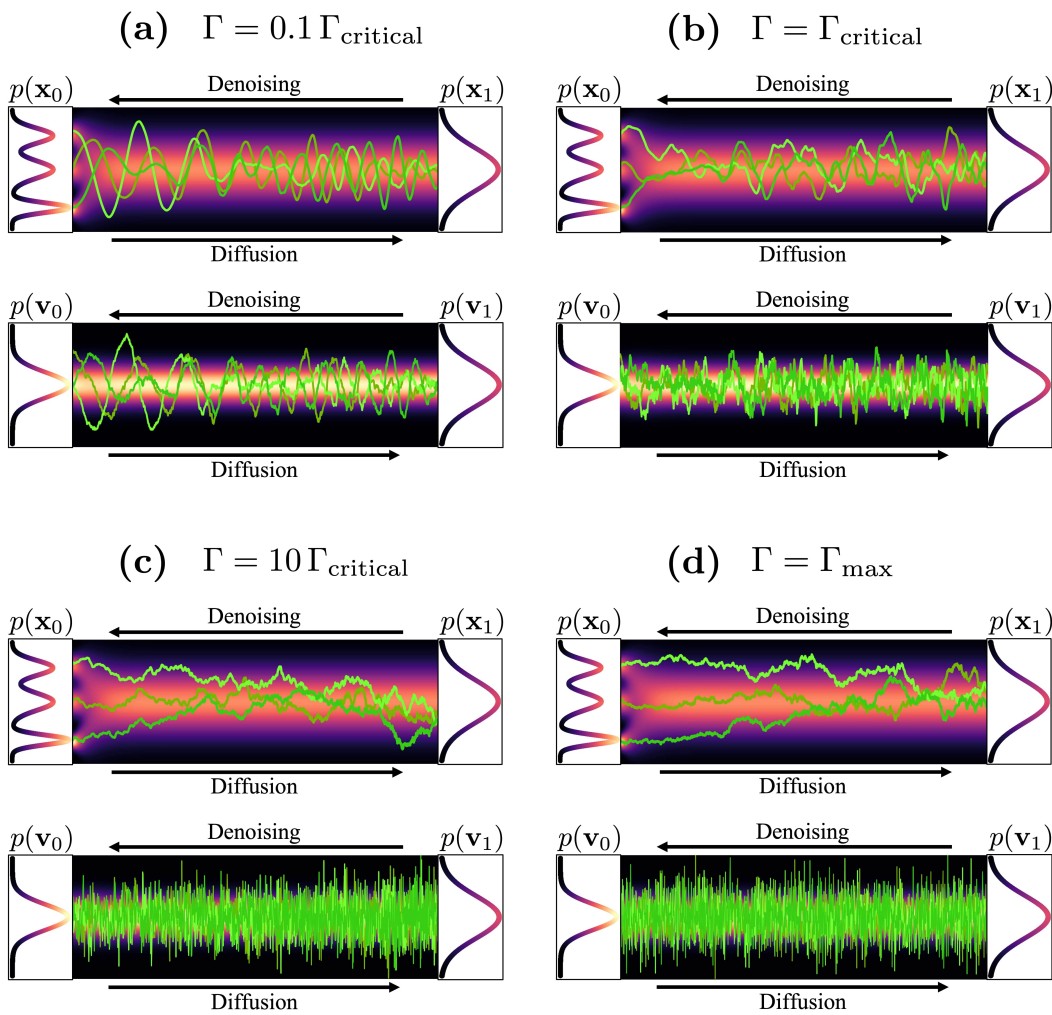

Figure 5: Langevin dynamics in different damping regimes. Each pair of visualizations corresponds to the (coupled) evolution of data $\mathbf{x}_t$ and velocities $\mathbf{v}_t$. We show the marginal **(red) probabilities** and the projections of the **(green) trajectories**. The probabilities always correspond to the same optimal setting $\Gamma = \Gamma_{\text{critical}}$ (recall that $\Gamma_{\text{critical}} = 2\sqrt{M}$ and $\Gamma_{\text{max}} = M/(\beta(t)\delta t)$; see Sec. A.2). The trajectories correspond to different Langevin trajectories run in the different regimes with indicated friction coefficients $\Gamma$. We see in **(b)**, that for *critical damping* the $\mathbf{x}_t$ trajectories quickly explore the space and converge according to the distribution indicated by the underlying probability. In the *under-damped regime* **(a)**, even though the trajectories mix quickly we observe undesired oscillatory behavior. For *over-damped Langevin dynamics*, **(c)** and **(d)**, the $\mathbf{x}_t$ trajectories mix and converge only very slowly. Note that the visualized diffusion uses different hyperparameters compared to the diffusion shown in Fig. 1 in the main text: Here, we have chosen a much larger $\beta$, such that also the slow overdamped Langevin dynamics trajectories shown here mix a little bit over the visualized diffusion time (while the probability distribution and the trajectories for critical damping converge almost instantly).

Now the velocity update, Eq. (17), does not depend on the current step velocity on the right-hand-side anymore. Hence, we can insert Eq. (17) directly into Eq. (16) and obtain:

$$\begin{aligned}
\mathbf{x}_{n+1} &= \mathbf{x}_n - \beta(t)^2 M^{-1}\mathbf{x}_n\delta t^2 + \sqrt{2\beta(t)^2\delta t M^{-1}}\mathcal{N}(\mathbf{0}_d, \delta t \boldsymbol{I}_d) \\
&= \mathbf{x}_n - \beta(t)^2 M^{-1}\mathbf{x}_n\delta t^2 + \sqrt{2\beta(t)^2\delta t^2 M^{-1}}\mathcal{N}(\mathbf{0}_d, \boldsymbol{I}_d).
\end{aligned} \tag{18}$$

Re-defining $\delta t' := \delta t^2$ and $\beta'(t) := \beta(t)^2$, we obtain

$$\mathbf{x}_{n+1} = \mathbf{x}_n - \beta'(t)M^{-1}\mathbf{x}_n\delta t' + \sqrt{2\beta'(t)\delta t' M^{-1}}\mathcal{N}(\mathbf{0}_d, \boldsymbol{I}_d), \tag{19}$$

which corresponds to the high-friction overdamped Langevin dynamics that are frequently run, for example, to train energy-based generative models (Du & Mordatch, 2019; Xiao et al., 2021). Let's further absorb the mass $M^{-1}$ and the time step $\delta t'$ into the time rescaling, defining $\hat{\beta}(t) :=$

$2\beta'(t)M^{-1}\delta t'$. We obtain:

$$
\begin{aligned}
\mathbf{x}_{n+1} &= \mathbf{x}_n - \frac{1}{2}\hat{\beta}(t)\mathbf{x}_n + \sqrt{\hat{\beta}(t)}\mathcal{N}(\mathbf{0}_d, \boldsymbol{I}_d) \\
&= (1 - \frac{1}{2}\hat{\beta}(t))\mathbf{x}_n + \sqrt{\hat{\beta}(t)}\mathcal{N}(\mathbf{0}_d, \boldsymbol{I}_d) \\
&\approx \sqrt{1 - \hat{\beta}(t)}\mathbf{x}_n + \sqrt{\hat{\beta}(t)}\mathcal{N}(\mathbf{0}_d, \boldsymbol{I}_d),
\end{aligned}
\tag{20}
$$

where the last approximation is true for sufficiently small $\hat{\beta}(t)$. However, this expression corresponds to

$$
\mathbf{x}_{n+1} \sim \mathcal{N}(\mathbf{x}_{n+1}; \sqrt{1 - \hat{\beta}(t)}\mathbf{x}_n, \hat{\beta}(t)\boldsymbol{I}_d)
\tag{21}
$$

which is exactly the transition kernel of the VPSDE's Markov chain (Ho et al., 2020; Song et al., 2021c). We see that the VPSDE corresponds to the high-friction limit of a more general Langevin dynamics-based diffusion process of the form of Eq. (11).

If we assume a diffusion as above but with the potential term **(ii)** set to 0, we can similarly derive the VESDE Song et al. (2021c) as a high-friction limit of the corresponding diffusion. Generally, all previously used diffusions that inject noise directly into the data variables correspond to such high-friction diffusions.

In conclusion, we see that previous high-friction diffusions require an excessive amount of noise to be injected to bring the dynamics to the prior, which intuitively makes denoising harder. For our CLD in the critical damping regime we can run the diffusion for a much shorter time or, equivalently, can inject less noise to converge to the equilibrium, i.e., the prior.

## B  CRITICALLY-DAMPED LANGEVIN DIFFUSION

Here, we present further details about our proposed critically-damped Langevin diffusion (CLD). We provide the derivations and formulas that were not presented in the main paper in the interest of brevity.

### B.1  PERTURBATION KERNEL

To recap from the main text, in this work we propose to augment the data $\mathbf{x}_t \in \mathbb{R}^d$ with auxiliary *velocity* variables $\mathbf{v}_t \in \mathbb{R}^d$. We then run the following diffusion process in the joint $\mathbf{x}_t$-$\mathbf{v}_t$-space

$$
d\mathbf{u}_t \coloneqq \begin{pmatrix} d\mathbf{x}_t \\ d\mathbf{v}_t \end{pmatrix} = \boldsymbol{f}(\mathbf{u}_t, t)dt + \boldsymbol{G}(\mathbf{u}_t, t)d\mathbf{w}_t
\tag{22}
$$

$$
\boldsymbol{f}(\mathbf{u}_t, t) = (f(t) \otimes \boldsymbol{I}_d)\mathbf{u}_t, \quad f(t) \coloneqq \begin{pmatrix} 0 & \beta(t)M^{-1} \\ -\beta(t) & -\beta(t)\Gamma M^{-1} \end{pmatrix},
\tag{23}
$$

$$
\boldsymbol{G}(\mathbf{u}_t, t) = G(t) \otimes \boldsymbol{I}_d, \quad G(t) \coloneqq \begin{pmatrix} 0 & 0 \\ 0 & \sqrt{2\Gamma\beta(t)} \end{pmatrix},
\tag{24}
$$

where $\mathbf{w}_t$ is a standard Wiener process in $\mathbb{R}^{2d}$ and $\beta\colon [0, T] \to \mathbb{R}_0^+$ is a time rescaling.[3] In particular, we consider the *critically-damped Langevin diffusion* which can be obtained by setting $M = \Gamma^2/4$, resulting in the following drift kernel

$$
\boldsymbol{f}_{\mathrm{CLD}}(\mathbf{u}_t, t) = (f_{\mathrm{CLD}}(t) \otimes \boldsymbol{I}_d)\mathbf{u}_t, \quad f_{\mathrm{CLD}}(t) \coloneqq \begin{pmatrix} 0 & 4\beta(t)\Gamma^{-2} \\ -\beta(t) & -4\beta(t)\Gamma^{-1} \end{pmatrix}.
\tag{25}
$$

Since we only consider the critically-damped case in this work, we redefine $\boldsymbol{f} \coloneqq \boldsymbol{f}_{\mathrm{CLD}}$ and $f \coloneqq f_{\mathrm{CLD}}$ for simplicity. Since our drift $\boldsymbol{f}$ and diffusion $\boldsymbol{G}$ coefficients are affine, $\mathbf{u}_t$ is Normally distributed for all $t \in [0, T]$ if $\mathbf{u}_0$ is Normally distributed at $t = 0$ (Särkkä & Solin, 2019). In particular, given that $\mathbf{u}_0 \sim \mathcal{N}(\mathbf{u}_0; \boldsymbol{\mu}_0, \boldsymbol{\Sigma}_0 = \Sigma_0 \otimes \boldsymbol{I}_d)$, where $\Sigma_0 = \mathrm{diag}(\Sigma_0^{xx}, \Sigma_0^{vv})$ is a positive

---

[3]For our experiments, we only used constant $\beta$; however, for generality, we present all derivations for time-dependent $\beta(t)$.

semi-definite diagonal 2-by-2 matrix (we restrict our derivation to diagonal covariance matrices at $t = 0$ for simplicity, since in our situation velocity and data are generally independent at $t = 0$), we derive expressions for $\boldsymbol{\mu}_t$ and $\boldsymbol{\Sigma}_t$, the mean and the covariance matrix of $\mathbf{u}_t$, respectively.

Following Särkkä & Solin (2019) (Section 6.1), the mean and covariance matrix of $\mathbf{u}_t$ obey the following respective ordinary differential equations (ODEs)

$$\frac{d\boldsymbol{\mu}_t}{dt} = (f(t) \otimes \boldsymbol{I}_d)\boldsymbol{\mu}_t, \tag{26}$$

$$\frac{d\boldsymbol{\Sigma}_t}{dt} = (f(t) \otimes \boldsymbol{I}_d)\boldsymbol{\Sigma}_t + [(f(t) \otimes \boldsymbol{I}_d)\boldsymbol{\Sigma}_t]^\top + \left(G(t)G(t)^\top\right) \otimes \boldsymbol{I}_d. \tag{27}$$

Notating $\boldsymbol{\mu}_0 = (\mathbf{x}_0, \mathbf{v}_0)^\top$, the solutions to the above ODEs are

$$\boldsymbol{\mu}_t = \begin{pmatrix} 2\mathcal{B}(t)\Gamma^{-1}\mathbf{x}_0 + 4\mathcal{B}(t)\Gamma^{-2}\mathbf{v}_0 + \mathbf{x}_0 \\ -\mathcal{B}(t)\mathbf{x}_0 - 2\mathcal{B}(t)\Gamma^{-1}\mathbf{v}_0 + \mathbf{v}_0 \end{pmatrix} e^{-2\mathcal{B}(t)\Gamma^{-1}}, \tag{28}$$

and

$$\boldsymbol{\Sigma}_t = \Sigma_t \otimes \boldsymbol{I}_d, \tag{29}$$

$$\Sigma_t = \begin{pmatrix} \Sigma_t^{xx} & \Sigma_t^{xv} \\ \Sigma_t^{xv} & \Sigma_t^{vv} \end{pmatrix} e^{-4\mathcal{B}(t)\Gamma^{-1}}, \tag{30}$$

$$\Sigma_t^{xx} = \Sigma_0^{xx} + e^{4\mathcal{B}(t)\Gamma^{-1}} - 1 + 4\mathcal{B}(t)\Gamma^{-1}\left(\Sigma_0^{xx} - 1\right) + 4\mathcal{B}^2(t)\Gamma^{-2}\left(\Sigma_0^{xx} - 2\right) + 16\mathcal{B}(t)^2\Gamma^{-4}\Sigma_0^{vv}, \tag{31}$$

$$\Sigma_t^{xv} = -\mathcal{B}(t)\Sigma_0^{xx} + 4\mathcal{B}(t)\Gamma^{-2}\Sigma_0^{vv} - 2\mathcal{B}^2(t)\Gamma^{-1}\left(\Sigma_0^{xx} - 2\right) - 8\mathcal{B}^2(t)\Gamma^{-3}\Sigma_0^{vv}, \tag{32}$$

$$\Sigma_t^{vv} = \frac{\Gamma^2}{4}\left(e^{4\mathcal{B}(t)\Gamma^{-1}} - 1\right) + \mathcal{B}(t)\Gamma + \Sigma_0^{vv}\left(1 + 4\mathcal{B}(t)^2\Gamma^{-2} - 4\mathcal{B}(t)\Gamma^{-1}\right) + \mathcal{B}(t)^2\left(\Sigma_0^{xx} - 2\right), \tag{33}$$

where $\mathcal{B}(t) = \int_0^t \beta(\hat{t})\, d\hat{t}$. For constant $\beta(t) = \beta$ (as is used in all our experiments), we simply have $\mathcal{B}(t) = t\beta$. The correctness of the proposed mean and covariance matrix can be verified by simply plugging them back into their respective ODEs; see App. G.1.

With the above derivations, we can find analytical expressions for the perturbation kernel $p(\mathbf{u}_t|\cdot)$. For example, when conditioning on initial data and velocity samples $\mathbf{x}_0$ and $\mathbf{v}_0$ (as in denoising score matching (DSM)), the mean and covariance matrix of the perturbation kernel $p(\mathbf{u}_t|\mathbf{u}_0)$ can be obtained by setting $\boldsymbol{\mu}_0 = (\mathbf{x}_0, \mathbf{v}_0)^\top$, $\Sigma_0^{xx} = 0$, and $\Sigma_0^{vv} = 0$.

In our experiments, the initial velocity distribution is set to $\mathcal{N}(\mathbf{0}_d, \gamma M \boldsymbol{I}_d)$. Conditioning only on initial data samples $\mathbf{x}_0$ and marginalizing over the full initial velocity distribution (as in our hybrid score matching (HSM), see Sec. C), the mean and covariance matrix of the perturbation kernel $p(\mathbf{u}_t|\mathbf{x}_0)$ can be obtained by setting $\boldsymbol{\mu}_0 = (\mathbf{x}_0, \mathbf{0}_d)^\top$, $\Sigma_0^{xx} = 0$, and $\Sigma_0^{vv} = \gamma M$.

## B.2 Convergence and Equilibrium

Our CLD-based training of SGMs—as well as denoising diffusion models more generally—relies on the fact that the diffusion converges towards an analytically tractable equilibrium distribution for sufficiently large $t$. In fact, from the above equations we can easily see that,

$$\lim_{t \to \infty} \Sigma_t^{xx} = 1, \tag{34}$$

$$\lim_{t \to \infty} \Sigma_t^{xv} = 0, \tag{35}$$

$$\lim_{t \to \infty} \Sigma_t^{vv} = \frac{\Gamma^2}{4} = M, \tag{36}$$

$$\lim_{t \to \infty} \boldsymbol{\mu}_t = \mathbf{0}_{2d}, \tag{37}$$

which establishes $p_{\mathrm{EQ}}(\mathbf{u}) = \mathcal{N}(\mathbf{x}; \mathbf{0}_d, \boldsymbol{I}_d)\,\mathcal{N}(\mathbf{v}; \mathbf{0}_d, M\boldsymbol{I}_d)$.

Notice that our CLD is an instantiation of the more general Langevin dynamics defined by

$$\begin{pmatrix} d\mathbf{x}_t \\ d\mathbf{v}_t \end{pmatrix} = \begin{pmatrix} M^{-1}\mathbf{v}_t \\ \nabla_{\mathbf{x}_t} \log p_{\mathrm{pot}}(\mathbf{x}_t) \end{pmatrix} \beta dt + \begin{pmatrix} \mathbf{0}_d \\ -\Gamma M^{-1}\mathbf{v}_t \end{pmatrix} \beta dt + \begin{pmatrix} 0 \\ \sqrt{2\Gamma\beta} \end{pmatrix} d\mathbf{w}_t. \tag{38}$$

which has the equilibrium distribution $\hat{p}_{\text{EQ}}(\mathbf{u}) = p_{\text{pot}}(\mathbf{x})\mathcal{N}(\mathbf{v};\mathbf{0}_d, M\boldsymbol{I}_d)$ (Leimkuhler & Matthews, 2015; Tuckerman, 2010). However, the perturbation kernel of this Langevin dynamics is not available analytically anymore for arbitrary $p_{\text{pot}}(\mathbf{x})$. In our case, however, we have the analytically tractable $p_{\text{pot}}(\mathbf{x}) = \mathcal{N}(\mathbf{x};\mathbf{0}_d, \boldsymbol{I}_d)$. Note that this corresponds to the classical "harmonic oscillator" problem from physics.

## B.3 CLD Objective

To derive the objective for training CLD-based SGMs, we start with a derivation that targets maximum likelihood training in a similar fashion to Song et al. (2021b). Let $p_0$ and $q_0$ be two densities, then

$$
\begin{aligned}
D_{\text{KL}}(p_0 \parallel q_0) &= D_{\text{KL}}(p_0 \parallel q_0) - D_{\text{KL}}(p_T \parallel q_T) + D_{\text{KL}}(p_T \parallel q_T) \\
&= -\int_0^T \frac{\partial D_{\text{KL}}(p_t \parallel q_t)}{\partial t} dt + D_{\text{KL}}(p_T \parallel q_T),
\end{aligned}
\tag{39}
$$

where $p_t$ and $q_t$ are the marginal densities of $p_0$ and $q_0$, respectively, diffused by our critically-damped Langevin diffusion. As has been shown in Song et al. (2021b), Eq. (39) can be written as a mixture (over $t$) of score matching losses. To this end, let us consider the Fokker–Planck equation associated with the critically-damped Langevin diffusion:

$$
\begin{aligned}
\frac{\partial p_t(\mathbf{u}_t)}{\partial t} &= \nabla_{\mathbf{u}_t} \cdot \left[ \tfrac{1}{2}\left(G(t)G(t)^\top \otimes \boldsymbol{I}_d\right)\nabla_{\mathbf{u}_t} p_t(\mathbf{u}_t) - p_t(\mathbf{u}_t)(f(t)\otimes\boldsymbol{I}_d)\mathbf{u}_t \right] \\
&= \nabla_{\mathbf{u}_t} \cdot \left[ \boldsymbol{h}_p(\mathbf{u}_t, t)p_t(\mathbf{u}_t) \right], \quad \boldsymbol{h}_p(\mathbf{u}_t, t) \coloneqq \tfrac{1}{2}\left(G(t)G(t)^\top \otimes \boldsymbol{I}_d\right)\nabla_{\mathbf{u}_t}\log p_t(\mathbf{u}_t) - (f(t)\otimes\boldsymbol{I}_d)\mathbf{u}_t.
\end{aligned}
\tag{40}
$$

Similarly, we have $\frac{\partial q_t(\mathbf{u}_t)}{\partial t} = \nabla_{\mathbf{u}_t} \cdot [\boldsymbol{h}_q(\mathbf{u}_t, t)q_t(\mathbf{u}_t)]$. Assuming $\log p_t(\mathbf{u}_t)$ and $\log q_t(\mathbf{u}_t)$ are smooth functions with at most polynomial growth at infinity, we have

$$
\lim_{\mathbf{u}_t \to \infty} \boldsymbol{h}_p(\mathbf{u}_t, t)p_t(\mathbf{u}_t) = \lim_{\mathbf{u}_t \to \infty} \boldsymbol{h}_q(\mathbf{u}_t, t)q_t(\mathbf{u}_t) = 0.
\tag{41}
$$

Using the above fact, we can compute the time-derivative of the Kullback–Leibler divergence between $p_t$ and $q_t$ as

$$
\begin{aligned}
\frac{\partial D_{\text{KL}}(p_t \parallel q_t)}{\partial t} &= \frac{\partial}{\partial t}\int p_t(\mathbf{u}_t)\log\frac{p_t(\mathbf{u}_t)}{q_t(\mathbf{u}_t)}d\mathbf{u}_t \\
&= -\int p_t(\mathbf{u}_t)\left[\boldsymbol{h}_p(\mathbf{u}_t,t) - \boldsymbol{h}_q(\mathbf{u}_t,t)\right]^\top \left[\nabla_{\mathbf{u}_t}\log p_t(\mathbf{u}_t) - \nabla_{\mathbf{u}_t}\log q_t(\mathbf{u}_t)\right]d\mathbf{u}_t \\
&= -\frac{1}{2}\int p_t(\mathbf{u}_t)\left[\nabla_{\mathbf{u}_t}\log p_t(\mathbf{u}_t) - \nabla_{\mathbf{u}_t}\log q_t(\mathbf{u}_t)\right]^\top \left(G(t)G(t)^\top \otimes \boldsymbol{I}_d\right)\left[\nabla_{\mathbf{u}_t}\log p_t(\mathbf{u}_t)\right. \\
&\quad \left. - \nabla_{\mathbf{u}_t}\log q_t(\mathbf{u}_t)\right]d\mathbf{u}_t \\
&= -\beta(t)\Gamma\int p_t(\mathbf{u}_t)\|\nabla_{\mathbf{v}_t}\log p_t(\mathbf{u}_t) - \nabla_{\mathbf{v}_t}\log q_t(\mathbf{u}_t)\|_2^2 d\mathbf{u}_t.
\end{aligned}
\tag{42}
$$

Notice that due to the form of $G(t)$, we now have only gradients with respect to the velocity component $\mathbf{v}_t$. Combining the above with Eq. (39), we have

$$
\begin{aligned}
D_{\text{KL}}(p_0 \parallel q_0) &= \mathbb{E}_{t\sim\mathcal{U}[0,T], \mathbf{u}_t\sim p_t(\mathbf{u})}\left[\Gamma\beta(t)\|\nabla_{\mathbf{v}_t}\log p_t(\mathbf{u}_t) - \nabla_{\mathbf{v}_t}\log q_t(\mathbf{u}_t)\|_2^2\right] + D_{\text{KL}}(p_T \parallel q_T) \\
&\approx \mathbb{E}_{t\sim\mathcal{U}[0,T], \mathbf{u}_t\sim p_t(\mathbf{u})}\left[\Gamma\beta(t)\|\nabla_{\mathbf{v}_t}\log p_t(\mathbf{u}_t) - \nabla_{\mathbf{v}_t}\log q_t(\mathbf{u}_t)\|_2^2\right],
\end{aligned}
\tag{43}
$$

Note that the approximation holds if $p_T$ is sufficiently "close" to $q_T$. We obtain a more general objective function by replacing $\Gamma\beta(t)$ with an arbitrary function $\lambda(t)$, i.e.,

$$
\mathbb{E}_{t\sim\mathcal{U}[0,T], \mathbf{u}_t\sim p_t(\mathbf{u})}\left[\lambda(t)\|\nabla_{\mathbf{v}_t}\log p_t(\mathbf{u}_t) - \nabla_{\mathbf{v}_t}\log q_t(\mathbf{u}_t)\|_2^2\right]
\tag{44}
$$

As shown in App. C, the above can be rewritten, up to irrelevant constant terms, as either of the following two objectives:

$$\text{HSM}(\lambda(t)) := \mathbb{E}_{t \sim \mathcal{U}[0,T], \mathbf{x}_0 \sim p_0(\mathbf{x}_0), \mathbf{u}_t \sim p_t(\mathbf{u}_t | \mathbf{x}_0)} \left[ \lambda(t) \| \nabla_{\mathbf{v}_t} \log p_t(\mathbf{u}_t \mid \mathbf{x}_0) - \nabla_{\mathbf{v}_t} \log q_t(\mathbf{u}_t) \|_2^2 \right],$$
(45)

$$\text{DSM}(\lambda(t)) := \mathbb{E}_{t \sim \mathcal{U}[0,T], \mathbf{u}_0 \sim p_0(\mathbf{u}_0), \mathbf{u}_t \sim p_t(\mathbf{u}_t | \mathbf{u}_0)} \left[ \lambda(t) \| \nabla_{\mathbf{v}_t} \log p_t(\mathbf{u}_t \mid \mathbf{u}_0) - \nabla_{\mathbf{v}_t} \log q_t(\mathbf{u}_t) \|_2^2 \right].$$
(46)

For both HSM and DSM, we have shown in App. B.1 that the perturbation kernels $p_t(\mathbf{u}_t \mid \mathbf{x}_0)$ and $p_t(\mathbf{u}_t \mid \mathbf{u}_0)$ are Normal distributions with the following structure of the covariance matrix:

$$\boldsymbol{\Sigma}_t = \Sigma_t \otimes \boldsymbol{I}_d, \quad \Sigma_t = \begin{pmatrix} \Sigma_t^{xx} & \Sigma_t^{xv} \\ \Sigma_t^{xv} & \Sigma_t^{vv} \end{pmatrix}.$$
(47)

We can use this fact to compute the gradient $\nabla_{\mathbf{u}_t} \log p_t(\mathbf{u}_t \mid \cdot)$

$$\begin{aligned} \nabla_{\mathbf{u}_t} \log p_t(\mathbf{u}_t \mid \cdot) &= -\nabla_{\mathbf{u}_t} \tfrac{1}{2} (\mathbf{u}_t - \boldsymbol{\mu}_t) \boldsymbol{\Sigma}_t^{-1} (\mathbf{u}_t - \boldsymbol{\mu}_t) \\ &= -\boldsymbol{\Sigma}_t^{-1} (\mathbf{u}_t - \boldsymbol{\mu}_t) \\ &= -\boldsymbol{L}_t^{-\top} \boldsymbol{L}_t^{-1} (\mathbf{u}_t - \boldsymbol{\mu}_t) \\ &= -\boldsymbol{L}_t^{-\top} \boldsymbol{\epsilon}_{2d}, \end{aligned}$$
(48)

where $\boldsymbol{\epsilon}_{2d} \sim \mathcal{N}(\mathbf{0}, \boldsymbol{I}_{2d})$ and $\boldsymbol{\Sigma}_t = \boldsymbol{L}_t \boldsymbol{L}_t^\top$ is the Cholesky factorization of the covariance matrix $\boldsymbol{\Sigma}_t$. Note that the structure of $\boldsymbol{\Sigma}_t$ implies that $\boldsymbol{L}_t = L_t \otimes \boldsymbol{I}_d$, where $L_t L_t^\top$ is the Cholesky factorization of $\Sigma_t$, i.e,

$$L_t = \begin{pmatrix} L_t^{xx} & L_t^{xv} \\ L_t^{xv} & L_t^{vv} \end{pmatrix} = \begin{pmatrix} \sqrt{\Sigma_t^{xx}} & 0 \\ \frac{\Sigma_t^{xv}}{\sqrt{\Sigma_t^{xx}}} & \sqrt{\frac{\Sigma_t^{xx}\Sigma_t^{vv} - (\Sigma_t^{xv})^2}{\Sigma_t^{xx}}} \end{pmatrix}.$$
(49)

Furthermore, we have

$$\begin{aligned} \boldsymbol{L}_t^{-\top} &= L_t^{-\top} \otimes \boldsymbol{I}_d \\ &= \begin{pmatrix} \sqrt{\Sigma_t^{xx}} & \frac{\Sigma_t^{xv}}{\sqrt{\Sigma_t^{xx}}} \\ 0 & \sqrt{\frac{\Sigma_t^{xx}\Sigma_t^{vv} - (\Sigma_t^{xv})^2}{\Sigma_t^{xx}}} \end{pmatrix}^{-1} \otimes \boldsymbol{I}_d \\ &= \begin{pmatrix} \frac{1}{\sqrt{\Sigma_t^{xx}}} & \frac{-\Sigma_t^{xz}}{\sqrt{\Sigma_t^{xx}}\sqrt{\Sigma_t^{xx}\Sigma_t^{zz} - (\Sigma_t^{xv})^2}} \\ 0 & \sqrt{\frac{\Sigma_t^{xx}}{\Sigma_t^{xx}\Sigma_t^{vv} - (\Sigma_t^{xv})^2}} \end{pmatrix} \otimes \boldsymbol{I}_d. \end{aligned}$$
(50)

Using the above, we can compute

$$\begin{aligned} \nabla_{\mathbf{v}_t} \log p_t(\mathbf{u}_t \mid \cdot) &= [\nabla_{\mathbf{u}_t} \log p_t(\mathbf{u}_t \mid \cdot)]_{d:2d} \\ &= \left[ -\boldsymbol{L}_t^{-\top} \boldsymbol{\epsilon}_{2d} \right]_{d:2d} \\ &= -\ell_t \boldsymbol{\epsilon}_{d:2d}, \end{aligned}$$
(51)

where

$$\ell_t := \sqrt{\frac{\Sigma_t^{xx}}{\Sigma_t^{xx}\Sigma_t^{vv} - (\Sigma_t^{xv})^2}},$$
(52)

and $\boldsymbol{\epsilon}_{d:2d}$ denotes those (latter) $d$ components of $\boldsymbol{\epsilon}_{2d}$ that actually affect $\nabla_{\mathbf{v}_t} \log p_t(\mathbf{u}_t | \cdot)$.

Note that $\ell_t$ depends on the conditioning in the perturbation kernel, and therefore $\ell_t$ is different for DSM, which is based on $p(\mathbf{u}_t \mid \mathbf{u}_0)$, and HSM, which is based on $p(\mathbf{u}_t \mid \mathbf{x}_0)$. Therefore, we will henceforth refer to $\ell_t^{\text{HSM}}$ and $\ell_t^{\text{DSM}}$ if distinction of the two cases is necessary (otherwise we will simply refer to $\ell_t$ for both).

As discussed in Section 3.2, we model $\nabla_{\mathbf{v}_t} \log q_t(\mathbf{u}_t)$ as $s_{\boldsymbol{\theta}}(\mathbf{u}_t, t) = -\ell_t \alpha_{\boldsymbol{\theta}}(\mathbf{u}_t, t)$. Plugging everything back into our objective functions, Eq. (45) and Eq. (46), we obtain

$$\text{HSM}(\lambda(t)) = \mathbb{E}_{t \sim \mathcal{U}[0,T], \mathbf{x}_0 \sim p_0(\mathbf{x}_0), \mathbf{u}_t \sim p_t(\mathbf{u}_t | \mathbf{x}_0)} \left[ \lambda(t) \left( \ell_t^{\text{HSM}} \right)^2 \| \boldsymbol{\epsilon}_{d:2d} - \alpha_{\boldsymbol{\theta}}(\mathbf{u}_t, t) \|_2^2 \right], \quad (53)$$

$$\text{DSM}(\lambda(t)) = \mathbb{E}_{t \sim \mathcal{U}[0,T], \mathbf{u}_0 \sim p_0(\mathbf{u}_0), \mathbf{u}_t \sim p_t(\mathbf{u}_t | \mathbf{u}_0)} \left[ \lambda(t) \left( \ell_t^{\text{DSM}} \right)^2 \| \boldsymbol{\epsilon}_{d:2d} - \alpha_{\boldsymbol{\theta}}(\mathbf{u}_t, t) \|_2^2 \right], \quad (54)$$

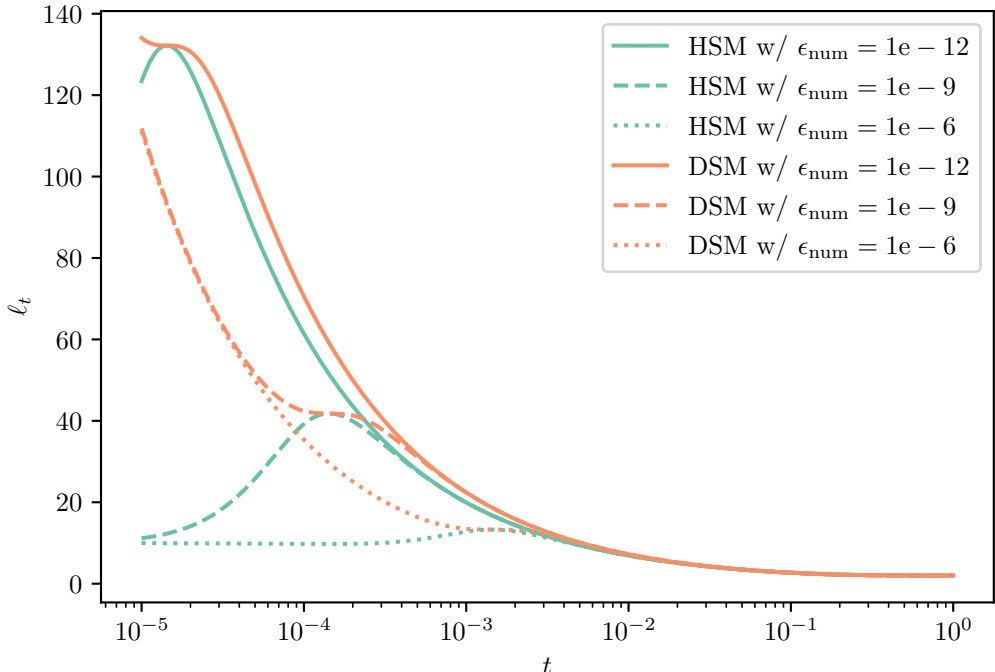

Figure 6: Comparison of $\ell_t^{\mathrm{HSM}}$ (in green) and $\ell_t^{\mathrm{DSM}}$ (in orange) for our main hyperparameter setting with $M = 0.25$ and $\gamma = 0.04$. In contrast to $\ell_t^{\mathrm{DSM}}$, $\ell_t^{\mathrm{HSM}}$ is analytically bounded. Nevertheless, numerical computation can be unstable (even when using double precision) in which case adding a numerical stabilization of $\epsilon_{\mathrm{num}} = 10^{-9}$ to the covariance matrix before computing $\ell_t$ suffices to make HSM work (see App. B.4).

where $\mathbf{u}_t$ is sampled via reparameterization:

$$\mathbf{u}_t = \boldsymbol{\mu}_t + \boldsymbol{L}_t \boldsymbol{\epsilon} = \boldsymbol{\mu}_t + \begin{pmatrix} L_t^{xx}\boldsymbol{\epsilon}_{0:d} \\ L_t^{xv}\boldsymbol{\epsilon}_{0:d} + L_t^{vv}\boldsymbol{\epsilon}_{d:2d} \end{pmatrix}. \tag{55}$$

Note again that $L_t$ is different for HSM and DSM.

Analogously to prior work (Ho et al., 2020; Vahdat et al., 2021; Song et al., 2021b) an objective better suited for high quality image synthesis can be obtained by "dropping the variance prefactor":

$$\mathrm{HSM}\left(\lambda(t) = \left(\ell_t^{\mathrm{HSM}}\right)^{-2}\right) = \mathbb{E}_{t \sim \mathcal{U}[0,T], \mathbf{x}_0 \sim p_0(\mathbf{x}_0), \mathbf{u}_t \sim p_t(\mathbf{u}_t|\mathbf{x}_0)}\left[\left\|\boldsymbol{\epsilon}_{d:2d} - \alpha_{\boldsymbol{\theta}}(\mathbf{u}_t, t)\right\|_2^2\right], \tag{56}$$

$$\mathrm{DSM}\left(\lambda(t) = \left(\ell_t^{\mathrm{DSM}}\right)^{-2}\right) = \mathbb{E}_{t \sim \mathcal{U}[0,T], \mathbf{u}_0 \sim p_0(\mathbf{u}_0), \mathbf{u}_t \sim p_t(\mathbf{u}_t|\mathbf{u}_0)}\left[\left\|\boldsymbol{\epsilon}_{d:2d} - \alpha_{\boldsymbol{\theta}}(\mathbf{u}_t, t)\right\|_2^2\right]. \tag{57}$$

### B.4 CLD-SPECIFIC IMPLEMENTATION DETAILS

Analytically, $\ell_t^{\mathrm{HSM}}$ is bounded (in particular, $\ell_0^{\mathrm{HSM}} = 1/\sqrt{\gamma M}$), whereas $\ell_t^{\mathrm{DSM}}$ is diverging for $t \to 0$. In practice, however, we found that computation of $\ell_t^{\mathrm{HSM}}$ can also be numerically unstable, even when using double precision. As is common practice for computing Cholesky decompositions, we add a numerical stabilization matrix $\epsilon_{\mathrm{num}}\boldsymbol{I}_{2d}$ to $\boldsymbol{\Sigma}_t$ before computing $\ell_t$. In Fig. 6, we visualize $\ell_t^{\mathrm{HSM}}$ and $\ell_t^{\mathrm{DSM}}$ for different values of $\epsilon_{\mathrm{num}}$ using our main experimental setup of $M = 0.25$ and $\gamma = 0.04$ (also, recall that in practice we have $T = 1$). Note that a very small numerical stabilization of $\epsilon_{\mathrm{num}} = 10^{-9}$ in combination with the use of double precision makes HSM work in practice.

### B.5 LOWER BOUNDS AND PROBABILITY FLOW ODE

Given the score model $s_{\boldsymbol{\theta}}(\mathbf{u}_t, t)$, we can synthesize novel samples via simulating the reverse-time diffusion SDE, Eq. (2) in the main text. This can be achieved, for example, via our novel SSCS,

Euler-Maruyama, or methods such as GGF (Jolicoeur-Martineau et al., 2021a). However, Song et al. (2021b;c) have shown that a corresponding ordinary differential equation can be defined that generates samples from the same distribution, in case $s_{\boldsymbol{\theta}}(\mathbf{u}_t, t)$ models the ground truth scores perfectly. This ODE is:

$$d\bar{\mathbf{u}}_t = \left[ -\boldsymbol{f}(\bar{\mathbf{u}}_t, T-t) + \frac{1}{2} \boldsymbol{G}(\bar{\mathbf{u}}_t, T-t) \boldsymbol{G}(\bar{\mathbf{u}}_t, T-t)^\top \nabla_{\bar{\mathbf{u}}_t} \log p_{T-t}(\bar{\mathbf{u}}_t) \right] dt \qquad (58)$$

This ODE is often referred to as the probability flow ODE. We can use it to generate novel data by sampling the prior and solving this ODE, like previous works (Song et al., 2021c). Note that in practice $s_{\boldsymbol{\theta}}(\mathbf{u}_t, t)$ won't be a perfect model, though, such that the generative models defined by simulating the reverse-time SDE and the probability flow ODE are not exactly equivalent (Song et al., 2021b). Nevertheless, they are very closely connected and it has been shown that their performance is usually very similar or almost the same, when we have learnt a good $s_{\boldsymbol{\theta}}(\mathbf{u}_t, t)$. In addition to sampling the generative SDE in our paper, we also sample from our CLD-based SGMs via this probability flow approach.

With the definition of our CLD, the ODE becomes:

$$\begin{pmatrix} d\bar{\mathbf{x}}_t \\ d\bar{\mathbf{v}}_t \end{pmatrix} = \underbrace{\begin{pmatrix} -M^{-1}\bar{\mathbf{v}}_t \\ \bar{\mathbf{x}}_t \end{pmatrix} \beta dt}_{A_H'} + \underbrace{\begin{pmatrix} \mathbf{0}_d \\ \Gamma \left[ \mathbf{s}(\bar{\mathbf{u}}_t, T-t) + M^{-1}\bar{\mathbf{v}}_t \right] \end{pmatrix} \beta dt}_{S'} \qquad (59)$$

Notice the interesting form of this probability flow ODE for CLD: It corresponds to Hamiltonian dynamics ($A_H'$) plus the score function term $S'$. Compared to the generative SDE (Sec. 3.3), the Ornstein-Uhlenbeck term disappears. Generally, symplectic integrators are best suited for integrating Hamiltonian systems (Neal, 2011; Tuckerman, 2010; Leimkuhler & Reich, 2005). However, our ODE is not perfectly Hamiltonian, due to the score term, and modern non-symplectic methods, such as the higher-order adaptive-step size Runge-Kutta 4(5) ODE integrator (Dormand & Prince, 1980), which we use in practice to solve the probability flow ODE, can also accurately simulate Hamiltonian systems over limited time horizons.

Importantly, the ODE formulation also allows us to estimate the log-likelihood of given test data, as it essentially defines a continuous Normalizing flow (Chen et al., 2018; Grathwohl et al., 2019), that we can easily run in either direction. However, in CLD the input into this ODE is not just the data $\mathbf{x}_0$, but also the velocity variable $\mathbf{v}_0$. In this case, we can still calculate a lower bound on the log-likelihood:

$$\begin{aligned} \log p(\mathbf{x}_0) &= \log \left( \int p(\mathbf{x}_0, \mathbf{v}_0) d\mathbf{v}_0 \right) \\ &= \log \left( \int p(\mathbf{v}_0) \frac{p(\mathbf{x}_0, \mathbf{v}_0)}{p(\mathbf{v}_0)} d\mathbf{v}_0 \right) \\ &\geq \mathbb{E}_{\mathbf{v}_0 \sim p(\mathbf{v}_0)} \left[ \log p(\mathbf{x}_0, \mathbf{v}_0) - \log p(\mathbf{v}_0) \right] \\ &= \mathbb{E}_{\mathbf{v}_0 \sim p(\mathbf{v}_0)} \left[ \log p(\mathbf{x}_0, \mathbf{v}_0) \right] + H(p(\mathbf{v}_0)) \end{aligned} \qquad (60)$$

where $H(p(\mathbf{v}_0))$ denotes the entropy of $p(\mathbf{v}_0)$ (we have $H(p(\mathbf{v}_0)) = \frac{1}{2} \log(2\pi e \gamma M)$). We can obtain a stochastic, but unbiased estimate of $\log p(\mathbf{x}_0, \mathbf{v}_0) \approx \log p_\varepsilon(\mathbf{x}_0, \mathbf{v}_0)$ via solving the probability flow ODE with initial conditions $(\mathbf{x}_0, \mathbf{v}_0)$ and calculating a stochastic estimate of the log-determinant of the Jacobian via Hutchinson's trace estimator (and also calculating the probability of the output under the prior), as done in Normalizing flows (Chen et al., 2018; Grathwohl et al., 2019) and previous works on SGMs (Song et al., 2021c;b). In the main paper, we report the negative of Eq. (60) as our upper bound on the negative log-likelihood (NLL).

Note that this bound can be potentially quite loose. In principle, it would be desirable to perform an importance-weighted estimation of the log-likelihood, as in importance-weighted autoencoders (Burda et al., 2015), using multiple samples from the velocity distribution. However, this isn't possible, as we only have access to a *stochastic* estimate $\log p_\varepsilon(\mathbf{x}_0, \mathbf{v}_0)$. The problems arising from this are discussed in detail in Appendix F of Vahdat et al. (2021). We could consider training a velocity encoder network, somewhat similar to Chen et al. (2020), to improve our bound, but we leave this for future research.

### B.6  On Introducing a Hamiltonian Component into the Diffusion

Here, we provide additional high-level intuitions and motivations about adding the Hamiltonian component to the diffusion process, as is done in our CLD.

Let us recall how the data distribution evolves in the forward diffusion process of SGMs: The role of the diffusion is to bring the initial non-equilibrium state quickly towards the equilibrium or prior distribution. Suppose for a moment, we could do so with "pure" Hamiltonian dynamics (no noise injection). In that case, we could generate data from the backward model without learning a score or neural network at all, because Hamiltonian dynamics is analytically invertible (flipping the sign of the velocity, we can just integrate backwards in reverse time direction). Now, this is not possible in practice, since Hamiltonian dynamics alone usually cannot convert the non-equilibrium distribution to the prior distribution. Nevertheless, Hamiltonian dynamics essentially achieves a certain amount of mixing on its own; moreover, since it is deterministic and analytically invertible, this mixing comes at no cost in the sense that we do not have to learn a complex score function to invert the Hamiltonian dynamics. Our thought experiment shows that we should strive for a diffusion process that behaves as deterministically (meaning that deterministic implies easily invertible) as possible with as little noise injection as possible. And this is exactly what is achieved by adding the Hamiltonian component in the overall diffusion process. In fact, recall that it is the diffusion coefficient $G$ of the forward SDE that ultimately scales the score function term of the backward generative SDE (and it is the score function that is hard to approximate with complex neural nets). Therefore, in other words, relying more on a deterministic Hamiltonian component for enhanced mixing (mixing just like in MCMC in that it brings us quickly towards the target distribution, in our case the prior) and less on pure noise injection will lead to a nicer generative SDE that relies less on a score function that requires complex and approximate neural network-based modeling, but more on a simple and analytical Hamiltonian component. Such an SDE could then be solved easier with an appropriate integrator (like our SSCS). In the end, we believe that this is the reason why our networks are "smoother" and why given the same network capacity and limited compute budgets we essentially outperform all previous results in the literature (on CIFAR-10).

We would also like to offer a second perspective, inspired by the Markov chain Monte Carlo (MCMC) literature. In MCMC, "mixing" helps to quickly traverse the high probability parts of the target distribution and, if an MCMC chain is initialized far from the high probability manifold, to quickly converge to this manifold. However, this is precisely the situation we are in with the forward diffusion process of SGMs: The system is initialized in a far-from-equilibrium state (the data distribution) and we need to traverse the space as efficiently as possible to converge to the equilibrium distribution, this is, the prior. Without efficient mixing, it takes longer to converge to the prior, which also implies a longer generation path in the reverse direction—which intuitively corresponds to a harder problem. Therefore, we believe that ideas from the MCMC literature that accelerate mixing and traversal of state space may be beneficial also for the diffusions in SGMs. In fact, leveraging Hamiltonian dynamics to accelerate sampling is popular in the MCMC field (Neal, 2011). Note that this line of reasoning extends to thermostating techniques from statistical mechanics and molecular dynamics, which essentially tackle similar problems like MCMC methods from the statistics literature (see discussion in Sec. 4).

## C  HSM: Hybrid Score Matching

We begin by recalling our objective function from App. B.3 (Eq. (44)):

$$\mathbb{E}_{t\sim\mathcal{U}[0,T]}\left[\lambda(t)\mathbb{E}_{\mathbf{u}_t\sim p_t(\mathbf{u})}[\|\nabla_{\mathbf{v}_t}\log p_t(\mathbf{u}_t) - s_{\boldsymbol{\theta}}(\mathbf{u}_t,t)\|_2^2]\right], \tag{61}$$

where $s_{\boldsymbol{\theta}}(\mathbf{u},t)$ is our score model. In the following, we dissect the "score matching" part of the above objective:

$$\mathcal{L}_{\mathrm{SM}} := \mathbb{E}_{\mathbf{u}_t\sim p_t(\mathbf{u}_t)}\left[\|\nabla_{\mathbf{v}_t}\log p_t(\mathbf{u}_t) - s_{\boldsymbol{\theta}}(\mathbf{u}_t,t)\|_2^2\right]$$
$$= \mathbb{E}_{\mathbf{u}_t\sim p_t(\mathbf{u}_t)}\|s_{\boldsymbol{\theta}}(\mathbf{u}_t,t)\|_2^2 - 2S(\boldsymbol{\theta}) + C_2(t). \tag{62}$$

where $C_2(t) := \mathbb{E}_{\mathbf{u}_t\sim p_t(\mathbf{u}_t)}\left[\|\nabla_{\mathbf{v}_t}\log p_t(\mathbf{u}_t)\|_2^2\right]$ and $S(\boldsymbol{\theta})$ is the cross term discussed below. Following Vincent (2011), we can rewrite $\mathcal{L}_{\mathrm{SM}}$ as an equivalent (up to addition of a time-dependent

constant) denoising score matching objective $\mathcal{L}_{\mathrm{DSM}}$:

$$\mathcal{L}_{\mathrm{DSM}} := \mathbb{E}_{\mathbf{u}_0 \sim p(\mathbf{u}_0), \mathbf{u}_t \sim p_t(\mathbf{u}_t | \mathbf{u}_0)} \| \nabla_{\mathbf{v}_t} \log p_t(\mathbf{u}_t \mid \mathbf{u}_0) - s_{\boldsymbol{\theta}}(\mathbf{u}_t, t) \|_2^2$$
$$= \mathcal{L}_{\mathrm{SM}} + C_3(t) - C_2(t),$$
(63)

where $C_3(t) := \mathbb{E}_{\mathbf{u}_0 \sim p(\mathbf{u}_0), \mathbf{u}_t \sim p_t(\mathbf{u}_t | \mathbf{u}_0)} \left[ \| \nabla_{\mathbf{v}_t} \log p_t(\mathbf{u}_t \mid \mathbf{u}_0) \|_2^2 \right]$. Something that might not necessarily be quite obvious is that there is no fundamental need to "denoise" with the distribution $p(\mathbf{u}_0)$ (this is, use samples from the joint $\mathbf{x}_0$-$\mathbf{v}_0$ distribution $p(\mathbf{u}_0)$, perturb them, and learn the score for denoising).

Instead, we can "denoise" only with the data distribution $p(\mathbf{x}_0)$ and marginalize over the entire initial velocity distribution $p(\mathbf{v}_0)$, which results in

$$\mathcal{L}_{\mathrm{HSM}} := \mathbb{E}_{\mathbf{x}_0 \sim p(\mathbf{x}_0), \mathbf{u}_t \sim p_t(\mathbf{u}_t | \mathbf{x}_0)} \| \nabla_{\mathbf{v}_t} \log p_t(\mathbf{u}_t \mid \mathbf{x}_0) - s_{\boldsymbol{\theta}}(\mathbf{u}_t, t) \|_2^2$$
$$= \mathbb{E}_{\mathbf{u}_t \sim p_t(\mathbf{u}_t)} \| s_{\boldsymbol{\theta}}(\mathbf{u}_t, t) \|_2^2 - 2 \mathbb{E}_{\mathbf{x}_0 \sim p(\mathbf{x}_0), \mathbf{u}_t \sim p_t(\mathbf{u}_t | \mathbf{x}_0)} [\langle \nabla_{\mathbf{v}_t} \log p_t(\mathbf{u}_t \mid \mathbf{x}_0), s_{\boldsymbol{\theta}}(\mathbf{u}_t, t) \rangle] + C_4(t),$$
(64)

where $C_4(t) := \mathbb{E}_{\mathbf{x}_0 \sim p(\mathbf{x}_0), \mathbf{u}_t \sim p_t(\mathbf{u}_t | \mathbf{x}_0)} \left[ \| \nabla_{\mathbf{v}_t} \log p_t(\mathbf{u}_t \mid \mathbf{x}_0) \|_2^2 \right]$ and $\langle \cdot, \cdot \rangle$ donates the inner product (notation chosen to be consistent with Vincent (2011)). In our case, this makes sense since $p(\mathbf{v}_0)$ is Normal, and therefore (as shown in App B.1), the perturbation kernel $p_t(\mathbf{u}_t \mid \mathbf{x}_0)$ is still Normal.

In the following, for completeness, we redo the derivation of Vincent (2011) and show that $\mathcal{L}_{\mathrm{SM}}$ is equivalent to $\mathcal{L}_{\mathrm{HSM}}$ (up to addition of a constant). Starting from $S(\boldsymbol{\theta})$, we have

$$S(\boldsymbol{\theta}) = \mathbb{E}_{\mathbf{u}_t \sim p_t(\mathbf{u}_t)} \langle \nabla_{\mathbf{v}_t} \log p_t(\mathbf{u}_t), s_{\boldsymbol{\theta}}(\mathbf{u}_t, t) \rangle$$
$$= \int_{\mathbf{u}_t} p_t(\mathbf{u}_t) \langle \nabla_{\mathbf{v}_t} \log p_t(\mathbf{u}_t), s_{\boldsymbol{\theta}}(\mathbf{u}_t, t) \rangle \, d\mathbf{u}_t$$
$$= \int_{\mathbf{u}_t} \langle \nabla_{\mathbf{v}_t} p_t(\mathbf{u}_t), s_{\boldsymbol{\theta}}(\mathbf{u}_t, t) \rangle \, d\mathbf{u}_t$$
$$= \int_{\mathbf{u}_t} \left\langle \nabla_{\mathbf{v}_t} \int_{\mathbf{x}_0} p_t(\mathbf{u}_t \mid \mathbf{x}_0) p_0(\mathbf{x}_0) \, d\mathbf{x}_0, s_{\boldsymbol{\theta}}(\mathbf{u}_t, t) \right\rangle \, d\mathbf{u}_t$$
(65)
$$= \int_{\mathbf{u}_t} \left\langle \int_{\mathbf{x}_0} p_t(\mathbf{u}_t \mid \mathbf{x}_0) p_0(\mathbf{x}_0) \nabla_{\mathbf{v}_t} \log p_t(\mathbf{u}_t \mid \mathbf{x}_0) \, d\mathbf{x}_0, s_{\boldsymbol{\theta}}(\mathbf{u}_t, t) \right\rangle \, d\mathbf{u}_t$$
$$= \int_{\mathbf{u}_t} \int_{\mathbf{x}_0} p_t(\mathbf{u}_t \mid \mathbf{x}_0) p_0(\mathbf{x}_0) \langle \nabla_{\mathbf{v}_t} \log p_t(\mathbf{u}_t \mid \mathbf{x}_0), s_{\boldsymbol{\theta}}(\mathbf{u}_t, t) \rangle \, d\mathbf{x}_0 \, d\mathbf{u}_t$$
$$= \mathbb{E}_{\mathbf{x}_0 \sim p_0(\mathbf{x}_0), \mathbf{u}_t \sim p(\mathbf{u}_t | \mathbf{x}_0)} [\langle \nabla_{\mathbf{v}_t} \log p_t(\mathbf{u}_t \mid \mathbf{x}_0), s_{\boldsymbol{\theta}}(\mathbf{u}_t, t) \rangle].$$

Hence, we have that

$$\mathcal{L}_{\mathrm{HSM}} = \mathcal{L}_{\mathrm{SM}} + C_4(t) - C_2(t).$$
(66)

This further implies that

$$\mathcal{L}_{\mathrm{HSM}} = \mathcal{L}_{\mathrm{DSM}} + C_4(t) - C_3(t).$$
(67)

Using the analysis from App B.1, we realize that $C_3$ and $C_4$ can be simplified to $d \left( \ell_t^{\mathrm{DSM}} \right)^2$ and $d \left( \ell_t^{\mathrm{HSM}} \right)^2$, respectively. Here, we used the fact that the expected squared norm of a multivariate standard Normal random variable is equal to its dimension, i.e., $\mathbb{E}_{\varepsilon \sim \mathcal{N}(\mathbf{0}_d, \boldsymbol{I}_d)} \| \varepsilon \|_2^2 = d$. This analysis then simplifies Eq. (67) to

$$\mathcal{L}_{\mathrm{HSM}} = \mathcal{L}_{\mathrm{DSM}} + d \left( \left( \ell_t^{\mathrm{DSM}} \right)^2 - \left( \ell_t^{\mathrm{HSM}} \right)^2 \right).$$
(68)

Using this relation, we can also find a connection between our CLD objective functions from App. B.3. In particular, we have

$$\mathrm{HSM}(\lambda(t)) = \mathbb{E}_{t \sim \mathcal{U}[0,T]} [\lambda(t) \mathcal{L}_{\mathrm{HSM}}]$$
$$= \mathbb{E}_{t \sim \mathcal{U}[0,T]} [\lambda(t) \mathcal{L}_{\mathrm{DSM}}] + d \, \mathbb{E}_{t \sim \mathcal{U}[0,T]} \left[ \lambda(t) \left( \left( \ell_t^{\mathrm{DSM}} \right)^2 - \left( \ell_t^{\mathrm{HSM}} \right)^2 \right) \right],$$
(69)
$$= \mathrm{DSM}(\lambda(t)) + d \, \mathbb{E}_{t \sim \mathcal{U}[0,T]} \left[ \lambda(t) \left( \left( \ell_t^{\mathrm{DSM}} \right)^2 - \left( \ell_t^{\mathrm{HSM}} \right)^2 \right) \right].$$

## C.1 GRADIENT VARIANCE REDUCTION VIA HSM

Above, we derived that $\mathcal{L}_{\text{HSM}} = \mathcal{L}_{\text{DSM}} + const$, so one might wonder why we advocate for HSM over DSM. As discussed in Sec. 3.2, one advantage of HSM is that it avoids unbounded scores at $t \to 0$. However, there is a second advantage: In practice, we never solve expectations analytically but rather approximate them using Monte Carlo estimates. In the remainder of this section, we will show that in practice (Monte Carlo) gradients based on HSM have lower variance than those based on DSM.

From Eq. (69), we have

$$
\begin{aligned}
\nabla_{\boldsymbol{\theta}} \text{HSM}(\lambda(t)) &= \mathbb{E}_{t \sim \mathcal{U}[0,T]} \left[ \lambda(t) \nabla_{\boldsymbol{\theta}} \mathcal{L}_{\text{HSM}} \right] \\
&= \mathbb{E}_{t \sim \mathcal{U}[0,T]} \left[ \lambda(t) \nabla_{\boldsymbol{\theta}} \mathcal{L}_{\text{DSM}} \right], \\
&= \nabla_{\boldsymbol{\theta}} \text{DSM}(\lambda(t)),
\end{aligned}
\tag{70}
$$

where $\boldsymbol{\theta}$ are the learnable parameters of the neural network. Instead of comparing the above expectations directly, we instead compare $\lambda(t) \nabla_{\boldsymbol{\theta}} \mathcal{L}_{\text{HSM}}$ with $\lambda(t) \nabla_{\boldsymbol{\theta}} \mathcal{L}_{\text{DSM}}$ for $t \in [0, 1]$ (we use $T = 1$ in all experiments) at discretized time values (as is done in practice). Replacing $\mathcal{L}_{\text{HSM}}$ and $\mathcal{L}_{\text{DSM}}$ with a single Monte Carlo estimate (as is used in practice), we have

$$
\lambda(t) \nabla_{\boldsymbol{\theta}} \mathcal{L}_{\text{HSM}} \approx \lambda(t) \nabla_{\boldsymbol{\theta}} s_\theta(\mathbf{u}_t, t) \nabla_{s_\theta(\mathbf{u}_t, t)} \| \nabla_{\mathbf{v}_t} \log p_t(\mathbf{u}_t \mid \mathbf{x}_0) - s_{\boldsymbol{\theta}}(\mathbf{u}_t, t) \|_2^2,
\tag{71}
$$
$$
\mathbf{x}_0 \sim p(\mathbf{x}_0), \mathbf{u}_t \sim p_t(\mathbf{u}_t \mid \mathbf{x}_0),
$$

$$
\lambda(t) \nabla_{\boldsymbol{\theta}} \mathcal{L}_{\text{DSM}} \approx \lambda(t) \nabla_{\boldsymbol{\theta}} s_\theta(\mathbf{u}_t, t) \nabla_{s_\theta(\mathbf{u}_t, t)} \| \nabla_{\mathbf{v}_t} \log p_t(\mathbf{u}_t \mid \mathbf{u}_0) - s_{\boldsymbol{\theta}}(\mathbf{u}_t, t) \|_2^2,
\tag{72}
$$
$$
\mathbf{u}_0 \sim p(\mathbf{u}_0), \mathbf{u}_t \sim p_t(\mathbf{u}_t \mid \mathbf{u}_0),
$$

where we applied the chain-rule. Note that in Eq. (71) and Eq. (72), $\mathbf{u}_t$ is sampled from the same distribution. Hence, $\lambda(t) \nabla_{\boldsymbol{\theta}} s_\theta(\mathbf{u}_t, t)$ acts as a common scaling factor, with the variance difference between HSM and DSM originating from the squared norm term. Hence, we ignore $\lambda(t) \nabla_{\boldsymbol{\theta}} s_\theta(\mathbf{u}_t, t)$ and only focus our analysis on the gradient of the norm terms, which we can further simplify:

$$
\frac{1}{2} \nabla_{s_\theta(\mathbf{u}_t, t)} \| \nabla_{\mathbf{v}_t} \log p_t(\mathbf{u}_t \mid \mathbf{x}_0) - s_{\boldsymbol{\theta}}(\mathbf{u}_t, t) \|_2^2 = s_{\boldsymbol{\theta}}(\mathbf{u}_t, t) - \nabla_{\mathbf{v}_t} \log p_t(\mathbf{u}_t \mid \mathbf{x}_0) =: \mathcal{K}_{\text{HSM}},
\tag{73}
$$

and

$$
\frac{1}{2} \nabla_{s_\theta(\mathbf{u}_t, t)} \| \nabla_{\mathbf{v}_t} \log p_t(\mathbf{u}_t \mid \mathbf{u}_0) - s_{\boldsymbol{\theta}}(\mathbf{u}_t, t) \|_2^2 = s_{\boldsymbol{\theta}}(\mathbf{u}_t, t) - \nabla_{\mathbf{v}_t} \log p_t(\mathbf{u}_t \mid \mathbf{u}_0) =: \mathcal{K}_{\text{DSM}}.
\tag{74}
$$

We explore this difference in a realistic setup; in particular, we evaluate $\mathcal{K}_{\text{HSM}}$ and $\mathcal{K}_{\text{DSM}}$ for all data points in the CIFAR-10 training set. We choose $s_{\boldsymbol{\theta}}$ to be our trained ablation CLD model (with the standard setup of $M^{-1} = \beta = 4$, see Sec. E.2.1 for model details). We then use these samples to compute the empirical covariance matrices $\text{Cov}_{\text{HSM}}$ and $\text{Cov}_{\text{DSM}}$ of the random variables $\mathcal{K}_{\text{HSM}}$ and $\mathcal{K}_{\text{DSM}}$, respectively.

As is common practice in statistics, we consider only the trace of the estimated covariance matrices.[4] The trace of the covariance matrix (of a random variable) is also commonly referred to as the total variation (of a random variable).

We visualize our results in Fig. 7. For HSM, there is barely any visual difference in $\text{Tr}(\text{Cov})$ for $\gamma = 0.04$ and $\gamma = 1$. For DSM, both $\gamma = 0.04$ and $\gamma = 1$ result in very large $\text{Tr}(\text{Cov})$ values for small $t$. For large $t$, $\text{Tr}(\text{Cov})$ is considerably smaller for $\gamma = 0.04$ than for $\gamma = 1$. However, in practice, we found that DSM is even unstable for small $\gamma$. Given this analysis, we believe this is due to the large gradient variance for small $t$. In conclusion, these results demonstrate a clear variance reduction by the HSM objective, in particular for large $\gamma$. Ultimately, this is expected: In HSM, we are effectively integrating out the initial velocity distribution when estimating gradients, while in DSM we use noisy samples for the initial velocity.

Note that re-introducing $\lambda(t)$ weightings would allow us to scale the $\text{Tr}(\text{Cov})$ curves according to the "reweighted" objective or the maximum likelihood objective. However, we believe it is most instructive to directly analyze the gradient of the relevant norm term itself.

---

[4]Arguably, the most prominent algorithm that follows this practice is principal component analysis (PCA).

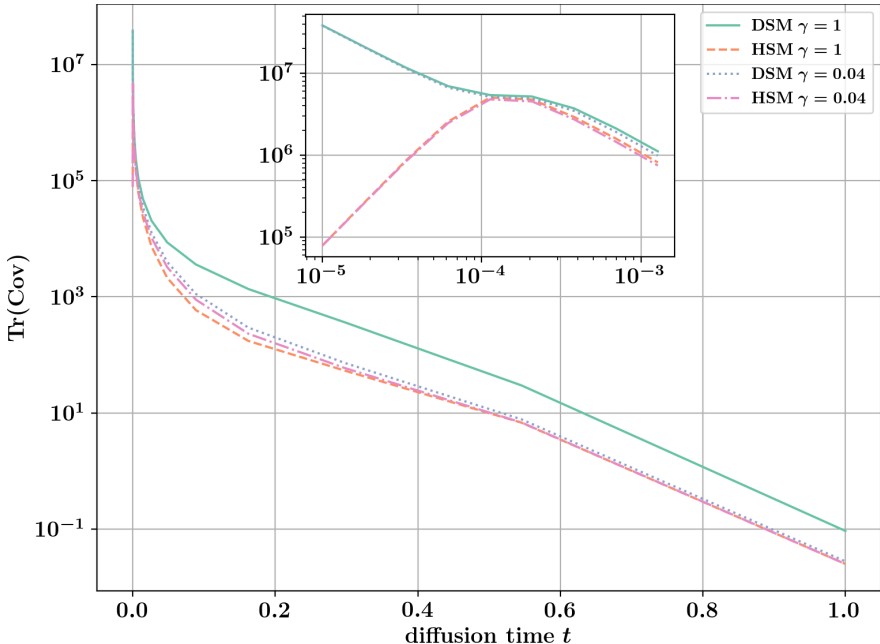

Figure 7: Traces of the estimated covariance matrices.

# D SYMMETRIC SPLITTING CLD SAMPLER (SSCS)

In this section, we present a more complete derivation and analysis of our novel Symmetric Splitting CLD Sampler (SSCS).

## D.1 BACKGROUND

Our derivation is inspired by methods from the statistical mechanics and molecular dynamics literature. In particular, we are leveraging symmetric splitting techniques as well as (Fokker–Planck) operator concepts. The high-level idea of symmetric splitting as well as the operator formalism are well-explained in Tuckerman (2010), in particular in their Section 3.10, which includes simple examples. Symmetric splitting methods for stochastic dynamics in particular are discussed in detail in Leimkuhler & Matthews (2015). We also recommend Leimkuhler & Matthews (2013), which discusses splitting methods for Langevin dynamics in a concise but insightful manner.

## D.2 DERIVATION AND ANALYSIS

**Generative SDE.** From Sec. 3.3, recall that our generative SDE can be written as (with $\bar{\mathbf{u}}_t = \mathbf{u}_{T-t}$, $\bar{\mathbf{x}}_t = \mathbf{x}_{T-t}$, $\bar{\mathbf{v}}_t = \mathbf{v}_{T-t}$):

$$\begin{pmatrix} d\bar{\mathbf{x}}_t \\ d\bar{\mathbf{v}}_t \end{pmatrix} = \underbrace{\begin{pmatrix} -M^{-1}\bar{\mathbf{v}}_t \\ \bar{\mathbf{x}}_t \end{pmatrix} \beta dt}_{A_H} + \underbrace{\begin{pmatrix} \mathbf{0}_d \\ -\Gamma M^{-1}\bar{\mathbf{v}}_t \end{pmatrix} \beta dt + \begin{pmatrix} \mathbf{0}_d \\ \sqrt{2\Gamma\beta}d\mathbf{w}_t \end{pmatrix}}_{A_O} + \underbrace{\begin{pmatrix} \mathbf{0}_d \\ 2\Gamma \left[ \mathbf{s}(\bar{\mathbf{u}}_t, T-t) + M^{-1}\bar{\mathbf{v}}_t \right] \end{pmatrix} \beta dt}_{S}.$$

(75)

**Fokker–Planck Equation and Fokker–Planck Operators.** The evolution of the probability distribution $p_{T-t}(\bar{\mathbf{u}}_t)$ is described by the general Fokker–Planck equation (Särkkä & Solin, 2019):

$$\frac{\partial p_{T-t}(\bar{\mathbf{u}}_t)}{\partial t} = -\sum_{i=1}^{2d} \frac{\partial}{\partial \bar{u}_i} \left[ \mu_i(\bar{\mathbf{u}}_t, T-t) p_{T-t}(\bar{\mathbf{u}}_t) \right] + \sum_{i=1}^{2d} \sum_{j=1}^{2d} \frac{\partial^2}{\partial \bar{u}_i \partial \bar{u}_j} \left[ D_{ij}(\bar{\mathbf{u}}_t, T-t) p_{T-t}(\bar{\mathbf{u}}_t) \right],$$

(76)

with

$$\boldsymbol{\mu}(\bar{\mathbf{u}}_t, T - t) = \begin{pmatrix} -M^{-1}\bar{\mathbf{v}}_t \\ \bar{\mathbf{x}}_t \end{pmatrix}\beta + \begin{pmatrix} \mathbf{0}_d \\ -\Gamma M^{-1}\bar{\mathbf{v}}_t \end{pmatrix}\beta + \begin{pmatrix} \mathbf{0}_d \\ 2\Gamma\left[\mathbf{s}(\bar{\mathbf{u}}_t, T - t) + M^{-1}\bar{\mathbf{v}}_t\right] \end{pmatrix}\beta, \quad (77)$$

$$D(\bar{\mathbf{u}}_t, T - t) = \begin{pmatrix} 0 & 0 \\ 0 & \Gamma\beta \end{pmatrix} \otimes \boldsymbol{I}_d. \tag{78}$$

For our SDE, we can write the Fokker–Planck equation in short form as

$$\frac{\partial p_{T-t}(\bar{\mathbf{u}}_t)}{\partial t} = (\hat{\mathcal{L}}_A^* + \hat{\mathcal{L}}_S^*)p_{T-t}(\bar{\mathbf{u}}_t), \tag{79}$$

with the Fokker–Planck operators (defined via their action on functions of the variables $\phi(\bar{\mathbf{u}}_t)$):

$$\hat{\mathcal{L}}_A^*\phi(\bar{\mathbf{u}}_t) := \beta M^{-1}\bar{\mathbf{v}}_t\nabla_{\bar{\mathbf{x}}_t}\phi(\bar{\mathbf{u}}_t) - \beta\bar{\mathbf{x}}_t\nabla_{\bar{\mathbf{v}}_t}\phi(\bar{\mathbf{u}}_t) + \Gamma\beta M^{-1}\nabla_{\bar{\mathbf{v}}_t}\left[\bar{\mathbf{v}}_t\phi(\bar{\mathbf{u}}_t)\right] + \Gamma\beta\Delta_{\bar{\mathbf{v}}_t}\phi(\bar{\mathbf{u}}_t),$$

$$\tag{80}$$

$$\hat{\mathcal{L}}_S^*\phi(\bar{\mathbf{u}}_t) := -2\Gamma\beta\nabla_{\bar{\mathbf{v}}_t}\left[\left(\mathbf{s}(\bar{\mathbf{u}}_t, T - t) + M^{-1}\bar{\mathbf{v}}_t\right)\phi(\bar{\mathbf{u}}_t)\right], \tag{81}$$

$$\Delta_{\bar{\mathbf{v}}_t} := \sum_{i=1}^{d}\left(\frac{\partial^2}{\partial\bar{x}_i^2} + \frac{\partial^2}{\partial\bar{v}_i^2}\right). \tag{82}$$

We are providing these formulas for transparency and completeness. We do not directly leverage them. However, working with these operators can be convenient. In particular, the operators describe the time evolution of states $\bar{\mathbf{u}}_t$ under the stochastic dynamics defined by the SDE. Given an initial state $\bar{\mathbf{u}}_0$, we can construct a formal solution to the generative SDE via (Tuckerman, 2010; Leimkuhler & Matthews, 2015):

$$\bar{\mathbf{u}}_t = e^{t(\hat{\mathcal{L}}_A^* + \hat{\mathcal{L}}_S^*)}\bar{\mathbf{u}}_0, \tag{83}$$

where the operator $e^{t(\hat{\mathcal{L}}_A^* + \hat{\mathcal{L}}_S^*)}$ is known as the classical propagator that propagates states $\bar{\mathbf{u}}_0$ for time $t$ according to the dynamics defined by the combined Fokker–Planck operators $\hat{\mathcal{L}}_A^* + \hat{\mathcal{L}}_S^*$ (to avoid confusion, note that in Eq. (83) the operator $e^{t(\hat{\mathcal{L}}_A^* + \hat{\mathcal{L}}_S^*)}$ is applied on $\bar{\mathbf{u}}_0$ in an element-wise or "vectorized" fashion on all elements of $\bar{\mathbf{u}}_0$ in parallel). The problem with that expression is that we cannot analytically evaluate it. However, we can leverage it to design an integration method.

**Symmetric Splitting Integration.** Using the symmetric Trotter theorem or Strang splitting formula as well as the Baker–Campbell–Hausdorff formula (Trotter, 1959; Strang, 1968; Tuckerman, 2010), it can be shown that:

$$e^{t(\hat{\mathcal{L}}_A^* + \hat{\mathcal{L}}_S^*)} = \lim_{N\to\infty}\left[e^{\frac{\delta t}{2}\hat{\mathcal{L}}_A^*}e^{\delta t\hat{\mathcal{L}}_S^*}e^{\frac{\delta t}{2}\hat{\mathcal{L}}_A^*}\right]^N \approx \left[e^{\frac{\delta t}{2}\hat{\mathcal{L}}_A^*}e^{\delta t\hat{\mathcal{L}}_S^*}e^{\frac{\delta t}{2}\hat{\mathcal{L}}_A^*}\right]^N + \mathcal{O}(N\delta t^3), \quad (84)$$

for large $N \in \mathbb{N}^+$ and time step $\delta t := t/N$. The expression suggests that instead of directly evaluating the intractable $e^{t(\hat{\mathcal{L}}_A^* + \hat{\mathcal{L}}_S^*)}$, we can discretize the dynamics over $t$ into $N$ pieces of step size $\delta t$, such that we only need to apply the *individual* $e^{\frac{\delta t}{2}\hat{\mathcal{L}}_A^*}$ and $e^{\delta t\hat{\mathcal{L}}_S^*}$ many times one after another for small time steps $\delta t$. A finer discretization implies a smaller error (since $N = t/\delta t$ the error effectively scales as $\mathcal{O}(\delta t^2)$ for fixed $t$). Hence, this implies an integration method. The general idea of such splitting schemes is to split an initially intractable propagator into separate terms, each of which is analytically tractable. In that case, the overall integration error for many steps is only due to the splitting scheme error,[5] but not due to the evaluation of the individual updates. Such techniques are, for example, popular in molecular dynamics to develop symplectic integrators as well as accurate samplers (Tuckerman et al., 1992; Tuckerman, 2010; Leimkuhler & Matthews, 2013; 2015; Bussi & Parrinello, 2007).

**Analyzing the Splitting Terms.** Next, we need to analyze the two individual terms:

**(i)** Let us first analyze $e^{\frac{\delta t}{2}\hat{\mathcal{L}}_A^*}\bar{\mathbf{u}}_t$: This term describes the stochastic evolution of $\bar{\mathbf{u}}_t$ under the dynamics of an SDE like Eq. (75), but with $S$ set to zero. However, if $S$ is set to zero, the remaining

---

[5]In principle, the error of the splitting scheme is defined more specifically by the commutator of the non-commuting Fokker–Planck operators. See, for example Leimkuhler & Matthews (2013; 2015); Tuckerman (2010).

SDE has affine drift and diffusion coefficients. In that case, if the input is Normal (or a discrete state corresponding to a Normal with $0$ variance) then the distribution is Normal at all times and we can calculate the evolution analytically. In particular, we can solve the differential equations for the mean $\bar{\boldsymbol{\mu}}_{\frac{\delta t}{2}}$ and covariance $\bar{\boldsymbol{\Sigma}}_{\frac{\delta t}{2}}$ of the Normal (see Sec. B.1), and obtain

$$\bar{\boldsymbol{\mu}}_{\frac{\delta t}{2}}(\bar{\mathbf{u}}_t) = \begin{pmatrix} 2\beta\frac{\delta t}{2}\Gamma^{-1}\bar{\mathbf{x}}_t - 4\beta\frac{\delta t}{2}\Gamma^{-2}\bar{\mathbf{v}}_t + \bar{\mathbf{x}}_t \\ \beta\frac{\delta t}{2}\bar{\mathbf{x}}_t - 2\beta\frac{\delta t}{2}\Gamma^{-1}\bar{\mathbf{v}}_t + \bar{\mathbf{v}}_t \end{pmatrix} e^{-2\beta\frac{\delta t}{2}\Gamma^{-1}}, \tag{85}$$

as well as

$$\bar{\boldsymbol{\Sigma}}_{\frac{\delta t}{2}} = \bar{\Sigma}_{\frac{\delta t}{2}} \otimes \boldsymbol{I}_d, \tag{86}$$

$$\bar{\Sigma}_{\frac{\delta t}{2}} = \begin{pmatrix} \bar{\Sigma}^{xx}_{\frac{\delta t}{2}} & \bar{\Sigma}^{xv}_{\frac{\delta t}{2}} \\ \bar{\Sigma}^{xv}_{\frac{\delta t}{2}} & \bar{\Sigma}^{vv}_{\frac{\delta t}{2}} \end{pmatrix} e^{-4\beta\frac{\delta t}{2}\Gamma^{-1}}, \tag{87}$$

$$\bar{\Sigma}^{xx}_{\frac{\delta t}{2}} = e^{4\beta\frac{\delta t}{2}\Gamma^{-1}} - 1 - 4\beta\frac{\delta t}{2}\Gamma^{-1} - 8\left(\beta\frac{\delta t}{2}\right)^2\Gamma^{-2}, \tag{88}$$

$$\bar{\Sigma}^{xv}_{\frac{\delta t}{2}} = -4\left(\beta\frac{\delta t}{2}\right)^2\Gamma^{-1}, \tag{89}$$

$$\bar{\Sigma}^{vv}_{\frac{\delta t}{2}} = \frac{\Gamma^2}{4}\left(e^{4\beta\frac{\delta t}{2}\Gamma^{-1}} - 1\right) + \beta\frac{\delta t}{2}\Gamma - 2\left(\beta\frac{\delta t}{2}\right)^2. \tag{90}$$

The correctness of the proposed mean and covariance matrix can be verified by simply plugging them back in their respective ODEs; see App. G.2.

Now, we can write the action of the the propagator $e^{\frac{\delta t}{2}\hat{\mathcal{L}}^*_A}$ on a state $\bar{\mathbf{u}}_t$ as:

$$e^{\frac{\delta t}{2}\hat{\mathcal{L}}^*_A}\bar{\mathbf{u}}_t \sim \mathcal{N}(\bar{\mathbf{u}}_{t+\frac{\delta t}{2}}; \bar{\boldsymbol{\mu}}_{\frac{\delta t}{2}}(\bar{\mathbf{u}}_t), \bar{\boldsymbol{\Sigma}}_{\frac{\delta t}{2}}). \tag{91}$$

**(ii)**: Next, we need to analyze $e^{\delta t\hat{\mathcal{L}}^*_S}\bar{\mathbf{u}}_t$. Unfortunately, we cannot calculate the action of the propagator $e^{\delta t\hat{\mathcal{L}}^*_S}$ on $\bar{\mathbf{u}}_t$ analytically and we need to make an approximation. From Eq. (75), we can easily see that the propagator $e^{\delta t\hat{\mathcal{L}}^*_S}$ describes the evolution of the velocity component $\bar{\mathbf{v}}_t$ for time step $\delta t$ under the ODE (this can be easily seen by noticing that the $S$ term in Eq. (75) only acts on the velocity component of the joint state $\bar{\mathbf{u}}_t$):

$$d\bar{\mathbf{v}}_t = 2\beta\Gamma\left[\mathbf{s}(\bar{\mathbf{u}}_t, T-t) + M^{-1}\bar{\mathbf{v}}_t\right]dt. \tag{92}$$

We propose to simply solve this ODE for the step $\delta t$ via a simple step of Euler's method, resulting in:

$$e^{\delta t\hat{\mathcal{L}}^*_S}\bar{\mathbf{u}}_t \approx \bar{\mathbf{u}}_t + \delta t\begin{pmatrix} \mathbf{0}_d \\ 2\beta\Gamma\left[\mathbf{s}(\bar{\mathbf{u}}_t, T-t) + M^{-1}\bar{\mathbf{v}}_t\right] \end{pmatrix} + \mathcal{O}(\delta t^2)$$

$$= e^{\delta t\hat{\mathcal{L}}^{*\,\mathrm{Euler}}_S}\bar{\mathbf{u}}_t + \mathcal{O}(\delta t^2), \tag{93}$$

with the informal definition

$$e^{\delta t\hat{\mathcal{L}}^{*\,\mathrm{Euler}}_S}\bar{\mathbf{u}}_t := \bar{\mathbf{u}}_t + \delta t\begin{pmatrix} \mathbf{0}_d \\ 2\beta\Gamma\left[\mathbf{s}(\bar{\mathbf{u}}_t, T-t) + M^{-1}\bar{\mathbf{v}}_t\right] \end{pmatrix}. \tag{94}$$

**Error Analysis.** It is now instructive to study the overall error of our proposed integrator. With the additional Euler integration in one of the splitting terms, we have

$$e^{t(\hat{\mathcal{L}}^*_A + \hat{\mathcal{L}}^*_S)} \approx \left[e^{\frac{\delta t}{2}\hat{\mathcal{L}}^*_A}\left(e^{\delta t\hat{\mathcal{L}}^{*\,\mathrm{Euler}}_S} + \mathcal{O}(\delta t^2)\right)e^{\frac{\delta t}{2}\hat{\mathcal{L}}^*_A}\right]^N + \mathcal{O}(N\delta t^3)$$

$$= \left[e^{\frac{\delta t}{2}\hat{\mathcal{L}}^*_A}\left(e^{\delta t\hat{\mathcal{L}}^{*\,\mathrm{Euler}}_S}\right)e^{\frac{\delta t}{2}\hat{\mathcal{L}}^*_A}\right]^N + N\mathcal{O}(\delta t^2) \tag{95}$$

$$= \left[e^{\frac{\delta t}{2}\hat{\mathcal{L}}^*_A}\left(e^{\delta t\hat{\mathcal{L}}^{*\,\mathrm{Euler}}_S}\right)e^{\frac{\delta t}{2}\hat{\mathcal{L}}^*_A}\right]^N + \mathcal{O}(\delta t),$$

where we used $N = \frac{t}{\delta t}$ and only kept the dominating error terms of lowest order in $\delta t$. We see that, just like Euler's method, also our SSCS is a first-order integrator with local error $\sim\delta t^2$ and global

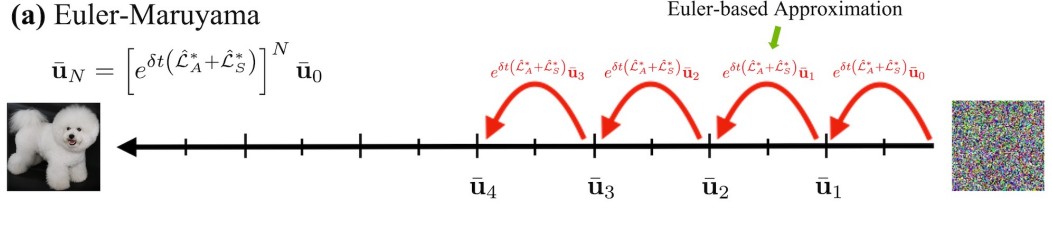

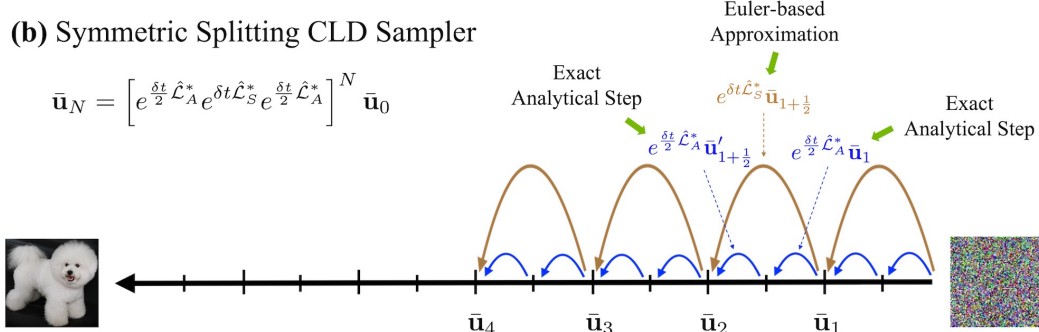

Figure 8: Conceptual visualization of our new SSCS sampler and comparison to Euler-Maruyama (for image synthesis): **(a)** In EM-based sampling, in each integration step the entire SDE is integrated using an Euler-based approximation. This can be formally expressed as solving the full-step propagator $\exp\left\{\delta t(\hat{\mathcal{L}}_A^* + \hat{\mathcal{L}}_S^*)\right\}$ via Euler-based approximation for $N$ small steps of size $\delta t$ (see **red** steps; for simplicity, this visualization assumes constant $\delta t$). **(b)**: In contrast, in our SSCS the propagator is partitioned into an analytically tractable component $\exp\left\{\frac{\delta t}{2}\hat{\mathcal{L}}_A^*\right\}$ (**blue**) and the score model term $\exp\left\{\delta t\hat{\mathcal{L}}_S^*\right\}$ (**brown**). Only the latter requires numerical approximation, which results in an overall more accurate integration scheme.

error $\sim\delta t$, which can be also seen from the last two lines of Eq. (95). This is expected, considering that we used an Euler step for the $S$ term. Nevertheless, as long as the dynamics is not dominated by the $S$ component, our proposed integration scheme is still expected to be more accurate than EM, since we split off the analytically tractable part and only use an Euler approximation for the $S$ term.

To this end, recall that the model only needs to learn the score of the conditional distribution $p_t(\mathbf{v}_t|\mathbf{x}_t)$, which is close to Normal for much of the diffusion, in which case the $S$ term will indeed be small. This suggests that the generative SDE dynamics are in fact dominated by $A_H$ and $A_O$ in practice. From another perspective, note that (recalling that $s_{\boldsymbol{\theta}}(\mathbf{u}_t, t) = -\ell_t\alpha_{\boldsymbol{\theta}}(\mathbf{u}_t, t)$ with $\alpha_{\boldsymbol{\theta}}(\mathbf{u}_t, t) = \ell_t^{-1}\mathbf{v}_t/\Sigma_t^{vv} + \alpha'_{\boldsymbol{\theta}}(\mathbf{u}_t, t)$ from Sec. 3.2):

$$\begin{aligned}
\mathbf{s}(\bar{\mathbf{u}}_t, T - t) + M^{-1}\bar{\mathbf{v}}_t &= -\ell_t\alpha'_{\boldsymbol{\theta}}(\mathbf{u}_t, t) - \frac{\mathbf{v}_t}{\Sigma_t^{vv}} + M^{-1}\bar{\mathbf{v}}_t \\
&= -\ell_t\alpha'_{\boldsymbol{\theta}}(\mathbf{u}_t, t) + \bar{\mathbf{v}}_t\left(\frac{1}{M} - \frac{1}{\Sigma_t^{vv}}\right).
\end{aligned}$$

(96)

For large parts of the diffusion, $\Sigma_t^{vv}$ is indeed close to $M$, such that the $\bar{\mathbf{v}}_t$ term is very small (this cancellation is the reason why we pulled the $M^{-1}\bar{\mathbf{v}}_t$ term into the $S$ component). In Sec. 3, we have also seen that our neural network component $\alpha'_{\boldsymbol{\theta}}(\mathbf{u}_t, t)$ can be much smoother than that of previous SGMs. Overall, this suggests that the error of SSCS indeed might be smaller than the error we would obtain when applying a naive Euler–Maruyama integrator to the full generative SDE. Our positive experimental results in Sec. 5.2 validate that. Only in the limit for very small steps, both our SSCS and EM make only very small errors and are expected to perform equally well, which is exactly what we observe in our experiments. Our SSCS turns out to be well-suited for integrating the generative SDE of CLD-SGMs with relatively few synthesis steps.

Note that error analysis of stochastic differential equation solvers is usually performed in terms of weak and strong convergence (Kloeden & Platen, 1992). Due to the use of Euler's method for the $S$ component, as argued above, we expect our SSCS to formally have the same weak and strong convergence properties like EM, this is, weak convergence of order 1 and strong convergence of order 1 as well, since the noise is additive in our case (and assuming appropriate smoothness conditions for the drift and diffusion coefficients; furthermore, without additive noise, we would have strong convergence of order 0.5). We leave a more detailed analysis to future work.

In practice, we do not use SSCS to integrate all the way from $t=0$ to $t=T$, but only up to $t=T-\epsilon$, and perform a denoising step, similar to previous works (Jolicoeur-Martineau et al., 2021a; Song et al., 2021c). It is worth noting that our SSCS scheme would also be applicable when we used time-dependent $\beta(t)$, as in our more general derivation of the CLD perturbation kernel in App. B. However, since we only used constant $\beta$ in the main paper, we also presented SSCS in that way.

A promising direction for future work would be to extend SSCS to adaptive step sizes and to use techniques to facilitate higher-order integration, while still leveraging the advantages of SSCS.

**SSCS Algorithm.** Finally, we summarize SSCS in terms of a concise algorithm:

---
**Algorithm 1** *Symmetric Splitting CLD Sampler (SSCS)*
---

**Input:** Score function $\mathbf{s}_{\boldsymbol{\theta}}(\bar{\mathbf{u}}_t, T-t)$, CLD parameters $\Gamma$, $\beta$, $M=\Gamma^2/4$, number of sampling steps $N$, step sizes $\{\delta t_n \geq 0\}_{n=0}^{N-1}$ chosen such that $\epsilon := T - \sum_{n=0}^{N-1} \delta t_n \geq 0$ (stepsizes can vary, for example in QS).
**Output:** Synthesized model sample $\bar{\mathbf{x}}_N'$, along with a velocity sample $\bar{\mathbf{v}}_N'$.

$\bar{\mathbf{x}}_0 \sim \mathcal{N}(\bar{\mathbf{x}}_0; \mathbf{0}_d, \boldsymbol{I}_d), \bar{\mathbf{v}}_0 \sim \mathcal{N}(\bar{\mathbf{v}}_0; \mathbf{0}_d, M\boldsymbol{I}_d), \bar{\mathbf{u}}_0 = (\bar{\mathbf{x}}_0, \bar{\mathbf{v}}_0)$  ▷ Draw initial prior samples from $p_{\mathrm{EQ}}(\mathbf{u})$
$t = 0$  ▷ Initialize time
**for** $n = 0$ **to** $N-1$ **do**
  $\bar{\mathbf{u}}_{n+\frac{1}{2}} \sim \mathcal{N}(\bar{\mathbf{u}}_{n+\frac{1}{2}}; \bar{\boldsymbol{\mu}}_{\frac{\delta t_n}{2}}(\bar{\mathbf{u}}_n), \bar{\boldsymbol{\Sigma}}_{\frac{\delta t_n}{2}})$  ▷ First half-step: apply $\exp\{\frac{\delta t_n}{2}\hat{\mathcal{L}}_A^*\}$ on $\bar{\mathbf{u}}_n$
  $\bar{\mathbf{u}}_{n+\frac{1}{2}}' \leftarrow \bar{\mathbf{u}}_{n+\frac{1}{2}} + \delta t_n \begin{pmatrix} \mathbf{0}_d \\ 2\beta\Gamma\left[\mathbf{s}(\bar{\mathbf{u}}_t, T-t) + M^{-1}\bar{\mathbf{v}}_t\right] \end{pmatrix}$  ▷ Full step: apply $\exp\{\delta t_n \hat{\mathcal{L}}_S^*\}$ on $\bar{\mathbf{u}}_{n+\frac{1}{2}}$
  $\bar{\mathbf{u}}_{n+1} \sim \mathcal{N}(\bar{\mathbf{u}}_{n+1}; \bar{\boldsymbol{\mu}}_{\frac{\delta t_n}{2}}(\bar{\mathbf{u}}_{n+\frac{1}{2}}'), \bar{\boldsymbol{\Sigma}}_{\frac{\delta t_n}{2}})$  ▷ Second half-step: apply $\exp\{\frac{\delta t_n}{2}\hat{\mathcal{L}}_A^*\}$ on $\bar{\mathbf{u}}_{n+\frac{1}{2}}'$
  $t \leftarrow t + \delta t_n$  ▷ Update time
**end for**
$\bar{\mathbf{u}}_N' \leftarrow \bar{\mathbf{u}}_N - \epsilon \left( \begin{pmatrix} 0 & \beta M^{-1} \\ -\beta & -\Gamma\beta M^{-1} \end{pmatrix} \otimes \boldsymbol{I}_d \right) \bar{\mathbf{u}}_N + \epsilon \begin{pmatrix} \mathbf{0}_d \\ 2\beta\Gamma\mathbf{s}(\bar{\mathbf{u}}_t, \epsilon) \end{pmatrix}$  ▷ Denoising
$(\bar{\mathbf{x}}_N', \bar{\mathbf{v}}_N') = \bar{\mathbf{u}}_N'$  ▷ Extract output data and velocity samples

---

Note that the algorithm uses the expressions in Eqs. (85) and (86) for $\bar{\boldsymbol{\mu}}_t$ and $\bar{\boldsymbol{\Sigma}}_t$. Furthermore, in practice in the denoising step at the end, we usually only update the $\bar{\mathbf{x}}_N'$ component of $\bar{\mathbf{u}}_N'$, since we are only interested in the data sample. This saves us the final neural network call during denoising, which only affects the $\bar{\mathbf{v}}_N'$ component (also see App. E.2.4). However, we wrote the algorithm in the general way, which also allows to correctly generate the velocity sample $\bar{\mathbf{v}}_N'$. In Fig. 8, we show a conceptual visualization of our SSCS and contrast it to EM.

Also note that we could combine the second half-step from one iteration of SSCS with the first half-step from the next iteration of SSCS. This is commonly done in the Leapfrog integrator (Leimkuhler & Reich, 2005; Tuckerman, 2010; Neal, 2011; Leimkuhler & Matthews, 2015),[6] which follows a similar structure as our SSCS. However, it is not important in our case, as the only computationally costly operation is in the center full step, which involves the neural network evaluation. The first and last half-steps come at virtually no computational cost.

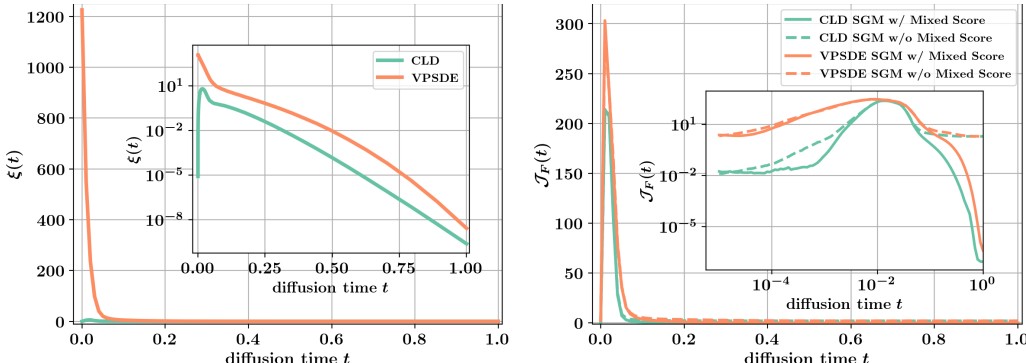

(a) Difference $\xi(t)$ (via $L2$ norm) between score of diffused data and score of Normal distribution.

(b) Frobenius norm of Jacobian $\mathcal{J}_F(t)$ of the neural network defining the score function for different $t$.

Figure 9: Toy experiments for mixture of Normals dataset.

# E  IMPLEMENTATION AND EXPERIMENT DETAILS

## E.1  SCORE AND JACOBIAN EXPERIMENTS

In this section, we provide details for the experiments presented in Sec. 3.1. For both experiments, we consider a two-dimensional simple mixture of Normals of the form

$$p_{\text{data}}(\mathbf{x}) = \sum_{k=1}^{9} \frac{1}{9} p^{(k)}(\mathbf{x}), \tag{97}$$

where $p^{(k)}(\mathbf{x}) = \mathcal{N}(\mathbf{x}; \boldsymbol{\mu}_k; 0.04^2 \boldsymbol{I}_2)$ and

$$\boldsymbol{\mu}_1 = \begin{pmatrix} -a \\ 0 \end{pmatrix}, \qquad \boldsymbol{\mu}_2 = \begin{pmatrix} -a/2 \\ a/2 \end{pmatrix}, \quad \boldsymbol{\mu}_3 = \begin{pmatrix} 0 \\ a \end{pmatrix},$$

$$\boldsymbol{\mu}_4 = \begin{pmatrix} -a/2 \\ -a/2 \end{pmatrix}, \quad \boldsymbol{\mu}_5 = \begin{pmatrix} 0 \\ 0 \end{pmatrix}, \qquad \boldsymbol{\mu}_6 = \begin{pmatrix} a/2 \\ a/2 \end{pmatrix},$$

$$\boldsymbol{\mu}_7 = \begin{pmatrix} 0 \\ -a \end{pmatrix}, \qquad \boldsymbol{\mu}_8 = \begin{pmatrix} a/2 \\ -a/2 \end{pmatrix}, \quad \boldsymbol{\mu}_9 = \begin{pmatrix} a \\ 0 \end{pmatrix},$$

and $a = 2^{-\frac{1}{2}}$. The choice of this data distribution is not arbitrary. In fact, mixture of Normal distributions are diffused by simply diffusing the components, i.e., setting $p_0(\mathbf{x}_0) = p_{\text{data}}(\mathbf{x})$, we have

$$p_t(\mathbf{x}_t) = \sum_{k=1}^{9} \frac{1}{9} p_t^{(k)}(\mathbf{x}_t), \tag{98}$$

where $p_t^{(k)}$ are the diffused components (analogously for CLD with velocity augmentation). This means that for both CLD as well as VPSDE Song et al. (2021c) we can diffuse $p_{\text{data}}(\mathbf{x})$ with analytical access to the diffused marginal $p_t(\mathbf{x}_t)$ or $p_t(\mathbf{u}_t)$. This allows us to perform interesting analyses that would be impossible when working, for example, with image data. We visualize the data distribution in Fig. 10.

**Score experiment:**  We empirically verify the reduced complexity of the score of $p_t(\mathbf{v}_t|\mathbf{x}_t)$, which is learned in CLD, compared to the score of $p_t(\mathbf{x}_t)$, which is learned in VPSDE. To avoid scaling issues between VPSDE and CLD, we chose $M = \gamma = 1$ for CLD in this experiment; this results in an equilibrium distribution of $\mathcal{N}(\mathbf{0}_2, \boldsymbol{I}_2)$ (for both data and velocity components, which are independent at equilibrium), which is the same as the equilibrium distribution of the VPSDE. We then

---
[6]The *Leapfrog integrator* corresponds to the *velocity Verlet integrator* in molecular dynamics.

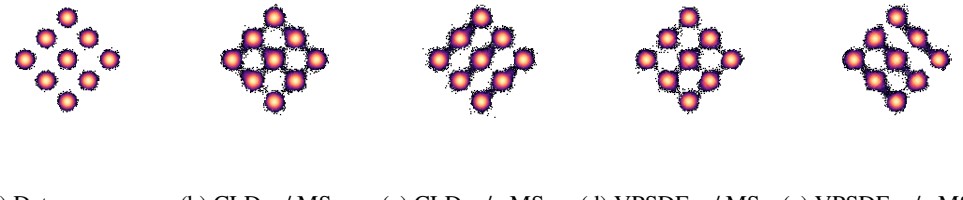

(a) Data $p_{\text{data}}$     (b) CLD w/ MS     (c) CLD w/o MS     (d) VPSDE w/ MS    (e) VPSDE w/o MS

Figure 10: Mixture of Normals: data and trained models (samples).

measure the difference of the respective scores at time $t$ and the equilibrium (or prior) scores, i.e. (recall that the score of a Normal distribution $p(\mathbf{x}) = \mathcal{N}(\mathbf{0}_2, \boldsymbol{I}_2)$ is simply $\nabla_{\mathbf{x}} \log p(\mathbf{x}) = -\mathbf{x}$),

$$\xi^{\text{VPSDE}}(t) \coloneqq \mathbb{E}_{\mathbf{x}_t \sim p(\mathbf{x}_t)} \|\nabla_{\mathbf{x}_t} \log p_t(\mathbf{x}_t) + \mathbf{x}_t\|_2^2, \tag{99}$$

$$\xi^{\text{CLD}}(t) \coloneqq \mathbb{E}_{\mathbf{u}_t \sim p(\mathbf{u}_t)} \|\nabla_{\mathbf{v}_t} \log p_t(\mathbf{v}_t \mid \mathbf{x}_t) + \mathbf{v}_t\|_2^2. \tag{100}$$

The expectations are approximated using $10^5$ samples from $p(\mathbf{x}_t)$ and $p(\mathbf{u}_t)$ for VPSDE and CLD, respectively. As can be seen in Fig. 9a, $\xi^{\text{CLD}}(t)$ is smaller than $\xi^{\text{VPSDE}}(t)$ for all $t \in [0, T]$. The difference is particularly striking for small time values $t$. Other previous SDEs, such as the VESDE, sub-VPSDE, etc., are expected to behave similarly. This result implies that the ground truth scores that need to be learnt in CLD are closer to Normal scores than the ground truth scores in previous SDEs like the VPSDE. Since the score of a Normal is very simple—and indeed directly leveraged in our mixed score formulation—we would intuitively expect that the CLD training task is easier.

**Complexity experiment:** Therefore, to understand the above observations in terms of learning neural networks, we train a small ResNet architecture (less than 100k parameters) for each of the following four setups: both CLD and VPSDE each with and without a mixed score parameterization. The mixed score of the VPSDE simply assumes a standard Normal data distribution (which is also the equilibrium distribution of VPSDE) resulting in adding $-\mathbf{x}_t$ to the score function. Formally, $-\mathbf{x}_t$ is the score of a Normal distribution with unit variance.

We train the models for 1M iterations using fresh data synthesized from $p_{\text{data}}$ at a batch size of 512. The model and data distributions are visualized in Fig. 10. We see that all models have learnt good representations of the data. We measure the complexity of the trained neural networks using the squared Frobenius norm of the networks' Jacobians. For CLD, we have

$$\mathcal{J}_{\text{F}}^{\text{CLD}}(t) \coloneqq \mathbb{E}_{\mathbf{u}_t \sim p(\mathbf{u}_t)} \|\nabla_{\mathbf{u}_t} \alpha'_{\boldsymbol{\theta}}(\mathbf{u}_t)\|_F^2. \tag{101}$$

Similarly, for the VPSDE we compute

$$\mathcal{J}_{\text{F}}^{\text{VPSDE}}(t) \coloneqq \mathbb{E}_{\mathbf{x}_t \sim p(\mathbf{x}_t)} \|\nabla_{\mathbf{x}_t} \alpha'_{\boldsymbol{\theta}}(\mathbf{x}_t)\|_F^2. \tag{102}$$

For both CLD and VPSDE, expectations are again approximated using $10^5$ samples. As can be seen in Fig. 9b the neural network complexity is significantly lower for CLD compared to VPSDE. A mixed score formulation further helps decreasing the neural network complexity for both CLD and VPSDE. This result implies that the arguably simpler training task in CLD indeed also translates to reduced model complexity in that the neural network is smoother as measured by $\mathcal{J}_{\text{F}}(t)$. In large-scale experiments, this would mean that, given similar model capacity, a CLD-based SGM could potentially have a higher expressivity. Or, on the other hand, similar performance could be achieved with a smoother and potentially smaller model. Indeed these findings are in line with our strong results on the CIFAR-10 benchmark.

### E.2    IMAGE MODELING EXPERIMENTS

We perform image modeling experiments on CIFAR-10 as well as CelebA-HQ-256. We report FID scores on CIFAR-10 for our main model for various different solvers; see Tab. 3 and Tab. 2. We further present generated samples for both models in Sec. 5 using Euler–Maruyama with 2000

| Hyperparameter | CIFAR10 (Main) | CelebA (Qualitative) | CIFAR10 (Ablation) |
|---|---|---|---|
| **Model** | | | |
| EMA rate | 0.9999 | 0.9999 | 0.9999 |
| # of ResBlocks per Resolution | 8 | 2 | 2 |
| Normalization | Group Normalization | Group Normalization | Group Normalization |
| Scaling by $\sigma$ | ✗ | ✗ | ✗ |
| Nonlinearity | Swish | Swish | Swish |
| Attention resolution | 16 | 16 | 16 |
| Embedding type | Fourier | Positional | Positional |
| Progressive | None | None | None |
| Progressive input | Residual | None | None |
| Progressive combine | Sum | N/A | N/A |
| Finite Impulse Response (Zhang, 2019) | ✓ | ✗ | ✗ |
| # of parameters | $\approx 108M$ | $\approx 68M$ | $\approx 39M$ |
| **Training** | | | |
| # of iterations | 800k | 320k | 1M |
| # of warmup iterations | 100k | 100k | 100k |
| Optimizer | Adam | Adam | Adam |
| Mixed precision | ✓ | ✓ | ✓ |
| Learning rate | $2 \cdot 10^{-4}$ | $10^{-4}$ | $2 \cdot 10^{-4}$ |
| Gradient norm clipping | 1.0 | 1.0 | 1.0 |
| Dropout | 0.1 | 0.1 | 0.1 |
| Batch size per GPU | 8 | 4 | 8 |
| # of GPUs | 16 | 16 | 16 |
| $t$-sampling cutoff during training | $10^{-5}$ | $10^{-5}$ | $10^{-5}$ |
| **SDE** | | | |
| $M$ | 0.25 | 0.25 | varies |
| $\gamma$ | 0.04 | 0.04 | varies |
| $\beta$ | 4 | 4 | varies |
| $\epsilon_{\text{num}}$ | $10^{-9}$ | $10^{-6}$ | $10^{-9}$ |

Table 6: Model architectures as well as SDE and training setups for our experiments on CIFAR-10 and CelebA-HQ-256.

quadratic striding steps and Runge–Kutta 4(5) for CIFAR10 and CelebA-HQ-256, respectively. We present additional samples for various solver settings in App. F. All (average) NFEs for the Runge–Kutta solver are computed using a batch size of 128.

### E.2.1  TRAINING DETAILS AND MODEL ARCHITECTURES

Our models are based on the NCSN++ and the DDPM++ architectures from Song et al. (2021c). Importantly, we changed the number of input channels from three to six to facilitate the additional velocity variables. Note that the number of additional neural network parameters due to this change is negligible.

For fair a comparison, we train our models using the same $t$-sampling cutoff during training as is used for VESDE and VPSDE in Song et al. (2021c). Note, however, that this is not strictly necessary for CLD as we do not have any "blow-up" of the SDE due to unbounded scores as $t \to 0$ (also see Fig. 18 and Fig. 19).

We summarize our three model architectures as well as our SDE and training setups in Tab. 6.

### E.2.2  CIFAR-10 RESULTS FOR VESDE AND VPSDE

The results reported for VESDE and VPSDE using the GGF sampler are taken from Jolicoeur-Martineau et al. (2021a). All other results for VESDE and VPSDE are generated using the provided PyTorch code as well as the provided checkpoints from Song et al. (2021c).[7] We used EM and PC

---

[7]https://github.com/yang-song/score_sde_pytorch

to sample from the VPSDE and VESDE models, respectively (see Sec. 5.2), since these choices correspond to their recommended settings.[8]

Furthermore, in App. F.2 we also used DDIM (Song et al., 2021a) to sample the VPSDE. DDIM's update rule is

$$\mathbf{x}_{t-1} = \frac{\alpha_{t-1}}{\alpha_t} \left[ \mathbf{x}_t + \sigma_t^2 s_{\boldsymbol{\theta}}(\mathbf{x}_t, t) \right] - \sigma_{t-1}\sigma_t s_{\boldsymbol{\theta}}(\mathbf{x}_t, t), \tag{103}$$

where $\alpha_t = \exp\left(-0.5 \int_0^t \beta(t)\, dt\right)$, $\sigma_t^2 = 1 - \exp\left(-\int_0^t \beta(t)\, dt\right)$, and $\beta(t) = 0.1 + 19.9t$.

### E.2.3   QUADRATIC STRIDING

When we simulate our generative SDE numerically, using for example EM or our SSCS, we need to choose a time discretization. Given a certain NFE budget $N_{\text{NFE}}$, how do we choose time step sizes? The standard approach is to use an equidistant discretization, corresponding to a set of evaluation time steps $\{\delta t_i \geq 0\}_{i=0}^{N_{\text{NFE}}-1}$ with $\delta t_i = \frac{1}{N_{\text{NFE}}} \forall\, i \in [0, N_{\text{NFE}} - 1]$. However, prior work (Song et al., 2021a; Kong & Ping, 2021; Watson et al., 2021) has shown that it can be beneficial to focus function evaluations (neural network calls) on times $t$ "close to the data". This is because the diffusion process distribution is most complex close to the data and almost perfectly Normal close to the prior. Among other techniques, these works used a useful heuristic, denoted as *quadratic striding* (QS), which discretizes the integration interval such that the evaluation times follow a quadratic schedule and the individual time steps follow a linear schedule. We also used this QS approach in our experiments.

We can formally define it as follows (assuming a time interval $[0.0, 1.0]$ here for simplicity): Denote the evaluation times as $\tau_i$ (including $0.0$ and $1.0$) and define:

$$\tau_i = c_\tau i^2 \quad \forall i \in [0, N_{\text{NFE}}]. \tag{104}$$

Hence,

$$\delta t_i = \tau_i - \tau_{i-1} = c_\tau(2i - 1) \quad \forall i \in [1, N_{\text{NFE}}], \tag{105}$$

and $c_\tau = \frac{1}{N_{\text{NFE}}^2}$ to ensure that $\tau_{\text{NFE}} = 1.0$.

This describes the time steps as going from $t = 0$ to $t = 1$. During synthesis, however, we are going backwards. Hence, we can define our time steps as

$$\delta t_j = c_\tau \left[ 2N_{\text{NFE}} - 2j + 1 \right] \quad \forall j \in [1, N_{\text{NFE}}], \tag{106}$$

where $j$ now counts time steps in the other direction. Note that this can be easily adapted to general integration intervals $[\epsilon, T]$.

### E.2.4   DENOISING

As has been pointed out in Jolicoeur-Martineau et al. (2021b), samples that are generated with models similar to ours can contain noise that is hard to detect visually but worsens FID scores significantly.

**Denoising Formulas.**   For a fair comparison we use the same denoising setup for all experiments we conducted (including VESDE (PC/ODE) and VPSDE (EM/ODE)) except for LSGM.[9] We simulate the underlying generative ODE/SDE until the time cutoff $\varepsilon = 10^{-3}$ and then take a single denoising step of the form

$$\mathbf{u}_0 = \mathbf{u}_\varepsilon - \varepsilon \boldsymbol{f}(\mathbf{u}_\varepsilon, \varepsilon) + \varepsilon \boldsymbol{G}(\mathbf{u}_\varepsilon, \varepsilon)\boldsymbol{G}(\mathbf{u}_\varepsilon, \varepsilon)^\top \begin{pmatrix} \mathbf{0}_d \\ s_{\boldsymbol{\theta}}(\mathbf{u}_\varepsilon, \varepsilon) \end{pmatrix}. \tag{107}$$

This denoising step can be considered as an Euler–Maruyama step without noise injection. For SDEs acting on data directly (VESDE, VPSDE, etc.) the corresponding denoising formula is

$$\mathbf{x}_0 = \mathbf{x}_\varepsilon - \varepsilon \boldsymbol{f}(\mathbf{x}_\varepsilon, \varepsilon) + \varepsilon \boldsymbol{G}(\mathbf{x}_\varepsilon, \varepsilon)\boldsymbol{G}(\mathbf{x}_\varepsilon, \varepsilon)^\top s_{\boldsymbol{\theta}}(\mathbf{x}_\varepsilon, \varepsilon). \tag{108}$$

---

[8] https://colab.research.google.com/drive/1dRR_0gNRmfLtPavX2APzUggBuXyjWW55

[9] Denoising has not been used in the original LSGM work (Vahdat et al., 2021) and is not needed in their case, since the output of the latent SGM lives in a smooth latent space and is further processed by a decoder.

Table 7: Influence of denoising step on FID scores (using our main CIFAR-10 model).

| | | FID at $n$ function evaluations ↓ | |
|---|---|---|---|
| Sampler | Denoising | $n{=}50$ | $n{=}500$ |
| SSCS | ✓ | 81.1 | 2.30 |
| SSCS-QS | ✓ | 20.5 | 2.25 |
| SSCS | ✗ | 78.9 | 2.32 |
| SSCS-QS | ✗ | 28.5 | 2.3 |

**Influence of Denoising on Results.**    For SDEs acting in the data space directly, it has been reported that this denoising step is crucial to obtain good FID scores Jolicoeur-Martineau et al. (2021b); Song et al. (2021c). When we simulate the generative probability flow ODE we found that denoising is important in order for the Runge–Kutta solver not to "blow-up" as $t \to 0$. On the other hand, when simulating CLD using our new SSCS solver, we found that denoising only slightly influences FID (see Tab. 7). We believe that this might be because the neural network does not have any influence on the denoising step for CLD. More specifically, the neural network only denoises the velocity component. However, we are primarily interested in the data component. Putting the drift and diffusion coefficients of CLD in the denoising formula in Eq. (107), we obtain

$$\mathbf{u}_0 = \mathbf{u}_\varepsilon - \varepsilon \left( \begin{pmatrix} 0 & \beta M^{-1} \\ -\beta & -\beta\Gamma M^{-1} \end{pmatrix} \otimes \boldsymbol{I}_d \right) \mathbf{u}_\varepsilon + \varepsilon \begin{pmatrix} \mathbf{0}_d \\ 2\Gamma\beta s_{\boldsymbol{\theta}}(\mathbf{u}_\varepsilon, \varepsilon) \end{pmatrix} \implies \mathbf{x}_0 = \mathbf{x}_\varepsilon - \varepsilon\beta M^{-1}\mathbf{v}_\varepsilon. \tag{109}$$

### E.2.5    SOLVER ERROR TOLERANCES FOR RUNGE–KUTTA 4(5)

In Tab. 2, we report FID scores for a Runge–Kutta 4(5) solver (Dormand & Prince, 1980) as well as the "Gotta Go Fast" solver from Jolicoeur-Martineau et al. (2021a) (see their Table 1). For simulating CLD with Runge–Kutta 4(5) we chose the solver error tolerances to hit certain regimes of NFEs to facilitate comparisons with VPSDE and VESDE. We obtain a mean number of function evaluations of 312 and 137 using Runge–Kutta 4(5) solver error tolerances of $10^{-5}$ and $10^{-3}$, respectively. For VESDE, VPSDE and LSGM we used $10^{-5}$ as the ODE solver error tolerance, following the recommended default setups (Song et al., 2021c; Vahdat et al., 2021). These values are used for both relative and absolute error tolerances.

### E.2.6    ABLATION EXPERIMENTS

The model architecture used for all ablation experiments can be found in Tab. 6. As pointed out in Sec. 5 we found that the hyperparameters $\gamma$ and $M$ only have small effects on CIFAR-10 FID scores. On the other hand, we found that the mixed score parameterization helps significantly in obtaining competitive FIDs.

### E.2.7    LSGM-100M MODEL

Our CLD-based SGM has $\approx 108M$ parameters, while the original CIFAR-10 Latent SGM from Vahdat et al. (2021), to which we compare in Tab. 1, uses $\approx 476M$ parameters. To establish a fairer comparison between our CLD-based SGMs and LSGM (Vahdat et al., 2021), we train another smaller LSGM model with $\approx 109M$ parameters. To do this, we followed the exact setup of the "CIFAR-10 (balanced)" model from LSGM (see Table 7 in Vahdat et al. (2021)), with a few minor modifications: We used a VAE backbone model with only 10 groups instead of 20, which corresponds to a reduction in parameters by a factor of 2 in the encoder and decoder networks. We also reduced the convolutional channels in the latent space SGM from 512 to 256 and reduced the number of the residual cells per scale from 8 to 4. With these modifications the resulting "LSGM-100M" uses only $\approx 109M$ parameters overall with approximately half of them in the encoder and decoder networks and the other half in the latent SGM. Other than these architecture modifications, our model is trained in exactly the same way as the bigger, original models in Vahdat et al. (2021).

Table 8: Performance (measured in negative log-likelihood) using analytical scores for non-adaptive stepsize solvers for varying numbers of synthesis steps $n$ (function evaluations).

| Model | Sampler | $-\log p(x)$ at $n$ function evaluations $\downarrow$ | | | |
|---|---|---|---|---|---|
| | | $n=20$ | $n=50$ | $n=100$ | $n=200$ |
| CLD | EM | 60.6 | 9.71 | 0.72 | -1.04 |
| CLD | SSCS | **10.5** | **1.55** | **-1.25** | **-1.54** |
| VPSDE | EM | 14.2 | 4.68 | -0.35 | -1.11 |

For evaluation, we follow the recommended setting by Vahdat et al. (2021) and use the same Runge-Kutta 4(5) ODE solver with an error tolerance of $10^{-5}$ to solve the probability flow ODE in LSGM's latent space. LSGM-100M achieves an FID of 4.60, an NLL bound of 2.96 bpd, and requires on average 131 NFE for sampling new images. We report these results in Tabs. 1 and 2 in the main text.

Note that we also tried training a model following the "CIFAR-10 (best FID)" setup, but found training to be unstable (however, the orignal "CIFAR-10 (best FID)" model from Vahdat et al. (2021) only performs marginally better in FID than their "CIFAR-10 (balanced)" model anyway). Furthermore, we also tried training another small LSGM with a similar number of parameters but with more parameters in the latent SGM and less in the encoder and decoder networks, compared to the reported LSGM-100M. However, this model performed significantly worse.

## F  ADDITIONAL EXPERIMENTS

### F.1  TOY EXPERIMENTS

#### F.1.1  ANALYTICAL SAMPLING

In order to test combinations of diffusions and numerical samplers in isolation, we consider a dataset for which we know the ground truth score function (for all $t$) analytically. In particular, we use the mixture of Normals introduced in App. E.1; see Fig. 10a for a visualization of the data distribution. In Fig. 11, we show samples for VPSDE (Euler–Maruyama (EM) sampler) and CLD (EM and SSCS samplers). For quantitative comparison, we also compute negative log-likelihoods for the three combinations (which can be done easily due to our access to the ground truth distribution): as can be seen in Tab. 8, for each number of steps $n \in \{20, 50, 100, 200\}$ CLD with SCSS outperforms both VPSDE and CLD with EM. As discussed in Sec. 3.3, we can see in Tab. 8 that EM is not well-suited for CLD. This is true, in particular, when using a small number of synthesis steps $n$ (function evaluations). In Fig. 11, we see that CLD with EM leads to sampling distributions which are too broad. These results are exactly in line with the "diverging" dynamics that is observed when solving Hamiltonian dynamics with a non-symplectic integrator, such as the standard Euler method (Neal, 2011). This problematic behavior of Euler-based techniques is more pronounced when using fewer steps with larger stepsizes, which is also what we observe in our experiments. These results further motivate the use of our novel SSCS, which addresses these challenges, for sampling from our CLD-based SGMs.

#### F.1.2  MAXIMUM LIKELIHOOD TRAINING

For maximum likelihood training, models based on overdamped Langevin dynamics such as VPSDE need to learn an unbounded score for $t \to 0$. Our model, on the other hand, only ever needs to learn a bounded score even for $t = 0$. For our image data experiments, we use a reweighted objective function to improve visual quality of samples (as is general practice).

Here, we also study training towards maximum likelihood on toy dataset tasks. To explore this, we repeat the neural network complexity experiment from App. E.1 with maximum likelihood training (instead of the reweighted objective). Furthermore, we also train VPSDE-based and CLD-based SGMs on a challenging toy dataset and find that CLD significantly outperforms VPSDE. We leave the study of CLD with maximum likelihood training for high-dimensional (image) datasets to future work.

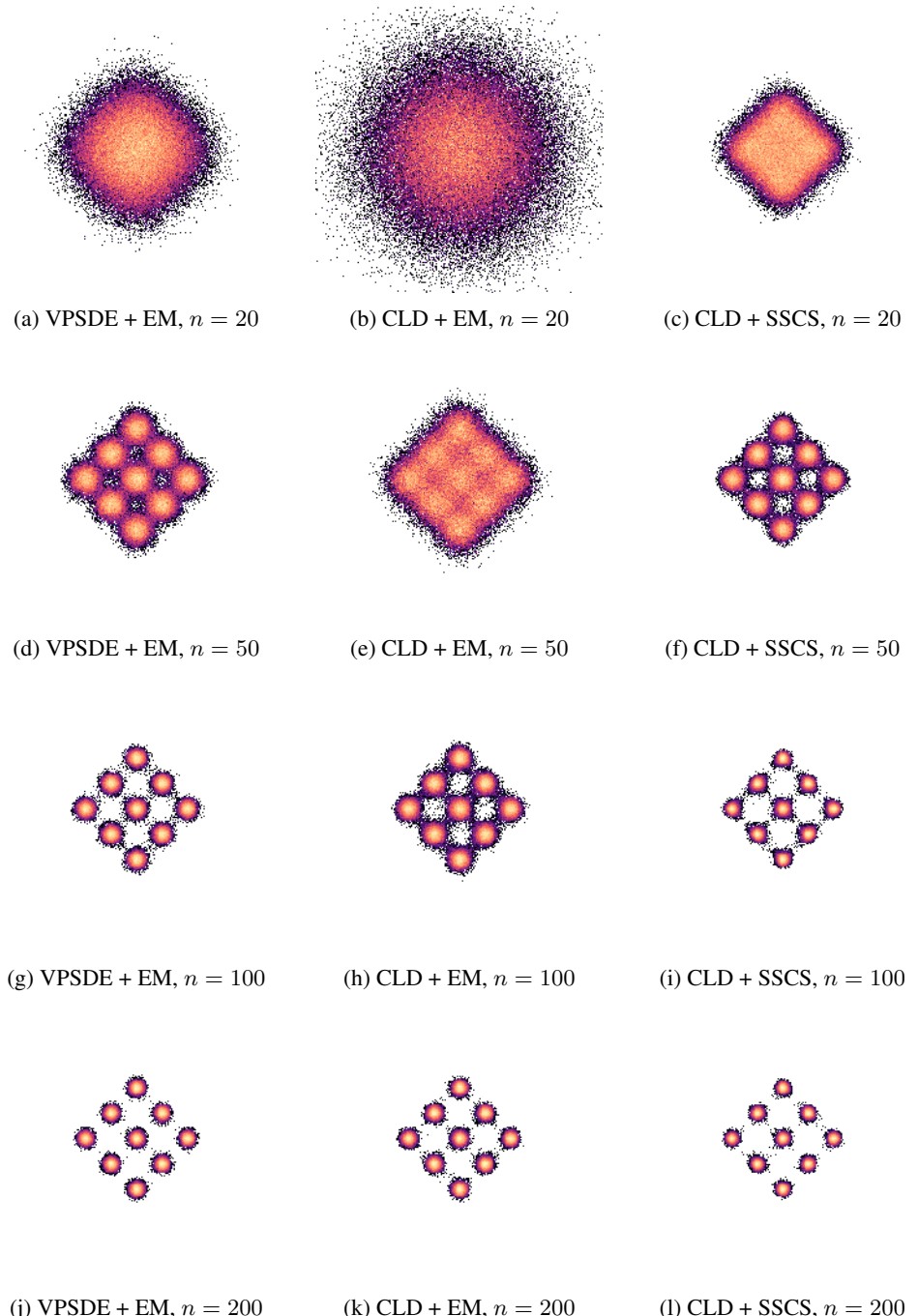

Figure 11: Mixture of Normals: numerical simulation with analytical score function for different diffusions (VPSDE with EM vs. CLD with EM/SSCS) and number of synthesis steps $n$. A visualization of the data distribution can be found in Fig. 10a.

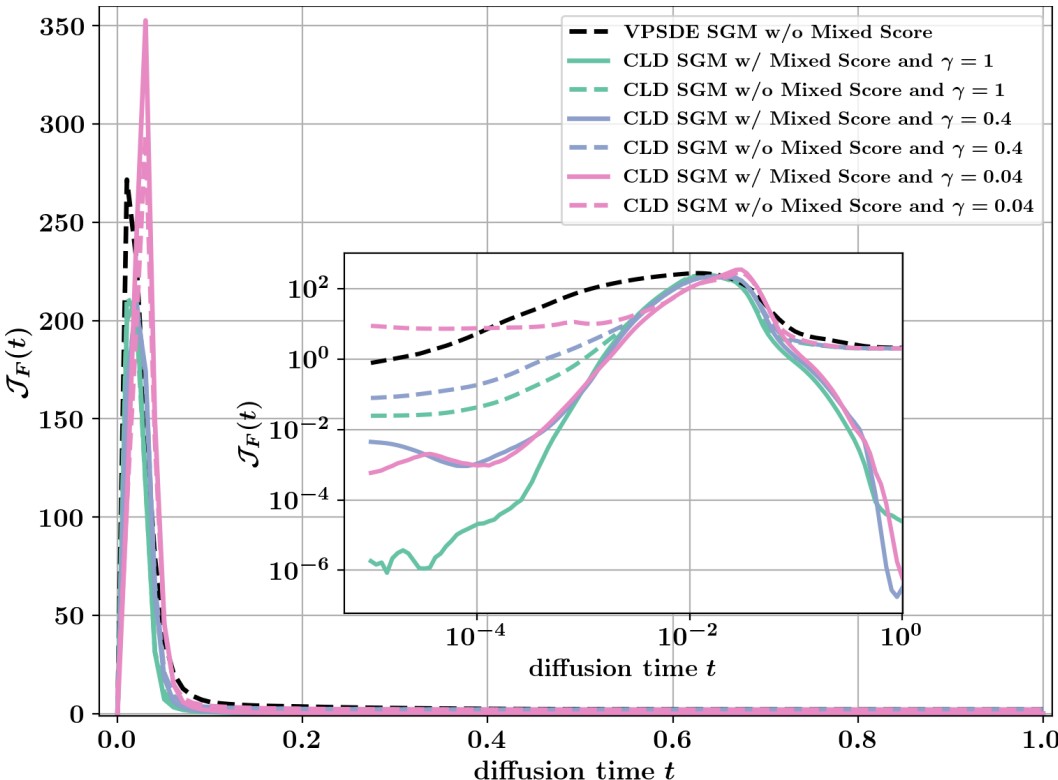

Figure 12: Frobenius norm $\mathcal{J}_F(t)$ of the neural network defining the score function for different $t$.

**Complexity Experiment.** The setup of this experiment is equivalent to the setup in App. E.1 up to the training objective: in this experiment we do maximum likelihood learning, i.e., we train CLD models with the objective from Eq. (8) with $\lambda(t) = \Gamma\beta$.[10] Furthermore, we test CLD in this setup for three different values of $\gamma$. The results of this experiment can be found in Fig. 12. For CLD, we find that larger values of $\gamma$ generally lead to less complex networks, in particular for smaller times $t$. However, even for $\gamma = 0.04$ the learned neural network is still significantly smoother than the network learned for the VPSDE when a mixed score parameterization is used.[11]

**Challenging Toy Dataset.** Using the same simple ResNet architecture (less than 100k parameters) from the above experiment, we trained a VPSDE-based as well as a CLD-based SGM to maximize the likelihood of a more challenging toy dataset (the dataset is essentially "multi-scale", as it involves both large scale—the placement of the swiss rolls—and fine scale—the swiss rolls themselves—structure). Similar to the other toy datasets, the models are trained for 1M iterations using fresh data synthesized from the data distribution in each batch at a batch size of 512.

In Fig. 13, we compare samples of the models to the data distribution. Even with our simple model architecture, CLD is able to capture the multi-scale structure of the dataset: the five rolls are adequately resembled and only a few samples are in between modes. VPSDE, on the other hand, only captures the main modes, but not the fine structure. Furthermore, VPSDE has the undesired behavior of "connecting" the modes.

Overall, we conclude that also in the maximum likelihood training setting CLD is a promising diffusion showing superior behavior compared to the VPSDE in our toy experiments.

---

[10]For the ML objective of the VPSDE, we refer the reader to Song et al. (2021b).

[11]The VPSDE-based model with mixed score parameterization did not converge to the target distribution, and therefore is not included in Fig. 12.

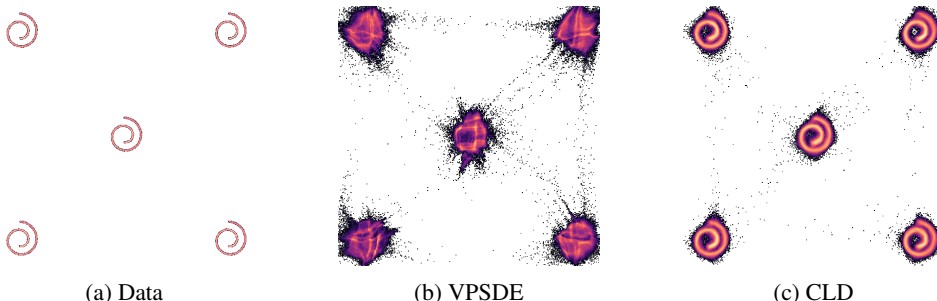

|        (a) Data        |        (b) VPSDE        |        (c) CLD        |

Figure 13: Data distribution and model samples for multi-scale toy experiment.

Table 9: Performance using non-adaptive step size solvers. Extended version of Tab. 3.

| Model | Sampler | FID at $n$ function evaluations ↓ | | | | | |
|---|---|---|---|---|---|---|---|
| | | $n$=50 | $n$=150 | $n$=275 | $n$=500 | $n$=1000 | $n$=2000 |
| CLD | EM | 143 | 31.5 | 10.9 | 3.96 | 2.50 | 2.27 |
| CLD | EM-QS | 52.7 | 7.00 | 3.24 | 2.41 | **2.27** | **2.23** |
| CLD | SSCS | 81.1 | 10.5 | 2.86 | 2.30 | 2.32 | 2.29 |
| CLD | SSCS-QS | 20.5 | **3.07** | **2.38** | **2.25** | 2.30 | 2.29 |
| VPSDE | EM | 92.0 | 30.3 | 13.1 | 4.42 | 2.46 | 2.43 |
| VPSDE | EM-QS | 28.2 | 4.06 | 2.65 | 2.47 | 2.66 | 2.60 |
| VPSDE | DDIM | 6.04 | 4.04 | 3.53 | 3.26 | 3.09 | 3.01 |
| VPSDE | DDIM-QS | **3.78** | 3.15 | 3.05 | 2.99 | 2.96 | 2.95 |
| VESDE | PC | 460 | 216 | 11.2 | 3.75 | 2.43 | **2.23** |
| VESDE | PC-QS | 461 | 388 | 155 | 5.47 | 11.4 | 11.2 |

## F.2    CIFAR-10 — EXTENDED RESULTS

In this section, we provide additional results on the CIFAR-10 image modeling benchmark.

An extended version of Tab. 3 (sampling the generative SDE with different fixed-step size solvers for different compute budgets) including additional baselines can be found in Tab. 9. Note that time stepping with quadratic striding (QS) improves sampling from VPSDE- and CLD-based models for all settings except for the combination of VPSDE and EM sampling in the setting $n = \{1000, 2000\}$. For the VESDE (using PC sampling), QS significantly worsens FID scores. The reason for this could be that the variance of the VESDE already follows an exponential schedule (see Fig. 5 in Song et al. (2021c)). We additionally present results for the VPSDE using the DDIM (Denoising Diffusion Implicit Models) sampler (Song et al., 2021a). As was observed by Song et al. (2021a), QS also helps for DDIM. Importantly, for any $n \geq 150$, our CLD with our novel SSCS (and QS) even outperforms DDIM. Only for $n = 50$, DDIM performs better. It needs to be mentioned, however, that the DDIM sampler was specifically designed for few-step sampling, whereas our CLD with SSCS is derived in a general fashion without this particular regime in mind. In particular, DDIM sampling can be interpreted as a non-Markovian sampling method and it is not clear how to calculate the log-likelihood of hold-out validation data under this non-Markovian synthesis approach. Nevertheless, it would be interesting to also explore non-Markovian DDIM-inspired techniques for CLD-SGMs to further improve sampling speed in CLD-SGMs.

Note that our DDIM results shown in Tab. 9 are better than those presented in Song et al. (2021a) itself, because we are relying on the DDPM++ model trained in Song et al. (2021c), whereas Song et al. (2021a) uses the DDPM model from Ho et al. (2020).

Finally, we present additional generated samples from our CLD-SGM model: see Fig. 14 and Fig. 15 for samples from EM-QS with 2000 evaluations and SSCS-QS with 150 evaluations, respectively.

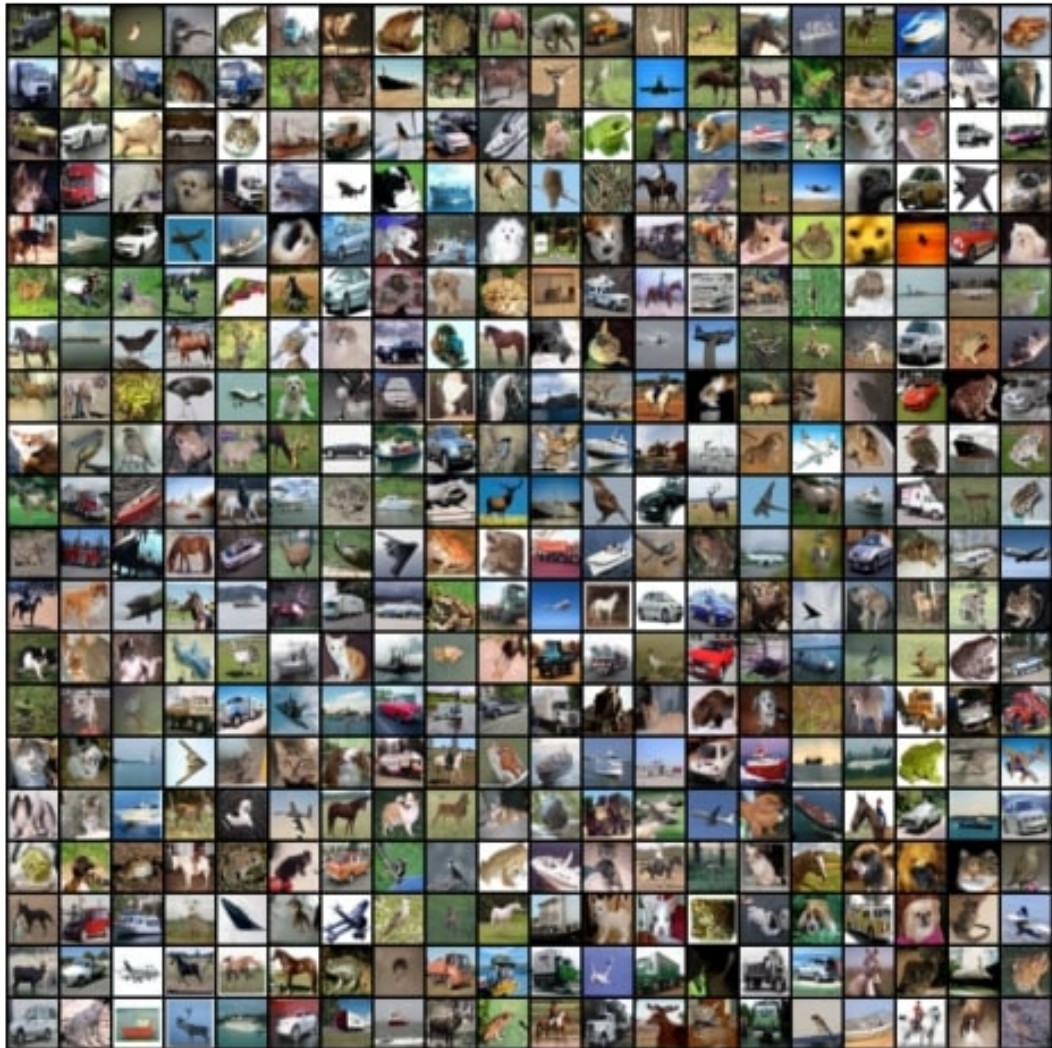

Figure 14: Additional samples using EM-QS with 2000 function evaluations. This setup gave us our best FID score of 2.23.

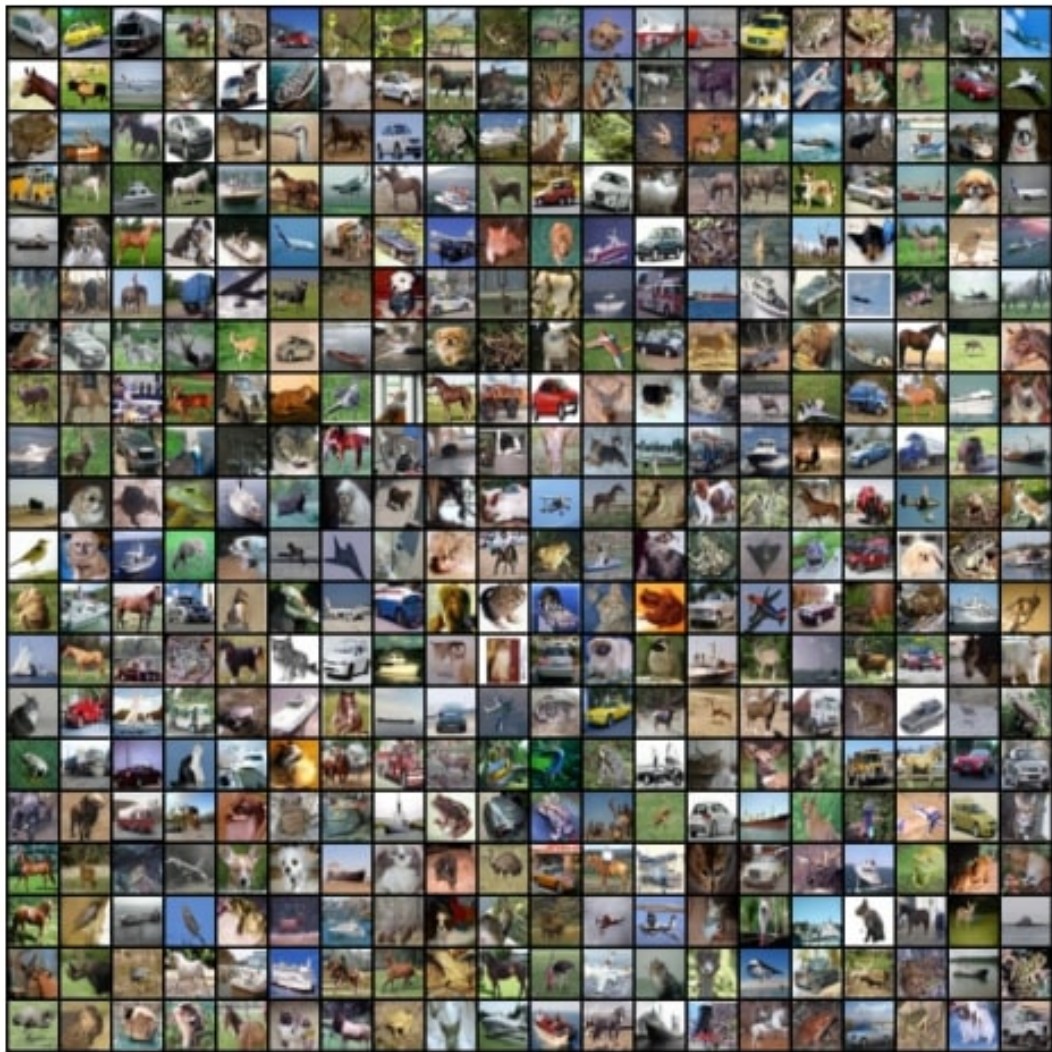

Figure 15: Additional samples using SSCS-QS. This setup resulted in an FID score of 3.07 using only 150 function evaluations.

### F.3 CELEBA-HQ-256 — EXTENDED RESULTS

In this section, we provide additional qualitative results on CelebA-HQ-256. For high quality samples using our new SSCS solver see Fig. 16.

Samples generated with an adaptive step size Runge–Kutta 4(5) solver at different solver tolerances can be found in Fig. 17. We found that our model still generates very good samples even for a solver error tolerance of $10^{-3}$ using an average of 129 neural network evaluations.

Lastly, we show "generation paths" of samples from our CelebA-HQ-256 model: see Fig. 18 and Fig. 19 for samples from the probability flow ODE and the generative SDE, respectively. We visualize the continuous generation paths via snapshots of data and velocity variables at eight different time steps. Interestingly, we can see that the velocity variables "encode" the data at intermediate $t$. On the other hand, at time $t = 1.0$, by construction, both data and velocity are distributed according to the "equilibrium distribution" of the diffusion, namely, $p_{\text{EQ}}(\mathbf{u}) = \mathcal{N}(\mathbf{x}; \mathbf{0}_d, \mathbf{I}_d)\,\mathcal{N}(\mathbf{v}; \mathbf{0}_d, M\mathbf{I}_d)$. Furthermore, as $t \to 0$ the data variables approximately converge to the data distribution, while the velocity variables approximately converge to another Normal distribution $\mathcal{N}(\mathbf{v}; \mathbf{0}_d, \gamma M \mathbf{I}_d)$ (with $\gamma = 0.04$ in our experiments).

Recall that for CLD, the neural network approximates the score $\nabla_{\mathbf{v}_t} \log p_t(\mathbf{v}_t|\mathbf{x}_t)$. We believe that the generation paths are further evidence that CLD-SGMs need to learn simpler models: for fixed $t$ the velocity variable $\mathbf{v}_t$ appears to be a "noisy" version of the data $\mathbf{x}_t$, and therefore we believe $p_t(\mathbf{v}_t|\mathbf{x}_t)$ to be relatively smooth and simple when compared to the marginal $p_t(\mathbf{x}_t)$.

Finally, note that in Figs. 18 and 19, when visualizing the velocity variables, we used a colorization scheme that corresponds exactly to the inverse of the color scheme used for visualizing the images themselves. Alternatively, we could also interpret this in such a way that we are not actually visualizing velocities, but negative velocities with flipped signs. When using this inverse colorization scheme for the velocities, we see that at intermediate $t$, where the velocities encode the data, the color values visualizing image data and velocities are, apart from the additional noise in the velocities, similar (i.e. the velocities appear as noisy versions of the actual images). This implies that image pixel values $\mathbf{x}_t$ translate into corresponding *negative* velocities $\mathbf{v}_t$ that pull the pixel values back towards the mean of the equilibrium distribution. This is a consequence of the Hamiltonian coupling between the data and velocity variables. In other words, it is a result of the negative sign in front of $\mathbf{x}_t$ in the $H$ term in Eq. (5) (and analogously for the reverse-time generative SDE). Also see the visualizations on our project page (https://nv-tlabs.github.io/CLD-SGM).

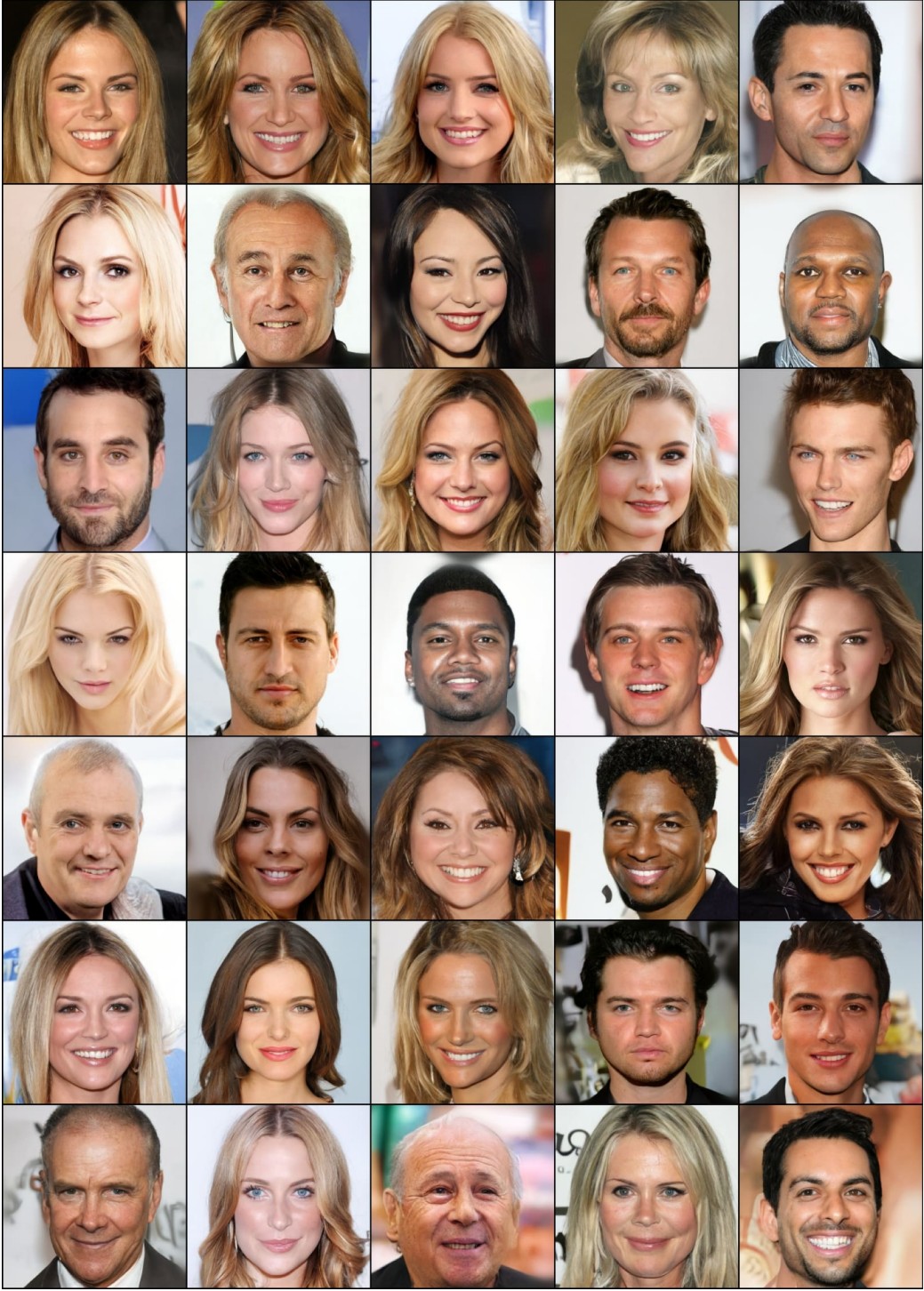

Figure 16: Samples generated by our model on the CelebA-HQ-256 dataset using our SSCS solver.

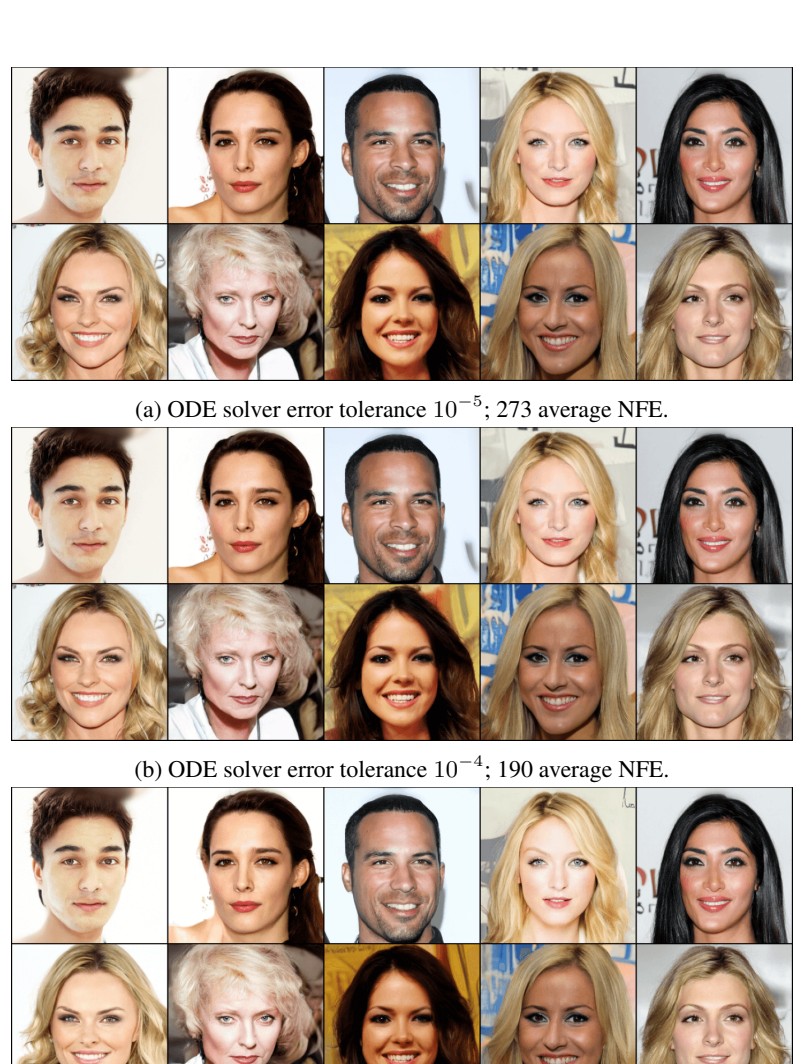

(a) ODE solver error tolerance $10^{-5}$; 273 average NFE.

(b) ODE solver error tolerance $10^{-4}$; 190 average NFE.

(c) ODE solver error tolerance $10^{-3}$; 129 average NFE.

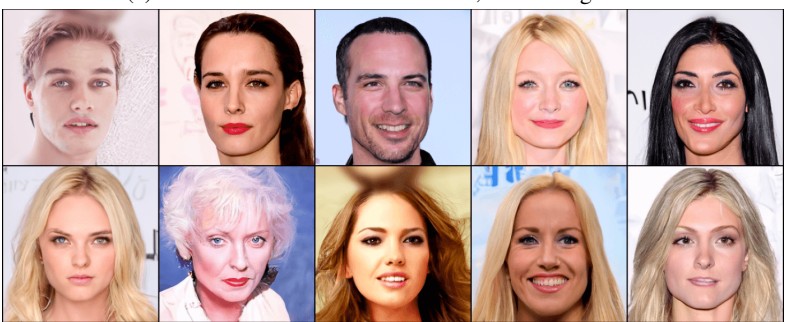

(d) ODE solver error tolerance $10^{-2}$; 99.4 average NFE.

Figure 17: Samples generated by our model on the CelebA-HQ-256 dataset using a Runge–Kutta 4(5) adaptive ODE solver to solve the probability flow ODE. We show the effect of the ODE solver error tolerance on the quality of samples ((a), (b), (c) and (d) were generated using the same prior samples). Little visual differences can be seen between $10^{-5}$ and $10^{-4}$. Low frequency artifacts can be observed at $10^{-3}$. Deterioration starts to set in at $10^{-2}$.

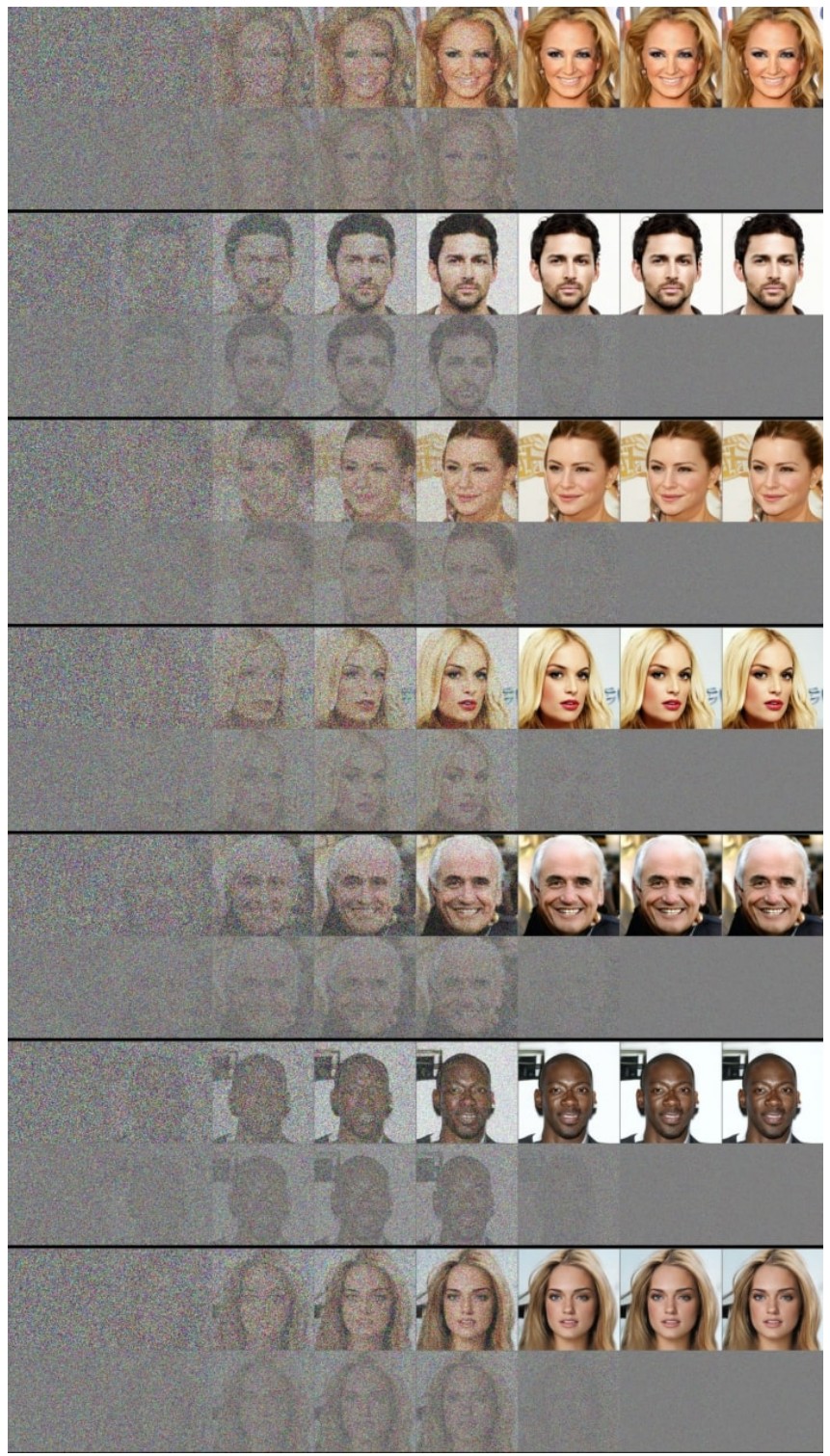

Figure 18: Generation paths of samples from our CelebA-HQ-256 model (Runge–Kutta 4(5) solver; mean NFE: 288). Odd and even rows visualize data and velocity variables, respectively. The eight columns correspond to times $t \in \{1.0, 0.5, 0.3, 0.2, 0.1, 10^{-2}, 10^{-3}, 10^{-5}\}$ (from left to right). The velocity distribution converges to a Normal (different variances) for both $t \to 0$ and $t \to 1$. See App. F.3 for visualization details and discussion.

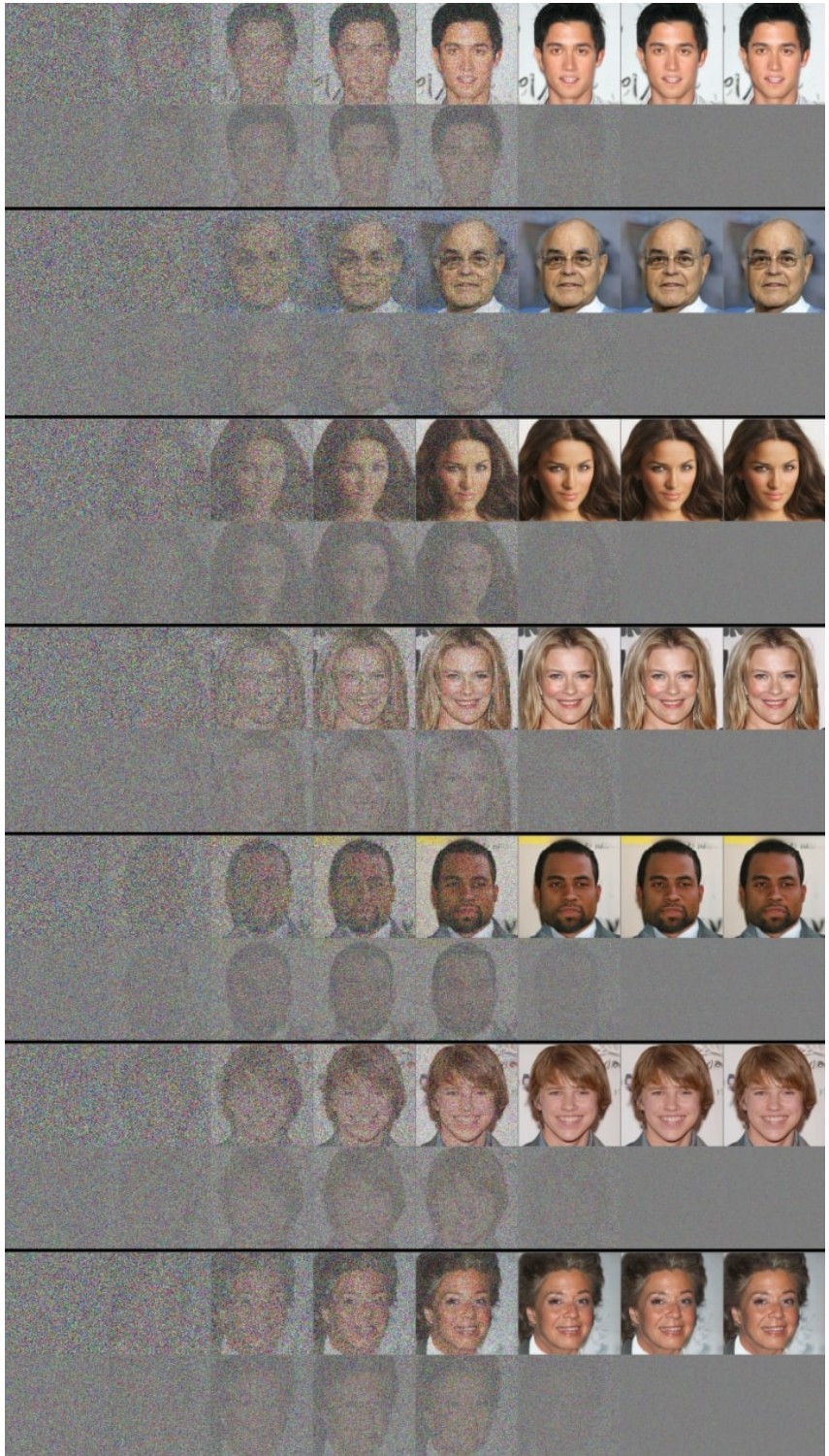

Figure 19: Generation paths of samples from our CelebA-HQ-256 model (SSCS-QS using only 150 steps). Odd and even rows visualize data and velocity variables, respectively. The eight columns correspond to times $t \in \{1.0, 0.5, 0.3, 0.2, 0.1, 10^{-2}, 10^{-3}, 10^{-5}\}$ (from left to right). The velocity distribution converges to a Normal (different variances) for both $t \to 0$ and $t \to 1$. See App. F.3 for visualization details and discussion.

## G   PROOFS OF PERTURBATION KERNELS

In this section, we prove the correctness of the perturbation kernels of the forward diffusion (App. B.1) as well as for the analytical splitting term in our SSCS (App. D.2). All derivations are presented for general time-dependent $\beta(t)$.

### G.1   FORWARD DIFFUSION

We have the following ODEs describing the evolution of the mean and the covariance matrix

$$\frac{d\boldsymbol{\mu}_t}{dt} = (f(t) \otimes \boldsymbol{I}_d)\boldsymbol{\mu}_t, \tag{110}$$

$$\frac{d\boldsymbol{\Sigma}_t}{dt} = (f(t) \otimes \boldsymbol{I}_d)\boldsymbol{\Sigma}_t + [(f(t) \otimes \boldsymbol{I}_d)\boldsymbol{\Sigma}_t]^\top + \left(G(t)G^\top(t)\right) \otimes \boldsymbol{I}_d, \tag{111}$$

where

$$f(t) := \begin{pmatrix} 0 & 4\beta(t)\Gamma^{-2} \\ -\beta(t) & -4\beta(t)\Gamma^{-1} \end{pmatrix}, \tag{112}$$

$$G(t) := \begin{pmatrix} 0 & 0 \\ 0 & \sqrt{2\Gamma\beta(t)} \end{pmatrix}. \tag{113}$$

In App. B.1, we claim the following solutions:

$$\boldsymbol{\mu}_t := C_t \hat{\boldsymbol{\mu}}_t, \tag{114}$$

$$\hat{\boldsymbol{\mu}}_t := \begin{pmatrix} \boldsymbol{\mu}_t^x \\ \boldsymbol{\mu}_t^v \end{pmatrix}, \tag{115}$$

$$C_t := e^{-2\mathcal{B}(t)\Gamma^{-1}}, \tag{116}$$

$$\boldsymbol{\mu}_t^x := 2\mathcal{B}(t)\Gamma^{-1}\mathbf{x}_0 + 4\mathcal{B}(t)\Gamma^{-2}\mathbf{v}_0 + \mathbf{x}_0, \tag{117}$$

$$\boldsymbol{\mu}_t^v := -\mathcal{B}(t)\mathbf{x}_0 - 2\mathcal{B}(t)\Gamma^{-1}\mathbf{v}_0 + \mathbf{v}_0, \tag{118}$$

and

$$\boldsymbol{\Sigma}_t := \Sigma_t \otimes \boldsymbol{I}_d, \tag{119}$$

$$\Sigma_t := D_t \hat{\Sigma}_t, \tag{120}$$

$$\hat{\Sigma}_t := \begin{pmatrix} \Sigma_t^{xx} & \Sigma_t^{xv} \\ \Sigma_t^{xv} & \Sigma_t^{vv} \end{pmatrix}, \tag{121}$$

$$D_t := e^{-4\mathcal{B}(t)\Gamma^{-1}}, \tag{122}$$

$$\Sigma_t^{xx} := \Sigma_0^{xx} + e^{4\mathcal{B}(t)\Gamma^{-1}} - 1 + 4\mathcal{B}(t)\Gamma^{-1}\left(\Sigma_0^{xx} - 1\right) + 4\mathcal{B}^2(t)\Gamma^{-2}\left(\Sigma_0^{xx} - 2\right) + 16\mathcal{B}^2(t)\Gamma^{-4}\Sigma_0^{vv}, \tag{123}$$

$$\Sigma_t^{xv} := -\mathcal{B}(t)\Sigma_0^{xx} + 4\mathcal{B}(t)\Gamma^{-2}\Sigma_0^{vv} - 2\mathcal{B}^2(t)\Gamma^{-1}\left(\Sigma_0^{xx} - 2\right) - 8\mathcal{B}^2(t)\Gamma^{-3}\Sigma_0^{vv}, \tag{124}$$

$$\Sigma_t^{vv} := \tfrac{\Gamma^2}{4}\left(e^{4\mathcal{B}(t)\Gamma^{-1}} - 1\right) + \mathcal{B}(t)\Gamma + \Sigma_0^{vv}\left(1 + 4\mathcal{B}^2(t)\Gamma^{-2} - 4\mathcal{B}(t)\Gamma^{-1}\right) + \mathcal{B}^2(t)\left(\Sigma_0^{xx} - 2\right), \tag{125}$$

where $\mathcal{B}(t) = \int_0^t \beta(\hat{t})\,d\hat{t}$ and $\boldsymbol{\mu}_0 = [\mathbf{x}_0, \mathbf{v}_0]^\top$ as well as $\Sigma_0^{xx}$ and $\Sigma_0^{vv}$ are initial conditions.

### G.1.1   PROOF OF CORRECTNESS OF THE MEAN

Plugging the claimed solution (Eqs. (114)-(118)) back into the ODE (Eq. 110), we obtain

$$\hat{\boldsymbol{\mu}}_t\frac{dC_t}{dt} + \frac{d\hat{\boldsymbol{\mu}}_t}{dt}C_t = C_t(f(t) \otimes \boldsymbol{I}_d)\hat{\boldsymbol{\mu}}_t. \tag{126}$$

The above can be decomposed into two equations:

$$-2\beta(t)\Gamma^{-1}\boldsymbol{\mu}_t^x + \frac{d\boldsymbol{\mu}_t^x}{dt} = 4\beta(t)\Gamma^{-2}\boldsymbol{\mu}_t^v, \tag{127}$$

$$-2\beta(t)\Gamma^{-1}\boldsymbol{\mu}_t^v + \frac{d\boldsymbol{\mu}_t^v}{dt} = -\beta(t)\boldsymbol{\mu}_t^x - 4\beta(t)\Gamma^{-1}\boldsymbol{\mu}_t^v, \tag{128}$$

where we used the fact that $\frac{dC_t}{dt} = -2\beta(t)\Gamma^{-1}C_t$.

**Eq. (127):** Plugging the claimed solution into Eq. (127), we obtain:

$$-2\beta(t)\Gamma^{-1}\left[2\mathcal{B}(t)\Gamma^{-1}\mathbf{x}_0 + 4\mathcal{B}(t)\Gamma^{-2}\mathbf{v}_0 + \mathbf{x}_0\right] + \left[2\beta(t)\Gamma^{-1}\mathbf{x}_0 + 4\beta(t)\Gamma^{-2}\mathbf{v}_0\right] = 4\beta(t)\Gamma^{-2}\left[-\mathcal{B}(t)\mathbf{x}_0 - 2\mathcal{B}(t)\Gamma^{-1}\mathbf{v}_0 + \mathbf{v}_0\right].$$
(129)

**Eq. (128):** After simplification, plugging in the claimed solution into Eq. (128), we obtain:

$$2\beta(t)\Gamma^{-1}\left[-\mathcal{B}(t)\mathbf{x}_0 - 2\mathcal{B}(t)\Gamma^{-1}\mathbf{v}_0 + \mathbf{v}_0\right] + \left[-\beta(t)\mathbf{x}_0 - 2\beta(t)\Gamma^{-1}\mathbf{v}_0\right] = -\beta(t)\left[2\mathcal{B}(t)\Gamma^{-1}\mathbf{x}_0 + 4\mathcal{B}(t)\Gamma^{-2}\mathbf{v}_0 + \mathbf{x}_0\right].$$
(130)

This completes the proof of the correctness of the mean.

### G.1.2 PROOF OF CORRECTNESS OF THE COVARIANCE

Plugging the claimed solution (Eqs. (119)-(125)) back in the ODE (Eq. (111)), we obtain

$$\left[\frac{d\hat{\Sigma}_t}{dt}D_t + \frac{dD_t}{dt}\Sigma_t\right] \otimes \mathbf{I}_d = D_t(f(t) \otimes \mathbf{I}_d)(\hat{\Sigma}_t \otimes \mathbf{I}_d) + D_t\left[(f(t) \otimes \mathbf{I}_d)(\hat{\Sigma}_t \otimes \mathbf{I}_d)\right]^\top + \left(G(t)G^\top(t)\right) \otimes \mathbf{I}_d.$$
(131)

Noting that

$$(f(t) \otimes \mathbf{I}_d)(\hat{\Sigma}_t \otimes \mathbf{I}_d) = (f(t)\hat{\Sigma}_t) \otimes \mathbf{I}_d$$

$$= \beta(t)\begin{pmatrix} 4\Gamma^{-2}\Sigma_t^{xv} & 4\Gamma^{-2}\Sigma_t^{vv} \\ -\Sigma_t^{xx} - 4\Gamma^{-1}\Sigma_t^{xv} & -\Sigma_t^{xv} - 4\Gamma^{-1}\Sigma^{vv} \end{pmatrix} \otimes \mathbf{I}_d,$$
(132)

and

$$G(t)G^\top(t) = \begin{pmatrix} 0 & 0 \\ 0 & 2\Gamma\beta(t) \end{pmatrix},$$
(133)

as well as the fact that $\frac{dD_t}{dt} = -4\beta(t)\Gamma^{-1}D_t$, we can decompose Eq. (131) into three equations:

$$-4\beta(t)\Gamma^{-1}\Sigma_t^{xx} + \frac{d\Sigma^{xx}}{dt} = 8\beta(t)\Gamma^{-2}\Sigma_t^{xv},$$
(134)

$$-4\beta(t)\Gamma^{-1}\Sigma_t^{xv} + \frac{d\Sigma^{xv}}{dt} = \beta(t)\left[-\Sigma_t^{xx} - 4\Gamma^{-1}\Sigma_t^{xv} + 4\Gamma^{-2}\Sigma_t^{vv}\right],$$
(135)

$$-4\beta(t)\Gamma^{-1}\Sigma_t^{vv} + \frac{d\Sigma^{vv}}{dt} = \beta(t)\left[-2\Sigma_t^{xv} - 8\Gamma^{-1}\Sigma_t^{vv}\right] + 2\Gamma\beta(t)D_t^{-1}.$$
(136)

**Eq. (134):** Plugging the claimed solution into Eq. (134), we obtain

$$-4\beta(t)\Gamma^{-1}\left[\Sigma_0^{xx} + e^{4\mathcal{B}(t)\Gamma^{-1}} - 1 + 4\mathcal{B}(t)\Gamma^{-1}(\Sigma_0^{xx} - 1) + 4\mathcal{B}^2(t)\Gamma^{-2}(\Sigma_0^{xx} - 2) + 16\mathcal{B}^2(t)\Gamma^{-4}\Sigma_0^{vv}\right]$$

$$+ \left[4\beta(t)\Gamma^{-1}e^{4\mathcal{B}(t)\Gamma^{-1}} + 4\beta(t)\Gamma^{-1}(\Sigma_0^{xx} - 1) + 8\beta(t)\mathcal{B}(t)\Gamma^{-2}(\Sigma_0^{xx} - 2) + 32\beta(t)\mathcal{B}(t)\Gamma^{-4}\Sigma_0^{vv}\right]$$

$$= 8\beta(t)\Gamma^{-2}\left[-\mathcal{B}(t)\Sigma_0^{xx} + 4\mathcal{B}(t)\Gamma^{-2}\Sigma_0^{vv} - 2\mathcal{B}^2(t)\Gamma^{-1}(\Sigma_0^{xx} - 2) - 8\mathcal{B}^2(t)\Gamma^{-3}\Sigma_0^{vv}\right].$$
(137)

**Eq. (135):** After simplification, plugging the claimed solution into Eq. (135), we obtain

$$\left[-\beta(t)\Sigma_0^{xx} + 4\beta(t)\Gamma^{-2}\Sigma_0^{vv} - 4\beta(t)\mathcal{B}(t)\Gamma^{-1}(\Sigma_0^{xx} - 2) - 16\beta(t)\mathcal{B}(t)\Gamma^{-3}\Sigma_0^{vv}\right]$$

$$= -\beta(t)\left[\Sigma_0^{xx} + e^{4\mathcal{B}(t)\Gamma^{-1}} - 1 + 4\mathcal{B}(t)\Gamma^{-1}(\Sigma_0^{xx} - 1) + 4\mathcal{B}^2(t)\Gamma^{-2}(\Sigma_0^{xx} - 2) + 16\mathcal{B}^2(t)\Gamma^{-4}\Sigma_0^{vv}\right]$$

$$+ 4\beta(t)\Gamma^{-2}\left[\frac{\Gamma^2}{4}\left(e^{4\mathcal{B}(t)\Gamma^{-1}} - 1\right) + \mathcal{B}(t)\Gamma + \Sigma_0^{vv}\left(1 + 4\mathcal{B}^2(t)\Gamma^{-2} - 4\mathcal{B}(t)\Gamma^{-1}\right) + \mathcal{B}^2(t)(\Sigma_0^{xx} - 2)\right].$$
(138)

**Eq. (136):** After simplification, plugging the claimed solution into Eq. (136), we obtain

$$
4\beta(t)\Gamma^{-1}\left[\tfrac{\Gamma^2}{4}\left(e^{4\mathcal{B}(t)\Gamma^{-1}}-1\right)+\mathcal{B}(t)\Gamma+\Sigma_0^{vv}\left(1+4\mathcal{B}^2(t)\Gamma^{-2}-4\mathcal{B}(t)\Gamma^{-1}\right)+\mathcal{B}^2(t)\left(\Sigma_0^{xx}-2\right)\right]
$$

$$
+\left[\tfrac{\Gamma^2}{4}4\beta(t)\Gamma^{-1}e^{4\mathcal{B}(t)\Gamma^{-1}}+\beta(t)\Gamma+\Sigma_0^{vv}\left(8\mathcal{B}(t)\beta(t)\Gamma^{-2}-4\beta(t)\Gamma^{-1}\right)+2\beta(t)\mathcal{B}(t)\left(\Sigma_0^{xx}-2\right)\right]
$$

$$
=-2\beta(t)\left[-\mathcal{B}(t)\Sigma_0^{xx}+4\mathcal{B}(t)\Gamma^{-2}\Sigma_0^{vv}-2\mathcal{B}^2(t)\Gamma^{-1}\left(\Sigma_0^{xx}-2\right)-8\mathcal{B}^2(t)\Gamma^{-3}\Sigma_0^{vv}\right]
$$

$$
+2\Gamma\beta(t)e^{4\mathcal{B}(t)\Gamma^{-1}}.
$$

(139)

This completes the proof of the correctness of the covariance.

## G.2 ANALYTICAL SPLITTING TERM OF SSCS

We have the following ODEs describing the evolution of the mean and the covariance matrix

$$
\frac{d\bar{\boldsymbol{\mu}}_t}{dt}=(f(T-t)\otimes\boldsymbol{I}_d)\bar{\boldsymbol{\mu}}_t, \tag{140}
$$

$$
\frac{d\bar{\boldsymbol{\Sigma}}_t}{dt}=(f(T-t)\otimes\boldsymbol{I}_d)\bar{\boldsymbol{\Sigma}}_t+\left[(f(T-t)\otimes\boldsymbol{I}_d)\bar{\boldsymbol{\Sigma}}_t\right]^\top+\left(G(T-t)G^\top(T-t)\right)\otimes\boldsymbol{I}_d, \tag{141}
$$

where

$$
f(T-t):=\begin{pmatrix} 0 & -4\beta(T-t)\Gamma^{-2} \\ +\beta(T-t) & -4\beta(T-t)\Gamma^{-1} \end{pmatrix}, \tag{142}
$$

$$
G(T-t):=\begin{pmatrix} 0 & 0 \\ 0 & \sqrt{2\Gamma\beta(T-t)} \end{pmatrix}. \tag{143}
$$

These ODEs are very similar to the ODEs of the forward diffusion in App. G.1, the only difference being flipped signs in the off-diagonal terms of $f(T-t)$ (highlighted in red).

In App. D.2, we claim the following solutions

$$
\bar{\boldsymbol{\mu}}_t=C_t\tilde{\boldsymbol{\mu}}_t, \tag{144}
$$

$$
\tilde{\boldsymbol{\mu}}_t=\begin{pmatrix} \bar{\boldsymbol{\mu}}_t^x \\ \bar{\boldsymbol{\mu}}_t^v \end{pmatrix}, \tag{145}
$$

$$
C_t=e^{-2\mathcal{B}(t)\Gamma^{-1}}, \tag{146}
$$

$$
\bar{\boldsymbol{\mu}}_t^x=2\mathcal{B}(t)\Gamma^{-1}\bar{\mathbf{x}}_{t'}-4\mathcal{B}(t)\Gamma^{-2}\bar{\mathbf{v}}_{t'}+\bar{\mathbf{x}}_{t'}, \tag{147}
$$

$$
\bar{\boldsymbol{\mu}}_t^v=+\mathcal{B}(t)\bar{\mathbf{x}}_{t'}-2\mathcal{B}(t)\Gamma^{-1}\bar{\mathbf{v}}_{t'}+\bar{\mathbf{v}}_{t'}, \tag{148}
$$

and

$$
\bar{\boldsymbol{\Sigma}}_t=\bar{\Sigma}_t\otimes\boldsymbol{I}_d, \tag{149}
$$

$$
\bar{\Sigma}_t=D_t\tilde{\Sigma}_t, \tag{150}
$$

$$
\tilde{\Sigma}_t=\begin{pmatrix} \bar{\Sigma}_t^{xx} & \bar{\Sigma}_t^{xv} \\ \bar{\Sigma}_t^{xv} & \bar{\Sigma}_t^{vv} \end{pmatrix}, \tag{151}
$$

$$
D_t=e^{-4\mathcal{B}(t)\Gamma^{-1}}, \tag{152}
$$

$$
\bar{\Sigma}_t^{xx}=e^{4\mathcal{B}(t)\Gamma^{-1}}-1-4\mathcal{B}(t)\Gamma^{-1}-8\mathcal{B}^2(t)\Gamma^{-2}, \tag{153}
$$

$$
\bar{\Sigma}_t^{xv}=-4\mathcal{B}^2(t)\Gamma^{-1}, \tag{154}
$$

$$
\bar{\Sigma}_t^{vv}=\tfrac{\Gamma^2}{4}\left(e^{4\mathcal{B}(t)\Gamma^{-1}}-1\right)+\mathcal{B}(t)\Gamma-2\mathcal{B}^2(t), \tag{155}
$$

where $\mathcal{B}(t)=\int_{t'}^t\beta(T-\hat{t})\,d\hat{t}$ and $\bar{\boldsymbol{\mu}}_{t'}=[\bar{\mathbf{x}}_{t'},\bar{\mathbf{v}}_{t'}]^\top$ is an initial condition. Differences of the above solution to the solutions of the forward diffusion are again highlighted in red. Note that by construction the initial covariance for the analytical splitting term of SSCS is the zero matrix, i.e., $\bar{\Sigma}_{t'}^{xx}=\bar{\Sigma}_{t'}^{xv}=\bar{\Sigma}_{t'}^{vv}=0$, since we always initialize from an "updated sample", which itself does not have any uncertainty. Also note that in this derivation we use general initial $t'$ (whereas in App. G.1 we set $t'=0$ for simplicity).

### G.2.1 PROOF OF CORRECTNESS OF THE MEAN

Plugging the claimed solution (Eqs. (144)-(148)) into the ODE (Eq. (140)), we obtain

$$\tilde{\boldsymbol{\mu}}_t \frac{dC_t}{dt} + \frac{d\tilde{\boldsymbol{\mu}}_t}{dt} C_t = C_t(f(T-t) \otimes \boldsymbol{I}_d)\tilde{\boldsymbol{\mu}}_t. \tag{156}$$

The above can be decomposed into two equations:

$$-2\beta(T-t)\Gamma^{-1}\bar{\boldsymbol{\mu}}_t^x + \frac{d\bar{\boldsymbol{\mu}}_t^x}{dt} = -4\beta(T-t)\Gamma^{-2}\bar{\boldsymbol{\mu}}_t^v, \tag{157}$$

$$-2\beta(T-t)\Gamma^{-1}\bar{\boldsymbol{\mu}}_t^v + \frac{d\bar{\boldsymbol{\mu}}_t^v}{dt} = \beta(T-t)\bar{\boldsymbol{\mu}}_t^x - 4\beta(T-t)\Gamma^{-1}\bar{\boldsymbol{\mu}}_t^v, \tag{158}$$

where we used the fact that $\frac{dC_t}{dt} = -2\beta(T-t)\Gamma^{-1}C_t$.

**Eq. (157):** Plugging the claimed solution into Eq. (157), we obtain:

$$- 2\beta(T-t)\Gamma^{-1} \left[ 2\mathcal{B}(t)\Gamma^{-1}\bar{\mathbf{x}}_{t'} - 4\mathcal{B}(t)\Gamma^{-2}\bar{\mathbf{v}}_{t'} + \bar{\mathbf{x}}_{t'} \right] + \left[ 2\beta(T-t)\Gamma^{-1}\bar{\mathbf{x}}_{t'} - 4\beta(T-t)\Gamma^{-2}\bar{\mathbf{v}}_{t'} \right]$$
$$= -4\beta(T-t)\Gamma^{-2} \left[ \mathcal{B}(t)\bar{\mathbf{x}}_{t'} - 2\mathcal{B}(t)\Gamma^{-1}\bar{\mathbf{v}}_{t'} + \bar{\mathbf{v}}_{t'} \right]. \tag{159}$$

**Eq. (128):** After simplification, plugging the claimed solution into Eq. (128), we obtain:

$$2\beta(T-t)\Gamma^{-1} \left[ \mathcal{B}(t)\bar{\mathbf{x}}_{t'} - 2\mathcal{B}(t)\Gamma^{-1}\bar{\mathbf{v}}_{t'} + \bar{\mathbf{v}}_{t'} \right] + \left[ \beta(T-t)\bar{\mathbf{x}}_{t'} - 2\beta(T-t)\Gamma^{-1}\bar{\mathbf{v}}_{t'} \right]$$
$$= \beta(T-t) \left[ 2\mathcal{B}(t)\Gamma^{-1}\bar{\mathbf{x}}_{t'} - 4\mathcal{B}(t)\Gamma^{-2}\bar{\mathbf{v}}_{t'} + \bar{\mathbf{x}}_{t'} \right]. \tag{160}$$

This completes the proof of the correctness of the mean.

### G.2.2 PROOF OF CORRECTNESS OF THE COVARIANCE

Plugging the claimed solution (Eqs. (149)-(155)) into the ODE (Eq. (141)), we obtain

$$\left[ \frac{d\tilde{\Sigma}_t}{dt}D_t + \frac{dD_t}{dt}\tilde{\Sigma}_t \right] \otimes \boldsymbol{I}_d = D_t(f \otimes \boldsymbol{I}_d)(\tilde{\Sigma}_t \otimes \boldsymbol{I}_d) + D_t \left[ (f \otimes \boldsymbol{I}_d)(\tilde{\Sigma}_t \otimes \boldsymbol{I}_d) \right]^\top + \left( GG^\top \right) \otimes \boldsymbol{I}_d \tag{161}$$

with $f = f(T-t)$ and $G = G(T-t)$.

Noting that

$$(f(T-t) \otimes \boldsymbol{I}_d)(\tilde{\Sigma}_t \otimes \boldsymbol{I}_d) = (f(T-t)\tilde{\Sigma}_t) \otimes \boldsymbol{I}_d$$
$$= \beta(T-t) \begin{pmatrix} -4\Gamma^{-2}\bar{\Sigma}_t^{xv} & -4\Gamma^{-2}\bar{\Sigma}_t^{vv} \\ \bar{\Sigma}_t^{xx} - 4\Gamma^{-1}\bar{\Sigma}_t^{xv} & \bar{\Sigma}_t^{xv} - 4\Gamma^{-1}\bar{\Sigma}^{vv} \end{pmatrix} \otimes \boldsymbol{I}_d, \tag{162}$$

and

$$G(T-t)G^\top(T-t) = \begin{pmatrix} 0 & 0 \\ 0 & 2\Gamma\beta(T-t) \end{pmatrix}, \tag{163}$$

as well as the fact $\frac{dD_t}{dt} = -4\beta(T-t)\Gamma^{-1}D_t$, we can decompose Eq. (161) into three equations:

$$-4\beta(T-t)\Gamma^{-1}\bar{\Sigma}_t^{xx} + \frac{d\bar{\Sigma}^{xx}}{dt} = -8\beta(T-t)\Gamma^{-2}\bar{\Sigma}_t^{xv}, \tag{164}$$

$$-4\beta(T-t)\Gamma^{-1}\bar{\Sigma}_t^{xv} + \frac{d\bar{\Sigma}^{xv}}{dt} = \beta(T-t) \left[ \bar{\Sigma}_t^{xx} - 4\Gamma^{-1}\bar{\Sigma}_t^{xv} - 4\Gamma^{-2}\bar{\Sigma}^{vv} \right], \tag{165}$$

$$-4\beta(T-t)\Gamma^{-1}\bar{\Sigma}_t^{vv} + \frac{d\bar{\Sigma}^{vv}}{dt} = \beta(T-t) \left[ 2\bar{\Sigma}_t^{xv} - 8\Gamma^{-1}\bar{\Sigma}_t^{vv} \right] + 2\Gamma\beta(T-t)D_t^{-1}. \tag{166}$$

**Eq. (164):** Plugging the claimed solution into Eq. (164), we obtain

$$- 4\beta(T-t)\Gamma^{-1} \left[ e^{4\mathcal{B}(t)\Gamma^{-1}} - 1 - 4\mathcal{B}(t)\Gamma^{-1} - 8\mathcal{B}^2(t)\Gamma^{-2} \right]$$
$$\left[ 4\beta(T-t)\Gamma^{-1}e^{4\mathcal{B}(t)\Gamma^{-1}} - 4\beta(T-t)\Gamma^{-1} - 16\beta(T-t)\mathcal{B}(t)\Gamma^{-2} \right] \tag{167}$$
$$= -8\beta(T-t)\Gamma^{-2} \left[ -4\mathcal{B}^2(t)\Gamma^{-1} \right].$$

**Eq. (165):** After simplification, plugging the claimed solution into Eq. (165), we obtain

$$
\begin{aligned}
-8\beta(T-t)&\mathcal{B}(t)\Gamma^{-1} \\
&= \beta(T-t)\left[e^{4\mathcal{B}(t)\Gamma^{-1}} - 1 - 4\mathcal{B}(t)\Gamma^{-1} - 8\mathcal{B}^2(t)\Gamma^{-2}\right] \\
&- 4\Gamma^{-2}\beta(T-t)\left[\tfrac{\Gamma^2}{4}\left(e^{4\mathcal{B}(t)\Gamma^{-1}} - 1\right) + \mathcal{B}(t)\Gamma - 2\mathcal{B}^2(t)\right].
\end{aligned}
\tag{168}
$$

**Eq. (166):** After simplification, plugging the claimed solution into Eq. (166), we obtain

$$
\begin{aligned}
4\beta(T-t)\Gamma^{-1}&\left[\tfrac{\Gamma^2}{4}\left(e^{4\mathcal{B}(t)\Gamma^{-1}} - 1\right) + \mathcal{B}(t)\Gamma - 2\mathcal{B}^2(t)\right] \\
&\left[\Gamma\beta(T-t)e^{4\mathcal{B}(t)\Gamma^{-1}} + \beta(T-t)\Gamma - 4\beta(T-t)\mathcal{B}(t)\right] \\
&= 2\beta(T-t)\left[-4\mathcal{B}^2(t)\Gamma^{-1}\right] + 2\Gamma\beta(T-t)e^{4\mathcal{B}(t)\Gamma^{-1}}.
\end{aligned}
\tag{169}
$$

This completes the proof of the correctness of the covariance.

To connect back to the SSCS as presented in App. D.2, recall that in practice we use constant $\beta$ (and $T = 1$) and that we solve for small time steps of size $\frac{\delta t}{2}$, such that $\mathcal{B}(t) = \beta\frac{\delta t}{2}$, which leads to the expressions presented in App. D.2.

