# OpenReview forum: "Score-Based Generative Modeling with Critically-Damped Langevin Diffusion"
_ICLR.cc/2022/Conference — ICLR 2022 Spotlight_

### Official Review · Reviewer_F5kX · 2021-10-31

**Correctness:** 4
**Technical Novelty And Significance:** 4
**Empirical Novelty And Significance:** 4
**Recommendation:** 8
**Confidence:** 3

**Main Review:**

Strengths of the paper:
- The paper uses tools from physics to design a novel score based generative model, allowing it to leverage existing insights in that field
- The paper results in training of generative networks that outperform existing approaches when making sure to balance compute budgets and model size
- The paper derive a SDE integrator that allows for efficient sampling from the proposed class of models.


My only question to the authors is: Do they envision that a class of methods of this sort are also possible?  For example, would the method work if they had an acceleration term in which the noise was injected?


**Summary Of The Paper:**

In this paper, the authors introduce a novel approach for training a score-based generative model.  With prior score based networks, generation is performed by solving a stochastic differential equation (SDE) based on Langevin Dynamics, using an estimate of gradient of the log likelihood of the underlying signal distribution.  In the present paper, the authors present a novel forward process, where diffusion is run in a joint data-velocity space.  The noise term is only applied to the velocity component.  This reduces the learning problem to only needing to learn the score of the conditional distribution of velocity given data, which is easier than learning the score of the data distribution directly.  The paper shows that the novel scheme (CLD) yields higher quality for image generation when compared to prior models of similar capacity and number of neural network evaluations.


**Summary Of The Review:**

I recommend the paper be accepted.  It presents a potentially significant improvement to the training of score-based generative models which is well grounded in theory from physics and demonstrates strong empirical performance for sample generation when compared to baselines of similar complexity and compute budgets.

---

> ### Author Response · Authors · 2021-11-13
> **Thank you for questions and feedback**
>
> We thank the reviewer for their positive feedback. We are glad the reviewer appreciates the tools and insights from physics, which we are leveraging.
>
> To answer the reviewer's question on whether other higher-order methods similar to our CLD can be applied: yes, we believe that there is a space for higher-order schemes such as the third-order scheme of [1]. To be specific, using an additional acceleration term with noise injection in that term, as suggested by the reviewer, may work as well, as long as the joint system is set up in such a way that it converges to an analytically tractable joint prior distribution.
>
> Generally, there are potentially many sophisticated and powerful diffusions in augmented spaces that could be promising. Overall, there are many tricks and techniques that are worth exploring in the context of diffusion processes for training score-based generative models (we tried to hint at that in our Related Work section by pointing to the huge literature on thermostats in molecular dynamics, which address closely related equilibration problems). We hope that our work is only the beginning of the exploration of new diffusions for SGMs that take inspiration from statistical mechanics or the MCMC literature. We believe that our work will spark new ideas and result in a fruitful line of follow-up research.
>
> Finally, please note that we added Appendices B.6, C.1, and F.1 to the revised manuscript to address the other reviewers' questions and provide further insights.
>
> References:
>
> [1] Mou et al. 2021. High-Order Langevin Diffusion Yields an Accelerated MCMC Algorithm. Journal of Machine Learning Research.

---

### Official Review · Reviewer_AJW8 · 2021-11-02

**Correctness:** 4
**Technical Novelty And Significance:** 4
**Empirical Novelty And Significance:** 4
**Recommendation:** 10
**Confidence:** 5

**Main Review:**

**Strengths**

- Section 1 and 4 provide a very thorough overview of recent work, with a fantastic visualisation of the method.
- I found the arguments and derivations throughout Section 3 compelling, with the appropriate mathematics reserved for the appendix. The synthetic experiment demonstrating the ease of numerical approximation is also clearly explained and useful.
- Particular praise should be afforded to the consistent use of appropriate references in Section 3, placing the derivations in context, as well as the connections drawn to the high-friction limit in Section A.2.
- Model evaluation is in-line with literature and on-par with the very best performing models. Up to extreme levels of compute, the method is essentially SoTA. Given the higher-order nature of the sampling, evaluation of NFEs is also well-motivated and is a clear improvement for constant compute. Finally, the robust hyperparameters are of practical relevance.

**Weaknesses**

There are very few weaknesses beyond typos and personal preferences for presentation.

My only comment would be the absence of references to the line of work on higher-order MCMC by Michael I Jordan and others which I believe to be pertinent:

[1] Ma, Yi-An, et al. "Is there an analog of Nesterov acceleration for MCMC?." arXiv preprint arXiv:1902.00996 (2019).

[2] Mou, Wenlong, et al. "High-order Langevin diffusion yields an accelerated MCMC algorithm." arXiv preprint arXiv:1908.10859 (2019).

Minor comments:
- The reference for Song et al.'s SDE paper is dated as ICLR 2020, rather than 2021.
- Penultimate para of sec 1: "diffusionand" -> "diffusion and"
- Section B.2: "SGMs" does not need to be repeated in sentence 1.
- One of the $\Sigma_{t}^{xv}$ matrices in Equation (9) should $\Sigma_{t}^{vt}$? This notation is used at the end of Section B.2, so consistency would make things a little smoother.
- Sec 5.1: "We are significantly outperforming this model" could be rephrased.
- Sec 5.1 "outperforms" -> "outperform"
- Sec 5.2, para 2: "as backbone" -> "as a/the backbone"
- Tab. 2 caption: "denotes" -> "denoted"
- A.1: "very inefficient too" -> "very inefficiently too"
- Sec B2, penultimate para. and Equation (67): What is $\Sigma_{t}^{zz}$? I assume this is a typo?

**Summary Of The Paper:**

The authors propose critically-damped Langevin dynamics (CLD) for score-based generative modeling. This consists of a higher-order Langevin dynamics scheme with particle velocity and position coupled to each other, as in Hamiltonian dynamics. The Langevin dynamics is critical in the sense that it is neither over- nor under-damped. A corresponding score matching objective is derived as an objective, with proof given that it is simply necessary to approximate the score of the velocity given the position. Empirical evidence is provided that this score is easier to estimate on a synthetic example. As DSM is analytically intractable for the higher-order scheme, Hybrid Score Matching (HSM) is proposed and the integration integral to this objective is addressed with a new numerical integration scheme. This approximation scheme, called Symmetric Splitting CLD Sample (SSCS), decomposes the SDE to be integrated into a tractable expression and a (hopefully small) Euler-Maruyama integration for improved accuracy (although still first order) overall. Synthetic examples are used in both the main text and the supplementary material to motivate the theory. Benchmark image datasets exhibit exceptionally strong performance, with improved sample efficiency after training and robust hyperparameters.

**Summary Of The Review:**

This paper is exceptionally well put together, in terms of derivations, presentation, and experimentation. The narrative is well-written, well-motivated, and logical throughout. To the best of my abilities, the proofs are entirely correct. The only way I can see to improve the experiments is to offer the authors more compute or data.

As a result, I believe this paper to be of the highest quality and recommend that it is given particular attention at the conference.

---

> ### Author Response · Authors · 2021-11-13
> **Thank you for questions and feedback**
>
> We thank the reviewer for the very positive feedback. We are delighted that the reviewer considers our work of "highest quality" and we appreciate the praise with regards to visualizations, related work section, and overall organization and presentation. We hope that our work provides a novel perspective on score-based generative models and stimulates further research and follow-up work that leverages insights from statistical mechanics or the MCMC literature.
>
> We also appreciate the pointers to the references. We have included them in the updated manuscript. In particular the third-order scheme of [1] is very interesting—it would be interesting to explore it in the context of diffusions for score-based generative models in follow-up work.
>
> Lastly, we are grateful for pointing out the typos that we missed. In our updated version, we implemented necessary changes. Also note that we added Appendices B.6, C.1, and F.1 to the revised manuscript to address the other reviewers' questions and provide further insights.
>
> References:
>
> [1] Mou et al. 2021. High-Order Langevin Diffusion Yields an Accelerated MCMC Algorithm. Journal of Machine Learning Research.

---

### Official Review · Reviewer_95SQ · 2021-11-03

**Correctness:** 3
**Technical Novelty And Significance:** 4
**Empirical Novelty And Significance:** 3
**Recommendation:** 8
**Confidence:** 4

**Main Review:**

This is a paper well-done. Diffusion models require designing (or sometimes training) an inference process that brings the data distribution closer to some tractable prior. Working on an augmented state-space to introduce a momentum (velocity) variable to simulate damped Langevin dynamics is very a natural extension that complements well the current score-based diffusion-based models literature. The authors then went on exploring several benefits of such an extended framework, including a smoother (conditional) score function to approximate, a novel smoother (score-matching) loss function to optimize, and a specialized numerical integrator for efficient sampling. I believe the novelties in all different aspects introduced in this work could largely benefit the community.

Overall, this is a great paper that is easy to follow and has multiple neat ideas. Strengths of the paper include
* Novelty: although it is a straightforward extension, the authors explored different aspects of the central idea of augmentation/acceleration and proposed a new loss that is smooth and a numerical method that works better for certain settings.
* Clarity: the paper is well written, ideas presented with clear theoretical development. Most of the time the design choices are explained, leaving room for future research on some relevant topics (such as optimizing for NLL, and adaptive solvers).
* Impact: the problem the proposed method solves is very specialized, particularly for SGDMs. The idea of state-space augmentation could potentially mitigate the exploding score of DSM-type training and can be a principled way of designing an SGDM that naturally admits faster numerical simulation.

I do not have any serious criticisms, but I do have the following (more detailed) questions / comments:
1. There is a paragraph in Section 3 describing the trade-off between being overdamped and underdamped. The explanation is quite intuitive, but is there any more quantitative evidence for opting for critical damping (either in theory or empirically)? I suppose an optimal ratio between $\Gamma$ and $M$ will be problem-specific. Would such generality be useful in practice?
2. A missing point about using damped Langevin dynamics is the potential speedup in convergence to the prior distribution (A la Ma et al. 2019, Proposition 1). Diffusing along with the momentum variable could allow for significant acceleration, which might reduce the integration time T needed for sampling as well.
3. For HSM, is marginalizing out $v_0$ the key to smoothing out the loss function (compared to DSM)? If so, is this a form of Blackwellization, and does the gradient estimate always have a smaller variance compared to DSM?
4. In 3.3, Leimkuhler & Reich, 2005 was cited for Euler’s methods not being suitable for Hamiltonian dynamics. Is there any intuition behind this? Can the standard midpoint integration scheme be employed which is well suited for Hamiltonian systems?
5. For SSCS, can the second half step of the current iteration be merged with the first half step of the next iteration, similarly to leap-frog integration.
6. Experiment-wise, are the numbers presented in the tables with one random seed (for experiment and for evaluation)? Just reading the number, I wouldn’t say the difference between EM and SSCS for large n in Table 3 is negligible, but perhaps that’s more due to the stochasticity of the nature of FID?
7. About $p(v_0)$, the reason for worse NLL for small $\gamma$ could be due to the fact that the upper bound requires subtracting the entropy of the augmented distribution (which will be smaller). It’d be worth noting that this could be potentially optimized for NLL. Aside from that, smaller $\gamma$ would also mean the magnitude for the score at $t\rightarrow0$ is also larger?

Ma et al 2019: Is There an Analog of Nesterov Acceleration for MCMC?


**Summary Of The Paper:**

This paper proposes a critically-damped version of score-based diffusion models, by extending the inference process to an augmented state-space and diffusing the data coupled with an auxiliary velocity variable. The authors further proposed a hybrid score matching loss for training the reverse generative process, which provides an empirical advantage of learning a conditional score function that does not blow up (e.g. to infinity) and therefore stabilizes training. For sampling, the authors derived a new numerical integration method based on first principles of statistical mechanics and MD, which they found has a better convergence behavior in practice if the computing budget is limited.


**Summary Of The Review:**

This is a timely paper that explores acceleration methods for diffusion-based models. The authors also explored different aspects / benefits of the proposed framework which I think will benefit the community of researchers working on SGDMs, including the smoothness of loss function and a new numerical integration method.

My recommendation is on the lower band of 8: accept, good paper.

---

> ### Author Response · Authors · 2021-11-13
> **Thank you for questions and feedback (1)**
>
> We thank the reviewer for the thoughtful feedback. We are delighted that the reviewer appreciates our contributions.
>
> Below we address specific questions and comments:
>
> - **"There is [...]".** This question concerns the motivation for critically-damped Langevin dynamics as opposed to the over- or under-damped versions.
>
>   First, we would like to point the reviewer to our App. A, in particular A.1 and Fig. 5, where we extensively visualize and discuss some intuitions about under-, over- and critically-damped Langevin dynamics.
>
>   Furthermore, over-, under-, and critical-damping can be understood very intuitively in the non-stochastic setting, where we have no noise component in the SDE: In that situation, our "diffusion without noise" simply corresponds to a damped Harmonic oscillator [1] (for each dimension), initialized far from equilibrium. It can be shown that the system will return to its rest or equilibrium position most quickly for critical damping [2,3,4] (this can also be found in standard text books, as cited in our paper, but these sources may quickly provide the necessary intuitions). In contrast, over-damping is slow and under-damping leads to undesired oscillatory behavior. Hence, critical damping is exactly what we need in our situation when designing the forward diffusion for training SGMs: We seek a diffusion process which brings us to equilibrium as quickly as possible. Compared to the Harmonic oscillator, we have additional noise injection, but the overall intuition remains the same. Also note that the critical damping coefficient is universal and only depends on the dynamic system, this is, the forward differential equation, but not the data distribution we are interested in learning (the data itself merely defines the initialization).
>
> - **"A missing [...]".** We tried to highlight throughout the paper that critically-damped Langevin dynamics leads to faster convergence to the stationary distribution and we refer the reviewer to the comparison of different Langevin dynamics in Appendix A as well as the reply to the previous point.
>
>   However, thank you for bringing [5] to our attention. We believe the analyses in [5] can be nicely complementary to the analyses in our paper. Hence, we included it in our Related Work (Section 4 of the paper).
>
>   We would also like to mention that in our CLD the effective integration time is already significantly reduced compared to the VPSDE: Both CLD and VPSDE integrate up to $T=1$. However, the effective total integration time is determined by $\beta(t)$, which can be interpreted as a time rescaling. In our case, we have a constant $\beta=4$ for all $t\in[0,1]$, whereas for the VPSDE $\beta(t)$ linearly grows from 0.01 to 20.0. Hence, for most $t$, the VPSDE's $\beta(t)$ is much larger than CLD's constant $\beta=4$. This means that the VPSDE integrates over an effectively longer time interval overall. This is the case, because it converges significantly slower than our quickly mixing CLD and therefore needs to be run for longer.
>
>   Also see Fig. 5 in the Appendix in that regard.
>
> - **"For HSM [...]".** Indeed, marginalizing out $\mathbf{v}_0$ is the key to smoothing out the loss function compared to DSM. However, to the best of our knowledge, this is not a form of Rao--Blackwellization (which usually corresponds to variance reduction by conditioning on a sufficient statistic) in the strict sense. Marginalization of $\mathbf{v}_0$ simply corresponds to replacing the single sample-based Monte Carlo estimate of DSM with an analytical distribution estimate. Therefore, this procedure always strictly reduces the variance of the gradient estimate.
>
>   In fact, we performed an additional experiment, where we compare the gradient variance of HSM and DSM (tested on a CIFAR-10 model) empirically. This experiment is presented in App. C.1. The variance comparison visualized in Fig. 7 validates that HSM reduces the gradient variance in comparison to DSM.

---

> > ### Author Response · Authors · 2021-11-13
> > **Thank you for questions and feedback (2)**
> >
> > - **"In 3.3 [...]".** Two fundamental properties of Hamiltonian dynamics are that they are volume preserving and reversible [6] (more formally, Hamiltonian systems have symplectic phase space structure). In contrast to, for example, the Leapfrog method (which is a symplectic integrator), Euler's method does not conserve volume and is not reversible. With regards to the midpoint method, the implicit midpoint method is also symplectic and therefore suited for Hamiltonian systems, while the explicit version is not. We also refer the reviewer to Section 2.3 of [6].
> >
> >   It is worth mentioning that our generative SDE is not a standard Hamiltonian ODE, but still a complex SDE which only has a Hamiltonian component in addition to the score term and the Ornstein--Uhlenbeck process. It is therefore non-trivial how to best solve it, which is why we opted to develop a novel sampler from first principles (our SSCS). However, adapting techniques such as the implicit midpoint method for our generative SDE might be a fruitful idea, too. In fact, we hope that our work inspires future research into such directions.
> >
> > - **"For SSCS [...]".** Yes, the reviewer is absolutely correct: these steps can be combined. In our work, these steps have however virtually no computational cost (compared to the costly evaluation of the neural network in the "center" step). Therefore, for ease of notation, we did not combine the steps in our presentation. However, we now added a corresponding comment at the end of App. D.2.
> >
> > - **"Experiment-wise [...]".** Yes, as is standard practice in the generative modeling literature both sampling and training is only done for a single random seed. Due to time and compute constraints, we are not able to retrain models over different random seeds. However, we repeated FID computation for the smaller ablation model five times and found that FID scores did not vary much; the obtained scores are 3.14, 3.14, 3.16, 3.15, and, 3.14. We therefore believe that our FID estimates are fairly low variance.
> >
> > - **"About $p(\mathbf{v}_0)$ [...]".** Yes, the reviewer is absolutely correct with regards to worse NLL for smaller $\gamma$ being due to the entropy term in the NLL bound; we also touch upon this issue in App. B.5, to which we refer for further discussion of this topic. There is certainly much room to further improve and tailor our approach towards maximum likelihood training and we would be delighted to see such work in the future.
> >
> >   In this work, however, we solely focused on perceptual image quality as commonly measured by FID and only calculated likelihood bounds to confirm that we are not dropping modes. The score function that needs to be learned at $t=0$ is $-\mathbf{v} \gamma^{-1} M^{-1}$, and therefore, as pointed out by the reviewer, decreasing $\gamma$ leads to an increased magnitude of the score function. However, we want to remind that we use a mixed score parameterization: at time $t=0$ the neural network needs to simply learn the zero function for *any* data distribution (see Sec. 3.2).
> >
> > We invite the reviewer to follow up on our above replies. Overall, we added Appendices B.6, C.1, and F.1 to the revised manuscript to address the reviewers' questions and to provide further insights.
> >
> > References:
> >
> > [1] <https://en.wikipedia.org/wiki/Harmonic_oscillator>
> >
> > [2] <https://en.wikipedia.org/wiki/Harmonic_oscillator#Damped_harmonic_oscillator>
> >
> > [3] <https://en.wikipedia.org/wiki/Damping>
> >
> > [4] <https://physics.stackexchange.com/questions/106091/faster-than-critical-damping-for-harmonic-oscillator>
> >
> > [5] Ma et al 2019: Is There an Analog of Nesterov Acceleration for MCMC?
> >
> > [6] Neal 2012. MCMC Using Hamiltonian Dynamics. Handbook of Markov Chain Monte Carlo.

---

### Official Review · Reviewer_tK3A · 2021-11-03

**Correctness:** 3
**Technical Novelty And Significance:** 4
**Empirical Novelty And Significance:** 3
**Recommendation:** 8
**Confidence:** 4

**Main Review:**

The paper claims the following contributions.
1. First, the paper proposes a novel class of continuous-time diffusion-based generative models, called critically-damped Langevine diffusion (CLD).
2. Second, the authors show that a score matching objective for the proposed method requires only $\nabla_{v_t} \log p_t(v_t | x_t)$, not $\nabla_{(x_t, v_t)} p_t(x_t, v_t)$.
3. Third, for the proposed model, the paper proposes a variant of denoising score matching, called hybrid score matching (HSM), observing that the training of CLD-based models is unstable by using DSM.
4. Fourth, the authors propose a leapfrog-style integrator designed for CLD-based models, benefitting from Hamiltonian dynamics' symplectic structure.
5. Finally, the paper claims to provide novel insights into continuous-time diffusion-based generative models and their connections to statistic mechanics.


**Strengths of the paper**

In general, the paper's contributions wrt the novelty are clear for several reasons:
- First, the paper provides novel diffusion-based models inspired by stochastic Hamiltonian dynamics. Moreover, the paper provides sufficient proofs to support that the proposed methods are well defined. I find that the proposed method will be very interesting contribution to the generative model community.
- Second, the authors demonstrate the scalability of the proposed methods by (1) showing that the increment of the model complexity is marginal thanks to reverse-time SDEs only requires log-density gradient of conditional distributions of velocities given data, and (2) proposing a modified DSM loss suited for CLD-based models.

In addition, I found that the paper has a well-organized structure so that it is clear to understand the proposed and other practical techniques to improve training.

**Weaknesses of the paper**
The following aspects of the paper can be improved.

- First, the discussion about the proposed integrator can be improved. The authors emphasize that the proposed leapfrog-style integrator for CLD-based models can generate better quality samples even when the number of discretization steps is small compared to the Euler-Maruyama method. However, it isn't easy to assess the statement based on the results described in Tables 2 and 3. In the experiments, the results are dependent on both the integrators and the qualities of learned models. To resolve this issue, one can add toy experiments to solidify the argument, where ground truth reverse-time SDEs are (approximately) accessible; thus, the resulting analysis can highlight the difference of the integrators.

- Second, the paper claims that the proposed method only needs to learn smoother signals, and thus the models can perform better. However, in the current submission, the discussion about this seems limited.

  While the authors have already discussed the property by using toy experiments (such as in Fig. 2), the benefits of the proposed model seem marginal in high-dimensional datasets.

  I assume that dropping weights to improve the generation qualities has diluted the benefits of the proposed method. Thus I believe that discussion about the proposed methods wrt maximum likelihood training can strengthen the significance of the proposed method. In my understanding, dropping weights for training diffusion-based generative models prevents modeling unbounded scores at $t$'s close to the data, where the weights are extremely high. As a result, the previous methods don't learn unbounded scores in practice, which can be a potential reason for the marginal benefits of the proposed method in high-dimensional settings. Therefore, the proposed method may benefit from not dropping weights, possibly resulting in high-quality samples being achieved with maximum likelihoods training. I anticipate that the comparison can be more observable if the models are trained in real space after logit transformation + uniform dequantization.

  Moreover, it can be helpful to add discussions about the trade-off between the variance of the DSM/HSM objective estimates and the variance of the $p_0(v_0)$. For example, I imagine that the high variance of $p_0(v_0)$ results in the high variance in $p_t(v_t|x_t)$, implying that the models only need to learn smoother scores. Again, I assume this aspect can be more observable if the models are trained to maximize likelihoods.

- Third, the motivation of Hamiltonian dynamics-style generative models can be improved. Aside from the applicability of the leapfrog-style integrator for the proposed method, the authors claim that the velocity in CLDs accelerates mixing in the diffusion process (for example, in Sec 4). However, it is unclear how important is this "mixing" in the context of diffusion-based generative models.
    In standard MCMC methods, the mixing properties are important since such methods aim at matching the distribution of recorded states (from a generated chain) to target distributions. On the other hand, the diffusion-based generative models rely on the distributions of collection of chains, esp. the chains' last states.
Knowing that the previous diffusion-based generative models are already expressive and capable of generating diverse samples, it is difficult to motivate the proposed method while its novelty.
    I believe that improving the discussion about the proposed integrators will resolve the issue, regardless of the proposed method's connection to statistic mechanics.

- Finally, the discussion about the motivation for HSM can be improved. In Sec 5.3, the authors state that for the proposed method, training with regular DSM is unstable. While the authors state some reasoning about the instability, but the analysis about it is limited. I believe that the single sample Monte Carlo estimate for the regular DSM objective can have a high variance for the proposed method, and thus I believe that the marginalizing out $v_0$ in HSM has reduced the variance. I consider that additional toy experiments that discuss the difference between DSM vs. HSM wrt their variance will further motivate the importance of HSM for CLD-based models.


**Summary Of The Paper:**

The paper proposes a novel class of continuous-time diffusion-based generative models. Specifically, the paper proposes forward diffusions that are simple forms of stochastic Hamiltonian dynamics, unlike the previous methods, where linear stochastic differential equations (linear SDE) diffuse the data distribution. Thus, the proposed forward diffusion transport a joint distribution of data $x_0$ and auxiliary random variables $v_0$ to a prior joint distribution. Then, the generative models are their reverse-time diffusion processes on the joint space whose initial condition is the prior joint distribution.

In the Hamiltonian dynamics-type forward diffusion, the auxiliary random variable can be interpreted as velocity (as in physics), while data corresponds to position. The forward diffusion updates data by the velocities without any noise, while the velocities are updated via linear SDEs, similarly to deterministic Hamiltonian dynamics. In particular, the paper shows that the data distribution will be transported to a prior distribution and that velocities will map to a prior distribution similar to previous approaches. The prior joint distributions are commonly chosen to simple distributions, such as standard Normal distributions.

The authors emphasize two interesting properties of the proposed method. First, the paper points out that the reverse-time SDEs should only contain $\nabla_{v_t} \log p_t(v_t| x_t)$, but not logarithmic gradient wrt $x_t$, which implies that the model complexity (the size of models) is almost similar to the previous approaches. Second, the authors shows that $p_t(v_t| x_t)$ is Gaussians for all $t \in [0, T]$, including $p_0(v_0|x_0) = p_0(v_0)$. This indicates that $\nabla_{v_t} \log p_t(v_t| x_t)$ is potentially bounded unlike previous diffusion-based generative models, where the scores closer to data are possibly unbounded. As a result, esp. with common practice to learn scores at all time $t$'s by a single neural networks, training the proposed model will be more stable.

In order to learn the $\nabla_{v_t} \log p_t(v_t| x_t)$, the paper first propose to use denoising score matching (DSM) following previous diffusion-based generative models. However, observing that training the proposed model via DSM is unstable, the paper proposes a modified objective called hybrid score matching (HSM). HSM can be obtained by marginalizing out $v_0$ from score matching loss similarly to deriving DSM; the marginalization is possible as the auxiliary random variable defined to follow a known distribution, such as Normal distribution.

In addition, as the reverse-time diffusion is also a Hamiltonian SDE, the paper proposes a new integrator for generations, benefitting from the symplectic structure of the Hamiltonian dynamics. Note that it has been well appreciated that the discretization methods that utilize symplectic structure are more accurate than the Euler-Maruyama method under similar computational costs. Thus, the proposed integrator will have a better quality of samples, esp. when the number of discretization is small.

In order to show the effectiveness of the proposed method, the paper runs three main experiments: First, with toy experiments, the authors demonstrate that learning $\nabla_{v_t} \log p_t(v_t| x_t)$ is less difficult in comparison to learning the scores wrt the data directly. Then, the authors evaluate the generation qualities of the proposed models on two image modeling benchmark datasets; CIFAR-10 and CelebA-256. Finally, the paper demonstrates the proposed integrator's effectiveness compared to the Euler-Maruyama method, including additional analysis ablating the effect of hyperparameters of the proposed integrator.

**Summary Of The Review:**

In general, the paper's contributions wrt the novelty are clear, and the proposed methods are well-defined. In addition, I found that the paper has a well-organized structure so that it is clear to understand the proposed and other practical techniques to improve training. Thus, I'm inclined to accept the paper. However, I found that several aspects of the submission can be improved. I'm inclined to improve my evaluation if the aforementioned weak points are well addressed.


===== POST-REBUTTAL COMMENTS ========

The rebuttal had addressed most of my concerns. Consequently, I raised my score from 6 to 8, and I also raised the "Empirical Novelty And Significance" score from 2 to 3.

---

> ### Author Response · Authors · 2021-11-13
> **Thank you for questions and feedback (1)**
>
> We thank the reviewer for their thoughtful feedback. We are delighted that the reviewer agrees with us that novel diffusions are a fruitful contribution to the literature on score-based generative models and diffusion models. We think that the reviewer raised several excellent questions and we believe that answering those questions has significantly improved our work. Consequently, we made multiple additions to the revised manuscript (due to the space limitations, primarily in the Appendix). Overall, we added Appendices B.6, C.1, and F.1.
>
> Below we address specific questions and comments:
>
> - **"First, the discussion [...]".** This comment concerns the comparisons between the different integrators, this is, our Symmetric Splitting CLD Sampler (SSCS) and Euler-Maruyama (EM). We would like to address this comment by **(i)** discussing Table 9 (the extended version of Table 3) and by **(ii)** sharing the results of the new experiment proposed by the reviewer.
>
>   **(i)** Score-based generative models essentially involve three ingredients: the learnt score model itself (including the trained neural network), the diffusion (this is, the stochastic differential equations [SDEs] for the forward process as well as the corresponding backward process), and the chosen numerical sampler (to solve the backward SDE). Now let us look at Table 9 (or the short version in Table 3): Given a learnt score model and diffusion process, we can assess sampling quality for different samplers. In particular, the results in rows one to four (CLD) *are based on the exact same diffusion and learnt score model* (including trained neural networks, such that the "quality" of the learnt model is the same for these rows), with the *only* difference being the employed integrator. Therefore, by studying these rows, we believe that we can come to the conclusion that SSCS generally outperforms EM for CLD on small $n$. In particular, we find that SSCS-QS significantly outperforms EM and EM-QS for $n \in \\{50, 150, 275, 500\\}$, while for $n \in \\{1000, 2000\\}$ EM-QS and SSCS-QS perform similar with a tiny advantage for EM-QS.
>
>   **(ii)** Nevertheless, we agree with the reviewer that it is instructive to also run a toy experiment for which we know the ground truth scores and study the integrators in this setting. As suggested by the reviewer, we designed such an experiment. In particular, we consider the two-dimensional mixture of Normals from Appendix E.1, for which we know the score function analytically (Fig. 9 (a) shows the ground truth distribution; the ground truth score function then simply corresponds to a mixture of scores of Normals). Therefore, we also have access to the exact generative SDE. CLD together with SSCS consistently outperforms VPSDE with EM (see the visualizations in Fig. 10 as well as negative log-likelihoods in Tab. 8). We refer the reviewer to Appendix F.1.1 for details of the experiment and its results.

---

> > ### Author Response · Authors · 2021-11-13
> > **Thank you for questions and feedback (2)**
> >
> > - **"Second, the paper [...]''.** The reviewer's core point of concern here is that dropping weights to improve synthesis quality has potentially diluted the benefits of the proposed method. Therefore, it is suggested to investigate the method under training towards maximum likelihood.
> >
> >   We agree that studying the maximum likelihood training setting in more detail would be very insightful. Unfortunately, due to time and compute constraints, it is not possible to retrain models on large-scale image datasets on short-notice; therefore, we leave this task to future work. Instead, we run multiple 2D toy experiments to compare our method to VPSDE for maximum likelihood training.
> >
> >   **(i)** We re-did the experiment from Fig. 2 (bottom) for the maximum likelihood setting. As can be seen in Fig. 11, the reviewer is correct that increasing $\gamma$ generally leads to simpler neural networks (this is, smooth as measured by the average Frobenius norm of Jacobians for varying $t$). However, using our mixed score formulation, even for $\gamma=0.04$ CLD models are still significantly less complex than the VPSDE model. We refer the reviewer to F.1.2 (*Complexity Experiment*) for more details.
> >
> >   **(ii)** Furthermore, we tested CLD-based SGMs and VPSDE-based SGMs on a challenging toy dataset with maximum likelihood training. We found that CLD significantly outperforms VPSDE. We refer the reviewer to F.1.2 (*Challenging Multi-Scale Toy Dataset*) for more details. These experiments support the strength of CLD also in the maximum likelihood training setting and the results are in line with reviewer's intuitions.
> >
> >   **(iii)** We are not entirely sure what exactly the reviewer means by "the trade-off between the variance of the DSM/HSM objective estimates and the variance of the $p(\mathbf{v}_0)$".
> >
> >   However, the question might be related to the reviewer's last point and we therefore refer to another additional experiment we ran in that context, which may provide insights: We explicitly estimated the difference of score function gradient variances between the HSM and DSM objectives for a CIFAR-10 model. We found that HSM leads to reduced gradient variance for $\gamma = 1$ (for all $t$) and $\gamma = 0.04$ (mostly for small $t$). That the gap is less pronounced for small $\gamma$ is expected, as for small $\gamma$ the input velocity distribution becomes lower entropy, such that using a sample (as in DSM) vs. marginalizing over the full distribution (as in HSM) behave more similarly. This gradient variance reduction of HSM holds for the "reweighted" training objective as well as for maximum likelihood training. The gradient variance for these two objectives is equivalent up to a time-dependent multiplier. For small $t$, in the maximum likelihood setting this multiplier is usually very large relative to the "reweighted setting". Eventually, this implies a significant difference in gradient variance between the HSM and DSM objectives in this maximum likelihood setting. This supports our CLD-based approach also for maximum likelihood training. This seems to be in line with the reviewer's feedback. We refer the reviewer to Appendix C.1 for more details and discussion.

---

> > > ### Author Response · Authors · 2021-11-13
> > > **Thank you for questions and feedback (3)**
> > >
> > > - **"Third, the motivation [...]".** The question is about the motivation for adding the Hamiltonian component into the diffusion process.
> > >
> > >   To provide some intuitions, let us recall how the data distribution evolves in the forward diffusion process of SGMs: The role of the diffusion is to bring the initial non-equilibrium state quickly towards the equilibrium or prior distribution. Suppose for a moment, we could do so with "pure" Hamiltonian dynamics (no noise injection). In that case, we could generate data from the backward model without learning a score or neural network at all, because Hamiltonian dynamics is analytically invertible (flipping the sign of the velocity, we can just integrate backwards in reverse time direction). Now, this is not possible in practice, since Hamiltonian dynamics alone usually cannot convert the non-equilibrium distribution to the prior distribution. Nevertheless, Hamiltonian dynamics essentially achieves a certain amount of mixing on its own; moreover, since it is deterministic and analytically invertible, this mixing comes at no cost in the sense that we do not have to learn a complex score function to invert the Hamiltonian dynamics. Our thought experiment clearly shows that we should strive for a diffusion process that behaves as deterministically (meaning that deterministic implies easily invertible) as possible with as little noise injection as possible.
> > >
> > >   And this is exactly what is achieved by adding the Hamiltonian component in the overall diffusion process. In fact, recall that it is the diffusion coefficient **$G$** of the forward SDE that ultimately scales the score function term of the backward generative SDE (and it is the score function that is hard to approximate with complex neural nets). Therefore, in other words, relying more on a deterministic Hamiltonian component for enhanced mixing (mixing just like in MCMC in that it brings us quickly towards the target distribution, in our case the prior) and less on pure noise injection will lead to a nicer generative SDE that relies less on a score function that requires complex and approximate neural network-based modeling, but more on a simple and analytical Hamiltonian component. Such SDE can then be solved easier with an appropriate integrator (like our SSCS). In the end, we believe that this is the reason why our networks are "smoother" and why given same network capacity and limited compute budgets we essentially outperform all previous results in the literature (on CIFAR-10). However, CIFAR-10 is somewhat saturated already in terms of performance, as it is such a widely used benchmark. We believe that our diffusion will be particularly promising for future work on more large-scale images and complex data.
> > >
> > >   We would also like to offer a second perspective, more directly inspired by the MCMC literature. In MCMC, "mixing" helps to quickly traverse the high probability parts of the target distribution and, if an MCMC chain is initialized far from the high probability manifold, to quickly converge to this manifold. However, this is precisely the situation we are in with the forward diffusion process of SGMs: The system is initialized in a far-from-equilibrium state (the data distribution) and we need to traverse the space as efficiently as possible to converge to the equilibrium distribution, this is, the prior. Without efficient mixing, it takes longer to converge to the prior, which also implies a longer generation path in the reverse direction—which intuitively corresponds to a harder problem. Therefore, we believe that ideas from the MCMC literature that accelerate mixing and traversal of state space may be beneficial also for the diffusions in SGMs.
> > >
> > >   We added these discussions in App. B.6. Generally, the reviewer made a valuable point. We believe that these are excellent questions that could be studied more rigorously in future research.
> > >
> > > - **"Finally, the discussion [...]".** Thank you for bringing this to our attention. As mentioned above, we designed a new experiment to measure the difference between parameter gradient variances of DSM vs. HSM. We find that HSM leads to significantly reduced gradient variance. As pointed out by the reviewer, this further motivates the use of HSM for training CLD-based SGMs. We refer again to Appendix C.1.
> > >
> > > We hope that we were able to address all questions and concerns. We added the new experiments and discussions into the Appendix, due to space limitations in the main paper and in the interest of providing a fast reply here. However, we may re-factor the manuscript for the camera-ready version. If our replies feel satisfactory, we would like to kindly ask the reviewer to consider raising the score accordingly. Otherwise, we would be happy to further discuss.

---

> > > > ### Comment · Reviewer_tK3A · 2021-11-22
> > > > **Reviewer response**
> > > >
> > > > Thank you for the thoughtful response and the updates.
> > > >
> > > > I find it might be helpful to clarify one point that I made in my review, related to **"(ii) Nevertheless, we agree with ..."** in the authors' feedback (1). As the authors point out in the paper that **"Euler methods are not well-suited for Hamiltonian dynamics ..."**, I understood that the proposed integrator provides more accurate results than Euler methods for CLD-based models. My point was that it would be helpful to see some toy experiments results that support this when one doesn't have errors in models. Thus, in Figure 10, it would be more helpful to see the difference between CLD w/ CLSS and CLD w/ EM instead of comparing CLS w/ CLSS and VPSDE w/ EM (similar to Figure 1 in [1]).
> > > >
> > > > Nevertheless, the rebuttal had addressed most of my concerns. Consequently, I will raise my score from 6 to 8.
> > > >
> > > >
> > > > [1] R. M. Neal, '12, MCMC Using Hamiltonian Dynamics. Handbook of Markov Chain Monte Carlo.

---

> > > > > ### Author Response · Authors · 2021-11-23
> > > > > **Author response**
> > > > >
> > > > > We would like to thank the reviewer for the reply and for clarifying the suggested experiment. We have just updated the manuscript with the proposed experiment: In particular, Section F.1.1 now also includes an experiment that uses EM sampling for the CLD-based model with the analytically known score, as suggested. We refer to the updated Table 8, Figure 10 and text in F.1.1.
> > > > >
> > > > > The results show that EM performs poorly when applied to sample from the CLD-based SGM, both quantitatively (negative log-likelihood in Table 8 ) and qualitatively (too broad distributions in Figure 10). EM performs particularly poorly when using few, but large synthesis steps. These findings are in line with the “diverging” dynamics that is often observed when solving Hamiltonian dynamics with a non-symplectic integrator,  such as the standard Euler method (demonstrated, for example, in [1]). We thank the reviewer for bringing this missing detail to our attention; we believe that these additional results clearly motivate our novel SSCS sampler.
> > > > >
> > > > > We thank the reviewer for updating their score.
> > > > >
> > > > > [1] R. M. Neal, '12, MCMC Using Hamiltonian Dynamics. Handbook of Markov Chain Monte Carlo.

---

### Author Response · Authors · 2021-11-19
**Discussion period ending soon**

Dear reviewers, \
we would like to kindly remind you that the Phase 1 discussion period ends in 3 days. It would be great if you could have a look at our replies and let us know if our replies are satisfactory or if there are any further follow-up questions. We would appreciate any feedback and would be happy to further discuss.

Thank you very much, \
the authors of “Score-Based Generative Modeling with Critically-Damped Langevin Diffusion”

---

### Decision · Program_Chairs · 2022-01-20

**Decision:**

Accept (Spotlight)

**Comment:**

The paper develops a diffusion-process based generative model that perturbs the data using a critically damped Langevin diffusion. The diffusion is set up through an auxiliary velocity term like in Hamiltonian dynamics. The idea is that picking a process that diffuses faster will lead to better results.The paper then constructs a new score matching objective adapted to this diffusion, along with a sampling scheme for critically damped Langevin score based generative models. The idea of a faster diffusion to make generative models is a good one. The paper is a solid accept.

Reviewer tK3A was lukewarm as evidenced by their original 2 for empirical novelty that moved to a 3. From my look, it felt like a straightforward application of ideas in one domain, sampling, to another, generative modeling. It's a good paper, but it does not stand out relative to other accepts.